# Investigation of the post-2007 methane renewed growth with high-resolution 3-D variational inverse modelling and isotopic constraints

Joël Thanwerdas[1,a*], Marielle Saunois[1], Antoine Berchet[1], Isabelle Pison[1], and Philippe Bousquet[1]

[1]Laboratoire des Sciences du Climat et de l'Environnement, CEA-CNRS-UVSQ, IPSL, Gif-sur-Yvette, France.
[a]now at: Empa, Swiss Federal Laboratories for Materials Science and Technology, Dübendorf, Switzerland.

**Correspondence:** J. Thanwerdas (joel.thanwerdas@empa.ch)

**Abstract.**

We investigate the causes of the renewed growth of atmospheric methane ($CH_4$) amount fractions after 2007 by using variational inverse modelling with a three-dimensional chemistry-transport model. Together with $CH_4$ amount fraction data, we use the additional information provided by observations of $CH_4$ isotopic compositions (in $^{13}C{:}^{12}C$ and in D:H) to better differentiate between the emission categories compared to assimilating $CH_4$ amount fractions alone. Our system allows us to optimize either the $CH_4$ emissions only or both the emissions and the source isotopic signatures ($\delta_{source}(^{13}C, CH_4)$ and $\delta_{source}(D, CH_4)$) of five emission categories. Consequently, we also assess here for the first time the influence of applying random errors to both emissions and source signatures in an inversion framework. As the computational cost of a single inversion is high at present, the methodology applied to prescribe source signature uncertainties is simple so that it serves as a basis for future work. Here, we investigate the post-2007 increase in atmospheric $CH_4$ using the differences between 2002-2007 and 2007-2014. When random uncertainties in source isotopic signatures are accounted for, our results suggest that the post-2007 increase (here defined using the two periods 2002-2007 and 2007-2014) in atmospheric $CH_4$ was caused by increases in emissions from 1) fossil sources (51 % of the net increase in emissions) and 2) agriculture and waste sources (49 %), slightly compensated by a small decrease in biofuels-biomass burning emissions. These conclusions are very similar when assimilating $CH_4$ amount fractions alone, suggesting that either random uncertainties in source signatures are too large at present to bring any additional constraint to the inversion problem or we overestimate these uncertainties in our setups. On the other hand, if the source isotopic signatures are considered perfectly known (i.e., ignoring their uncertainties), the relative contributions of the different emissions categories are significantly changed. Compared to the inversion where random uncertainties are accounted for, fossil emissions and biofuels-biomass burning emissions are increased by 24 % and 41 %, respectively, on average over 2002-2014. Wetlands emissions and agriculture and waste emissions are decreased by 14 % and 7 %, respectively. Also, in this case, our results suggest that the increase in $CH_4$ amount fractions after 2007 was caused, despite a large decrease in biofuels-biomass burning emissions, by increases in emissions from 1) fossil fuels (46 %), 2) agriculture and waste (37 %) and 3) wetlands (17 %). Additionally, some other sensitivity tests have been performed. While prescribed OH inter-annual variability can have a large impact on the results, assimilating $\delta(D, CH_4)$ observations in addition to the other constraints have a minor influence. Using all the information derived from these tests, the net increase in emissions is still primarily attributed

to fossil sources ($50 \pm 3$ %) and agriculture and waste sources ($47 \pm 5$ %). Although our methods have room for improvement, these results illustrate the full capacities of our inversion framework, which can be used to consistently account for random uncertainties in both emissions and source signatures.

## 1 Introduction

Atmospheric methane ($CH_4$) has a large influence on both climate and atmospheric chemistry. The globally averaged tropospheric $CH_4$ amount fractions has been multiplied by 2.6 since pre-industrial levels (Gulev et al., 2021) and reached a new high of 1895 $nmol \, mol^{-1}$ in 2021 (global average from marine surface sites; Lan et al., 2023). Neglecting indirect effects related to ozone, water vapor and nitrogen oxides production, this large increase in $CH_4$ amount fractions since the pre-industrial era contributes to 16 % of the current radiative forcing from well-mixed greenhouse gases (carbon dioxide, methane, nitrous

oxide, halogens) (Forster et al., 2021). $CH_4$ has therefore the second largest contribution to the additional greenhouse effect behind carbon dioxide ($CO_2$). $CH_4$ amount fractions increased quasi-continuously since the pre-industrial era but stabilized between 1999 and 2006. The growth resumed after 2007, at a rate exceeding 10 $nmol \, mol^{-1} \, a^{-1}$ for some years. Nisbet et al. (2019) pointed out that the $CH_4$ burden dramatic increase is contrary to pathways compatible with the goals of the 2015 United Nations Framework Convention on Climate Change Paris Agreement and that urgent action is required to bring $CH_4$ back to

a pathway more in line with the Paris goals. A proper understanding of the $CH_4$ budget could highly facilitate such actions by increasing the effectiveness of mitigation policies.

$CH_4$ is emitted into the atmosphere by multiple sources (wetlands, livestock, rice cultivation, waste, fossil fuels exploitation, biomass burning...), with distinct processes involved (microbial, thermogenic, pyrogenic). This species is mainly removed from the atmosphere through oxidation by the radical hydroxyl (OH), which represents about 92 % of the total sink (Saunois et al.,

2020; Thanwerdas et al., 2022b). Other sinks such as oxidation by atomic oxygen ($O^1D$), chlorine (Cl) and methanotrophs in the soil contribute about 1.5 %, 1.5 %, and 5 %, respectively, to the total removal of $CH_4$ (Saunois et al., 2020; Thanwerdas et al., 2022b), Note that these numbers come with non-negligible uncertainties and vary from one study to another.

Estimating these sources and sinks is challenging, especially at the global scale, yet necessary to better understand the $CH_4$ budget and to anticipate its evolution. The scientific community have developed two approaches to estimate $CH_4$ emissions at

different scales. On the one hand, bottom-up approaches aim to estimate these emissions using both inventories mixing statistical activity data with emission factors for anthropogenic emissions (e.g., Höglund-Isaksson, 2012, 2017; Janssens-Maenhout et al., 2019), and process-based models for natural and fire emissions (e.g., van der Werf et al., 2017; Poulter et al., 2017). Bottom-up estimates provide valuable sectorial and regional information, albeit having their global emissions not constrained by atmospheric observations. On the other hand, top-down approaches use inversion methods (Newsam and Enting, 1988; Ent-

ing and Newsam, 1990) and chemistry-transport models (CTMs) to statistically optimize model parameters (e.g., emissions) and minimize model-observations differences (e.g., Houweling et al., 2017, and references therein). These approaches provide posterior estimates that are both consistent with atmospheric observations (e.g., $CH_4$ amount fractions) and prior estimates (typically derived from bottom-up estimates). The inversion problem is considered as "ill-posed" because a wide range of

surface flux configurations can equally explain the observational data in the atmosphere. Since the 1980s, surface monitoring
networks have nevertheless significantly increased the spatial coverage and the precision of their observations, narrowing the
range of possible flux configurations and improving the relevance of inversion methods.

Although Saunois et al. (2020) recently showed that the consistency between top-down and bottom-up estimates improved
over time, the 1999-2006 plateau and the subsequent renewed growth still generate considerable attention and controversy
(Rice et al., 2016; Schwietzke et al., 2016; Schaefer et al., 2016; Nisbet et al., 2016; Patra et al., 2016; Bader et al., 2017;
Turner et al., 2017; Rigby et al., 2017; Worden et al., 2017; Saunois et al., 2017; Morimoto et al., 2017; McNorton et al.,
2018; Thompson et al., 2018; Schaefer, 2019; Nisbet et al., 2019; Fujita et al., 2020; Zimmermann et al., 2020; Jackson et al.,
2020; Chandra et al., 2021). Most of these studies suggested that this renewed growth was partially explained by an increase in
microbial emissions (wetlands, livestock and/or rice cultivation) and some of them further located this increase in the tropics
(Nisbet et al., 2016; Patra et al., 2016; Schaefer et al., 2016; Schwietzke et al., 2016). Multiple studies also concluded that the
renewed growth was driven by an increase in both microbial and fossil fuels emissions (Rice et al., 2016; Patra et al., 2016;
Bader et al., 2017; Worden et al., 2017; Saunois et al., 2017; McNorton et al., 2018; Thompson et al., 2018; Jackson et al.,
2020; Chandra et al., 2021; Lan et al., 2021; Basu et al., 2022), albeit providing a very wide range of individual contributions.
An increase in fossil fuel emissions was also supported by an independent work using ethane-based approaches (Hausmann
et al., 2016). However, other studies found that these emissions decreased or stabilized (Schwietzke et al., 2016; Schaefer et al.,
2016; Fujita et al., 2020) over the period of the renewed growth. Some studies also found that an increase in emissions was not
the main driver and that a large decrease in OH concentrations could have explained the recent variations (Turner et al., 2017;
Rigby et al., 2017).

Such controversy partly arises from the difficulty to separate contributions from individual $CH_4$ sources. Despite the high
number of observations over some regions, many of the sources are co-located and isolating the contribution from each
source to the local increase in $CH_4$ amount fractions is challenging. Carbon and hydrogen isotope atmospheric compositions,
$\delta(^{13}C, CH_4)$ and $\delta(D, CH_4)$, can help to differentiate co-emitted emission categories because each $CH_4$ production process
(microbial, thermogenic, pyrogenic) has its own characteristic isotopic signature (Sherwood et al., 2021, 2017). $\delta(^{13}C, CH_4)$
and $\delta(D, CH_4)$ are generally defined using a deviation of the sample atomic isotopic ratio ($R_{13} = X(^{13}CH_4)/X(^{12}CH_4)$ or
$R_D = X(CH_3D)/X(CH_4)$) relative to a specific standard ratio:

$$\delta(^{13}C, CH_4) = \frac{R_{13}}{R_{PDB}} - 1 = \frac{X(^{13}CH_4)/X(^{12}CH_4)}{R_{PDB}} - 1 \tag{1}$$

$$\delta(D, CH_4) = \frac{R_D}{R_{VSMOW}} - 1 = \frac{X(CH_3D)/X(CH_4)}{R_{VSMOW}} - 1 \tag{2}$$

$X(^{12}CH_4)$, $X(^{13}CH_4)$, $X(CH_3D)$ and $X(CH_4)$ denote the $^{12}CH_4$, $^{13}CH_4$, $CH_3D$ and $CH_4$ amount fractions, respectively.
$R_{PDB} = 1.12372 \times 10^{-2}$ is here the standard ratio of Pee Dee Belemnite (PDB) (Craig, 1957) and $R_{VSMOW} = 1.5595 \times 10^{-4}$ is the Vienna Standard Mean Ocean Water (VSMOW) ratio (Hagemann et al., 1970; Wit et al., 1980). $\delta(^{13}C, CH_4)$ and
$\delta(D, CH_4)$ are expressed in ‰. Broadly summarized, $CH_4$ sources have a $\delta(^{13}C, CH_4)$ isotopic signature, hereinafter denoted
by $\delta_{source}(^{13}C, CH_4)$, between $-65$ and $-55$ ‰ for microbial sources, between $-45$ and $-35$ ‰ for thermogenic sources and

between $-25\,\text{\textperthousand}$ and $-15\,\text{\textperthousand}$ for pyrogenic sources (Sherwood et al., 2021, 2017). However, the full distributions of isotopic signatures are wider than these ranges, with overlaps between distributions of the different production processes. Similarly, $\delta_{\text{source}}(\text{D}, \text{CH}_4)$ distributions also depend on the production process. They roughly range between $-350\,\text{\textperthousand}$ and $-100\,\text{\textperthousand}$ for
thermogenic sources, between $-400\,\text{\textperthousand}$ and $-250\,\text{\textperthousand}$ for microbial sources and between $-250\,\text{\textperthousand}$ and $-175\,\text{\textperthousand}$ for pyrogenic sources (Sherwood et al., 2021, 2017). Notably, microbial and thermogenic $\delta_{\text{source}}(\text{D}, \text{CH}_4)$ distributions have smaller overlaps than $\delta_{\text{source}}(^{13}\text{C}, \text{CH}_4)$ and thermogenic sources have signatures less distinguishable from others.

Variations in atmospheric isotopic composition are not caused by sources only. Reactions between sink species (OH, O$^1$D and Cl) and $\text{CH}_4$ have rates that depend on the isotopologue. This effect is called fractionation and is represented, for a specific
reaction, using the ratio of the reactions rates with the lightest and the heaviest member of a couple of isotopologues (e.g., $^{12}\text{CH}_4$ and $^{13}\text{CH}_4$). The fractionation effect explains why the atmospheric isotopic composition is not equal to the flux-weighted mean source signature of all the $\text{CH}_4$ sources. It acts at shifting this mean source composition towards less negative values when $\text{CH}_4$ enters the atmosphere and gets removed by the sinks. This effect is particularly important for $\delta_{\text{source}}(\text{D}, \text{CH}_4)$ because the flux-weighted mean source signature and the observed isotopic composition are approximately $-330\,\text{\textperthousand}$ and $-95\,\text{\textperthousand}$, respectively
(Sherwood et al., 2017). For $\delta(^{13}\text{C}, \text{CH}_4)$, this effect is smaller, shifting the source signature from approximately $-53.6\,\text{\textperthousand}$ to $-47.3\,\text{\textperthousand}$ in the atmosphere (Sherwood et al., 2017).

The post-2007 $\text{CH}_4$ increase is notably associated with a decrease of 0.2-0.3 $\text{\textperthousand}$ in $\delta(^{13}\text{C}, \text{CH}_4)$ since 2007 (Rice et al., 2016; Schaefer et al., 2016; Nisbet et al., 2016, 2019). Such significant isotopic variations provide an additional atmospheric constraint to better estimate the relative contribution of $\text{CH}_4$ sources to this renewed atmospheric growth. Some of the afore-
mentioned studies that focused on the drivers of both the plateau and the renewed growth were conducted using inversion methods including isotopic constraints. These studies implemented either three-dimensional (3-D) CTMs coupled with analytical inversion methods estimating emissions for aggregated large regions only (Rice et al., 2016; McNorton et al., 2018), box models with analytical inversion methods (Schwietzke et al., 2016; Turner et al., 2017; Rigby et al., 2017) or 2-D CTMs with variational inversion methods (Thompson et al., 2018). Analytical methods are not fit for large-dimension problems, i.e.,
with both a large number of optimized variables and observations and these methods generally necessitate to aggregate emissions onto large regions. By contrast, variational inversion methods can easily both optimize the emissions at the grid-cell scale (model horizontal resolution) and assimilate large observational datasets. Furthermore, 3-D CTMs can better capture the spatial variability of sources, sinks and observations than box models and 2-D CTMs.

This paper utilizes the system designed by Thanwerdas et al. (2022a) to investigate changes in $\text{CH}_4$ emissions from 1998 to
2018 by running 3-D variational inversions at the grid-cell scale. The original system has been improved and can assimilate both $^{13}\text{CH}_4$:$^{12}\text{CH}_4$ and $\text{CH}_3\text{D}$:$\text{CH}_4$ observations. Optimization of source isotopic signatures is also tested here because, at present, they remain a large source of uncertainty (Sherwood et al., 2017; Turner et al., 2017; Feinberg et al., 2018) that should be considered.

In Sect. 2, we provide a detailed methodology describing the inversions performed in this study. In Sect. 3, the results are
presented. First, we evaluate the agreement between model outputs and assimilated data and also compare our simulations to independent data. As a second step, we provide an analysis of posterior emissions and isotopic signatures estimated by the

reference inversion and the sensitivity tests. To the best of our knowledge, the methodology developed by Basu et al. (2022) is the only one presenting high similarities to ours. They investigated the same problem with a variational inversion framework and a 3-D CTM. However, substantial differences exist between our techniques. In their paper, they included a comparison between their work and Thanwerdas et al. (2022a). Based on our new results, we propose an updated comparison in Sect. 3.9. A conclusion and a discussion are drawn in Sect. 4.

## 2 Methods

### 2.1 The chemistry-transport model

The general circulation model (GCM) LMDz is the atmospheric component of the coupled model of the Institut Pierre-Simon Laplace (IPSL-CM) developed at the Laboratoire de Météorologie Dynamique (LMD) (Hourdin et al., 2006). The version of LMDz used here is an "offline" version dedicated to the inversion framework created by Chevallier et al. (2005): the precalculated meteorological fields provided by the online version of LMDz are given as input to the model, considerably reducing the computation time. The model is built at a horizontal resolution of 3.8 ° × 1.9 ° (96 grid cells in longitude and latitude) with 39 hybrid sigma-pressure levels reaching an altitude of about 75 km. The time step of the model is 30 min and the output values have a resolution of 3 hours. Horizontal winds are nudged towards the ECMWF meteorological analyses (ERA-Interim) in the online version of the model. Vertical diffusion is parameterised by a local approach of Louis (1979), and deep convection processes are parameterised by the scheme of Tiedtke (1989). The offline model LMDz is coupled with the Simplified Atmospheric Chemistry System (SACS) to represent $CH_4$ oxidation by radicals (Pison et al., 2009; Thanwerdas et al., 2022a).

We simulate atmospheric $^{12}CH_4$ and $^{13}CH_4$ amount fractions to retrieve both $CH_4$ amount fractions and $\delta(^{13}C, CH_4)$ signal. Four clumped isotopologues ($^{12}CH_4$, $^{12}CH_3D$, $^{13}CH_4$ and $^{13}CH_3D$) are simulated in one sensitivity simulation (see Sect. 2.7) to retrieve both the $\delta(^{13}C, CH_4)$ and $\delta(D, CH_4)$ compositions.

Oxidations by OH, $O(^1D)$ and Cl are included in the chemical scheme of LMDz-SACS. Time-varying 3-D fields of OH and $O(^1D)$ with daily resolution, simulated beforehand with the LMDz-INCA chemistry model (Hauglustaine et al., 2004), are prescribed for each oxidant species to simulate the associated chemical loss. The same meteorological data has been used for generating these fields and running the simulations presented in this study.

The resulting OH field, named OH-INCA, exhibits a global mean tropospheric mass-weighted concentration of $11.1 \times 10^5$ $cm^{-3}$ over 1998-2018, consistent with the previous estimates from Zhao et al. (2019) ($11.7 \times 10^5$ $cm^{-3}$), estimates from Prather et al. (2012) (($11.2 \pm 1.3) \times 10^5$ $cm^{-3}$), and well within the range derived from the Atmospheric Chemistry and climate Model Intercomparison Project (ACCMIP) (10.3-13.4 $\times 10^5$ $cm^{-3}$ ; Voulgarakis et al., 2013). It is however slightly larger than the very recent estimate from Zhao et al. (2023) obtained by constraining OH with observations of its precursors. The inter-hemispheric ratio is 1.14, lower than the mean value of 1.3 inferred by Zhao et al. (2019), although more consistent with recent estimates from Zhao et al. (2023) and an inter-hemispheric parity obtained from methyl-chloroform-based inversions (Bousquet et al., 2005; Patra et al., 2014). Global concentrations of OH-INCA increase by 4 % between 2002 and 2014.

As suggested by Thanwerdas et al. (2022b), the Cl concentrations derived by Wang et al. (2021) are prescribed here for all simulations. Their work suggests the Cl sink accounts for only 0.8 % of the total $CH_4$ oxidation, lower than other estimates used in the literature (1.8-5% ; Allan et al., 2007; Sherwen et al., 2016; Hossaini et al., 2016).

The fractionation effect must also be represented in the modelling framework. Table 1 provides the fractionation coefficients applied for each loss reaction. For the OH sink, we adopted the estimate derived by Saueressig et al. (2001). Burkholder (2020) recommends using the Saueressig et al. (2001) rates but suggests increasing the uncertainty in the OH fractionation to account for Cantrell et al. (1990) estimate (1.0054). As shown by Basu et al. (2022), switching from Saueressig et al. (2001) to Cantrell et al. (1990) estimates has a large influence on the results, despite the authors do not optimize source signatures in their setup. As Saueressig et al. (2001) indicate their data is of considerably higher experimental precision and reproducibility than previous studies, in particular Cantrell et al. (1990), we prefer to allocate computational time to a sensitivity inversion testing a different OH field rather than testing a different OH fractionation coefficient. In addition, these estimates of fractionation coefficients come with uncertainty ranges that we could also consider in our inversions (e.g. with a Monte-Carlo approach). In our case, the main limitation remains the large computational cost of one inversion (see Sect. 2.9). In the future, we hope to be able to increase the number of sensitivity tests and account for this uncertainty. For this work, the values we adopt are the best estimates for each fractionation coefficient.

**Table 1.** Fractionation coefficients for loss reactions with OH, $O(^1D)$, Cl and soil uptake. $T$ denotes the temperature.

| Species | $k(^{12}CH_4) / k(^{13}CH_4)$ | Réference | $k(CH_4) / k(CH_3D)$ | Reference |
|---|---|---|---|---|
| OH | 1.0039 | Saueressig et al. (2001) | $1.097 \times e^{-49K/T}$ | Saueressig et al. (2001) |
| $O(^1D)$ | 1.013 | Saueressig et al. (2001) | 1.06 | Saueressig et al. (2001) |
| Cl | $1.043 \times e^{6.455K/T}$ | Saueressig et al. (1995) | $1.278 \times e^{-53.31K/T}$ | Saueressig et al. (1996) |
| Soil uptake | 1.020 | Snover and Quay (2000) Reeburgh et al. (1997) Tyler et al. (1994) King et al. (1989) | 1.083 | Snover and Quay (2000) |

## 2.2 Inverse modelling with a variational approach

Inversions were performed using the Community Inversion Framework (CIF; Berchet et al., 2021). This framework was designed to rationalize and bridge development efforts made by the scientific community within the same flexible, transparent and open-source system. This system was recently enhanced by Thanwerdas et al. (2022a) to assimilate $\delta(^{13}C, CH_4)$ together with $CH_4$ observations and optimize both $CH_4$ emissions and source signatures $\delta_{source}(^{13}C, CH_4)$ at the same time. For the purpose of this study, $\delta(^{13}C, CH_4)$ and $\delta(D, CH_4)$ observations are assimilated together in the same inversion and the system optimizes both source signatures $\delta_{source}(^{13}C, CH_4)$ and $\delta_{source}(D, CH_4)$.

The notations introduced here to describe the variational inversion method follow the convention defined by Ide et al. (1997) and Rayner et al. (2019). $\mathbf{x}$ is the control vector and includes all the variables optimized by the inversion system. Prior information about the control variables is included in the vector $\mathbf{x}^b$. Its associated errors are assumed to be unbiased and Gaussian, and are described within the error covariance matrix $\mathbf{B}$.

The observation vector $\mathbf{y}^o$ includes here all available observations, namely atmospheric $CH_4$ amount fraction, $\delta(^{13}C, CH_4)$, and also $\delta(D, CH_4)$ data for one sensitivity test. The associated errors are also assumed to be unbiased and Gaussian, and are described within the error covariance matrix $\mathbf{R}$. This matrix accounts for all errors contributing to mismatches between simulated and observed values.

$\mathscr{H}$ is the observation operator that projects the control vector $\mathbf{x}$ into the observation space. This operator mainly consists of the CTM but is also followed by spatial, time and isotope-conversion operators. Following Thanwerdas et al. (2022a), prescribed source signatures and $CH_4$ fluxes are first combined to generate isotope fluxes. These fluxes are then fed to the model to simulate the mixing ratios of the different isotopes over the time period considered. After the forward run, the simulated fields are interpolated to produce simulated equivalents of the observed amount fractions and isotopic compositions at specific locations and times, ensuring that a comparison between simulations and observations is possible. Adjoint versions of these forward operations are also implemented in order to perform the complementary adjoint run.

In a variational formulation of the inversion problem that allows $\mathscr{H}$ to be nonlinear, the cost function $J$ is defined as:

$$J(\mathbf{x}) = \frac{1}{2}(\mathbf{x} - \mathbf{x}^b)^T \mathbf{B}^{-1}(\mathbf{x} - \mathbf{x}^b) + \frac{1}{2}(\mathscr{H}(\mathbf{x}) - \mathbf{y}^o)^T \mathbf{R}^{-1}(\mathscr{H}(\mathbf{x}) - \mathbf{y}^o) \tag{3}$$

Here, the minimum of $J$ is reached iteratively with the descent algorithm M1QN3 (Gilbert and Lemaréchal, 1989) that requires several computations (40-50) of the gradient of $J$ with respect to the control vector $\mathbf{x}$:

$$\nabla J_{\mathbf{x}} = \mathbf{B}^{-1}(\mathbf{x} - \mathbf{x}^b) + \mathscr{H}^*(\mathbf{R}^{-1}(\mathscr{H}(\mathbf{x}) - \mathbf{y}^o)) \tag{4}$$

$\mathscr{H}^*$ denotes the adjoint operator of $\mathscr{H}$.

The reference inversion (INV_REF) assimilates $CH_4$ and $\delta(^{13}C, CH_4)$ observations over 1998-2018. $CH_4$ emissions and $\delta(^{13}C, CH_4)$ source signatures for five categories of emissions are optimized: biofuels-biomass burning (BB), wetlands (WET), fossil fuels and geological sources (FFG), agriculture and waste (AGW) and other natural sources (NAT). $CH_4$ and $\delta(^{13}C, CH_4)$ initial conditions are also optimized (see Sect. 2.6).

## 2.3 Prior emissions and uncertainties

For prior $CH_4$ emissions, we adopt the bottom-up estimates compiled for the inversions performed as part of the Global Methane Budget and described in detail in Saunois et al. (2020). In short, anthropogenic (including biofuels) and fire emissions are based on EDGARv4.3.2 (Janssens-Maenhout et al., 2019) and GFED4s (van der Werf et al., 2017), respectively. Statistics from British Petroleum (BP) and the Food and Agriculture Organization of the United Nations (FAO) have been used to extend EDGARv4.3.2, ending 2012, until 2017. The natural sources emissions are based on averaged literature values : Poulter et al. (2017) for wetlands, Kirschke et al. (2013) for termites, Lambert and Schmidt (1993), Etiope (2015) for geological (onshore)

sources and oceanic sources that include geological (offshore) and hydrates sources. Prior emissions for 2018 are set equal to 2017. Globally averaged emissions over 1998-2018 are listed in Table 2.

BB emissions are the combination of biomass burning emissions from GFED4s and biofuels burning emissions from EDGARv4.3.2. FFG emissions are the combination of oil, gas, coal, industry and transport emissions from EDGARv4.3.2 and geological (onshore) sources from Etiope (2015) whose global emissions were scaled down to 15.0 Tg a$^{-1}$ in the protocol of Saunois et al. (2020). AGW emissions are the combination of enteric fermentation, rice agriculture, manure management and waste emissions from EDGARv4.3.2. NAT emissions are the combination of termites and oceanic emissions, i.e., emissions
from all natural sources apart from wetlands and geological sources.

**Table 2.** Information about emissions and flux-weighted isotopic signatures for the different categories. Emissions and source signatures are averaged over 1998-2018. The uncertainty (unc.) indicates the prior uncertainty as a percentage of the square of the maximum of prior emissions over the cell and its eight neighbors during each month (or over a continental region for the signatures). This uncertainty is used to fill the matrix **B**. The number of optimized scaling factors (optim.) can either be 1) 3PMPG: three scaling factors per month and per gridcell, 2) PYR: one scaling factor per year and per continental region or 3) PR: one scaling factor per continental region for the full assimilation window.

| Categories | Emissions [Tg a$^{-1}$] | Unc. | Optim. | $\delta_{source}(^{13}C, CH_4)$ [‰ vs PDB] | Unc. | Optim. | $\delta_{source}(D, CH_4)$ [‰ vs VSMOW] | Unc. | Optim. |
|---|---|---|---|---|---|---|---|---|---|
| WET | 180 [180 / 180] | 100% | 3PMPG | −60.8 | 10% | PR | −320.8 | 40% | PR |
| AGW | 213 [195 / 232] | 100% | 3PMPG | −59.1 | 10% | PYR | -310.0 | 30% | PR |
| FFG | 117 [99 / 133] | 100% | 3PMPG | −44.9 | 20% | PYR | -183.0 | 20% | PR |
| BB | 27 [24 / 35] | 100% | 3PMPG | −22.3 | 30% | PR | -200.0 | 35% | PR |
| NAT | 23 [23 / 23] | 100% | 3PMPG | −50.7 | 15% | PR | -230.0 | 35% | PR |

Emissions are optimized at the grid-cell scale (one scaling factor per grid cell). For each category, diagonal elements of the matrix **B** are filled with the variances set to 100 % of the square of the maximum of prior emissions over the cell and its eight neighbors during each month. Spatial error correlations (off-diagonal elements) are prescribed using an e-folding correlation length of 500 km on land and 1000 km over the oceans, without any correlation between land and ocean grid points. No
temporal error correlations are prescribed.

## 2.4 Prior source signatures and uncertainties

$\delta_{source}(^{13}C, CH_4)$ values for each emissions category are also optimized and therefore included in the control vector. Prior information is built using the references given in Table 3. When regional information could be found, regional source signature values were prescribed onto 11 continental regions (see Fig. 1, lower-right panel) for each subcategory. As isotopic signatures
for subcategories are flux-weighted averaged to create signatures for categories, it results in signatures that are grid-cell dependent, although signatures for subcategories are set constant within a continental region. Signatures for wetlands are the only ones prescribed at the grid-cell scale, following Ganesan et al. (2018). We optimize the source signatures at the regional scale rather than at the grid-cell scale to avoid substantial posterior differences between two adjacent grid cells. At present, there

would not be enough data to corroborate, explain or reject such differences. We therefore apply only one scaling factor per
235 continental region and per category.

Livestock source signatures have been likely decreasing over time since the 1990s due to changes in C3/C4 diet within
the major livestock producing countries (Chang et al., 2019). Also, FFG regional source signatures can vary over time due to
variations in the contributions from different sectors (coal, oil and gas) to the emissions of a specific region (Schwietzke et al.,
2016; Feinberg et al., 2018). For AGW and FFG source signatures, we therefore optimize one scaling factor per year, for each
240 continental region. As for the other emission categories, only one scaling factor for the entire period and for each continental
region is optimized. Error covariances are prescribed following the same methods as those applied to $CH_4$ emissions.

As for $\delta_{\text{source}}(D, CH_4)$, we adopted global values suggested by Warwick et al. (2016) and in agreement with the intervals
given by Röckmann et al. (2016). One exception is for WET sources for which the boreal ($-360\,\%o$) and tropical ($-320\,\%o$)
regions are differentiated. All values are summarized in Table 2. For each category and each continental region, only one
scaling factor is optimized for the entire period.

**Table 3.** Global flux-weighted values and references for $\delta_{\text{source}}(^{13}C, CH_4)$ source signatures associated to the different emission categories
and subcategories. Values for subcategories are taken from literature and either prescribed globally ($.^G$), regionally ($.^R$, see Fig. 1, lower-right
panel) or at the pixel scale ($.^P$).

E19: (Etiope et al., 2019) ; CH19: Chang et al. (2019) ; GA18: Ganesan et al. (2018) ; TH18: Thompson et al. (2018) ; SH17: Sherwood et al.
(2017) ; SH16: Schwietzke et al. (2016) ; WA16: Warwick et al. (2016) ; ZA16: Zazzeri et al. (2016) ; TO12: Townsend-Small et al. (2012) ;
KL10: Klevenhusen et al. (2010) ; BO06: Bousquet et al. (2006) ; BR01: Bréas et al. (2001) ; SA01: Sansone et al. (2001) ; CH00: Chanton
et al. (2000) ; HO00: Holmes et al. (2000) ; CH99: Chanton et al. (1999) ; BE98: Bergamaschi et al. (1998) ; LE93: Levin et al. (1993);

| Categories | Global signature (‰) | Subcategories | Global signature (‰) | References |
|---|---|---|---|---|
| AGW | -59.1 | Rice cultivation | -63.0 [G] | SH17; BO06; BR01 |
| | | Enteric fermentation | -64.7 [P] | CH19 |
| | | Agriculture waste | -52.0 [G] | KL10 ; LE93 |
| | | Landfills | -52.0 [G] | TO12 ; CH99 ; BE98 ; LE93 |
| | | Waste water | -48.0 [G] | TO12 ; CH99 ; BE98 ; LE93 |
| FFG | -44.9 | Oil and gas | -44.9 [R] | SH07 |
| | | Coal | -42.3 [R] | SH07 ; ZA16 |
| | | Geological sources | -49 [G] | E19 |
| BB | -22.3 | Biomass burning | -24.9 [R] | BO06 ; CH00 |
| | | Biofuel burning | -20 [G] | CH00 |
| WET | -60.8 | Wetlands | -60.8 [P] | GA18 |
| NAT | -50.7 | Oceanic sources | -42 [G] | BR01; HO00 ; SA01 |
| | | Termites | -63 [G] | TH18; SH16 ; SH17; WA16 |

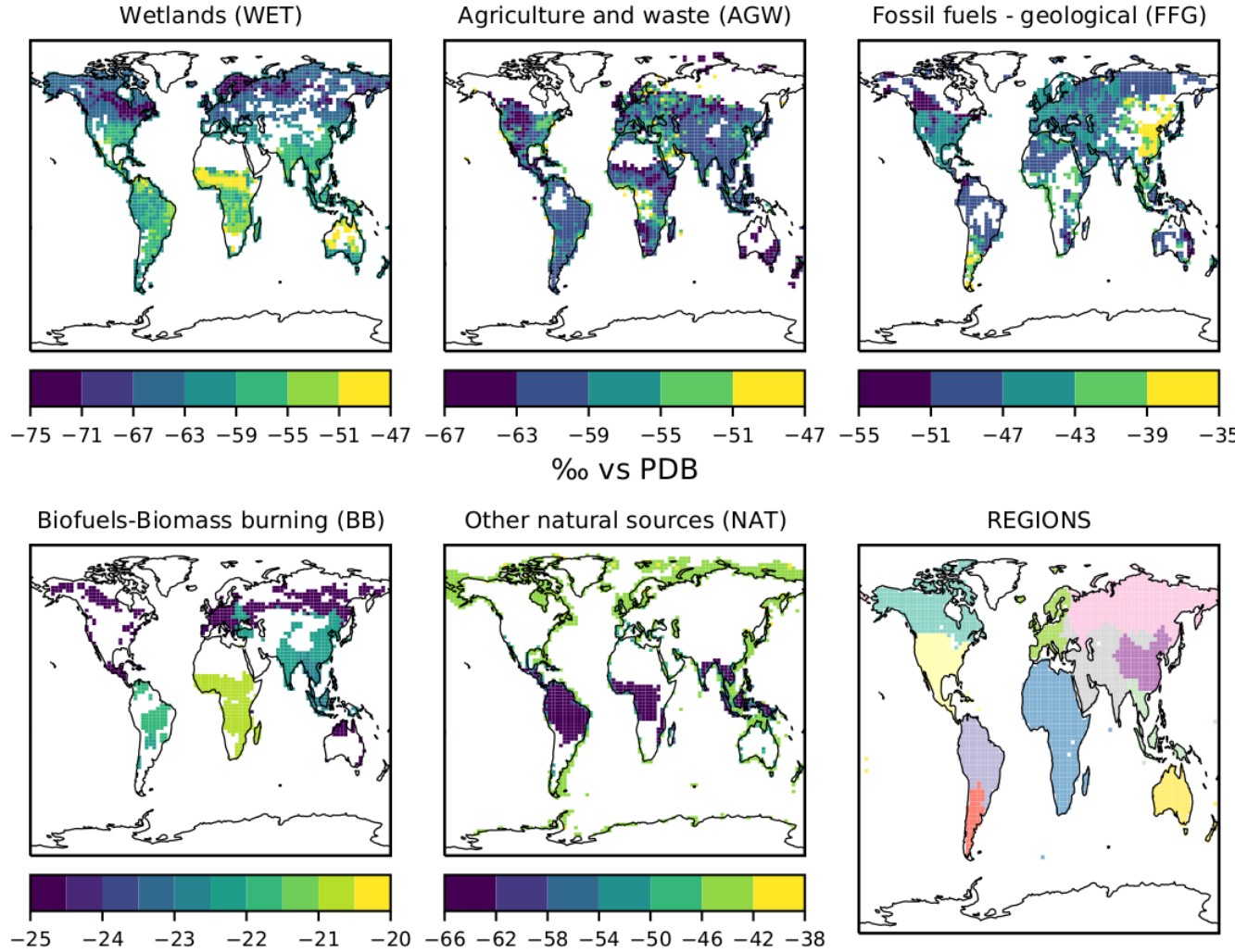

**Figure 1.** Prior estimates of $\delta_{source}(^{13}C, CH_4)$ isotopic signatures for each of the five emission categories averaged over the 1998-2018 period. The regions over which the values are optimized are shown in the lower-right panel. WET source signatures are dependent on the latitude, with more depleted values in boreal regions than in tropical regions. BB source signatures are dependent on the vegetation (C3/C4). Burning C4 vegetation tropical regions releases $CH_4$ that is more $^{13}C$-enriched than $CH_4$ released when burning C3 vegetation. AGW source signatures is dependent on the country/region and the C3 versus C4 livestock diet. FFG source signatures mainly depend both on the location and the contributions from coal, oil&gas and geological sources to the total FFG emissions of a specific country/region. For example, China $^{13}C$-enriched large coal emissions highly contributes to the FFG source signature in this region which is notably $^{13}C$-enriched compared to other regions.

$\delta_{source}(^{13}C, CH_4)$ uncertainty values that are used to fill the diagonal elements of the matrix **B** are summarized in Table 2. These values have been chosen by compiling data from several studies (Sherwood et al., 2017; Ganesan et al., 2018; Feinberg

et al., 2018; Zazzeri et al., 2016). The observed variability over the whole globe (standard deviation $\sigma$ or minimum-maximum range) presented in these studies for each category is compiled and applied here as an uncertainty (1-$\sigma$), therefore adopting the same value for all regions. Note that for BB sources, Sherwood et al. (2017) indicates a global standard deviation of about 20 %. However, this value is not weighted by the proportion of C3 versus C4 vegetation. Therefore, we inflated this uncertainty up to 30 % to account for the uncertainty in the type of vegetation. $\delta_{\text{source}}(D, CH_4)$ uncertainty values have been derived from the minimum-maximum ranges suggested by Röckmann et al. (2016). We could have also used the standard deviation provided by Sherwood et al. (2017). However, as the amount of data for AGW, BB, WET and NAT source signatures is very low compared to $\delta_{\text{source}}(^{13}C, CH_4)$ values, we prefer to use larger uncertainties and examine whether the assimilation of $\delta(D, CH_4)$ observations modifies the results. For future studies, additional $\delta(D, CH_4)$ data would be invaluable to derive realistic regional estimates, especially for non-fossil sources.

We acknowledge the fact that our methods are not perfect and that the prescribed regional uncertainties might be too large compared to the regional observed uncertainties that are currently estimated, especially for $\delta_{\text{source}}(^{13}C, CH_4)$. As our inversion system is used for the first time over a time period exceeding ten years, it is difficult to predict the influence of setup on results. As the time to run an inversion is very high at the moment (see Sect. 2.9), we prefer to assess the behavior of this system in response to a simple (and probably slightly loose) set-up and estimate whether such uncertainties are small enough to help better constrain the $CH_4$ budget. Additionally, source signature data representativeness is generally poor owing in part to a small number of samples and a lack of data for several regions, particularly for non-fossil sources. It might therefore be challenging to derive a robust uncertainty using a data-driven approach for each region of the world. However, there is definitely room for improvement and future work will build on the present work to improve this methodology and assess and prescribe better uncertainties.

Note that random uncertainties are only one side of the coin of uncertainties. Systematic uncertainties must also be investigated when using isotopic constraints. In particular, Oh et al. (2022) derived source signature maps for wetlands sources that carry systematic uncertainties. Typically, inverse modelers address such uncertainties by conducting numerous inversions using parameters designed to account for these systematic errors. The high computational cost associated to our system prevents us from running a large number of inversions. Nevertheless, only one scaling factor is applied for each region and each category. Consequently, there is a strong regional correlation between random errors. To some degree, this approach enables the detection and correction of regional systematic errors by our system. It is important to note, however, that this correction does not rely on any existing sensitivity analysis (e.g. Lan et al., 2021; Basu et al., 2022) but rather uses solely the information provided by atmospheric isotopic observations.

## 2.5 Observations

### 2.5.1 Assimilated data

Our study uses observations from the NOAA Global Monitoring Laboratory (NOAA GML) Global Greenhouse Gas Reference Network. Methane amount fractions are made by NOAA GML (Lan et al., 2022), and isotopic measurements are made at the

Stable Isotope Laboratory at the Institute of Arctic and Alpine Research (INSTAAR) (White et al., 2021, 2016). This ensemble was selected to provide the largest number of consistent $CH_4$ and isotopic data since 1998. 79 stations (among which 4 mobile stations) provided $CH_4$ measurements between 1998 and 2018 (not necessarily over the full period), 22 stations provided $\delta(^{13}C, CH_4)$ measurements between 1998 and 2018 and 15 stations provided $\delta(D, CH_4)$ between 2005 and 2010 (see Fig. 2).

Missing $CH_4$ instrumental errors are filled with the maximum value of this error at the station over the monitoring period. For $\delta(^{13}C, CH_4)$ and $\delta(D, CH_4)$ measurements, missing instrumental errors are filled with a value of 0.1 ‰ and 3 ‰, respectively (Quay et al., 1999). Variances (diagonal elements) in the covariance matrix **R** are defined as the sum of the instrumental and model errors (variances). For each station and each year, we used the Residual Standard Deviation (RSD) between the measurements and a fitting curve function as a proxy for the model error (Thanwerdas et al., 2022a; Locatelli et al., 2015;

Bousquet et al., 2006). The fitting function includes 3 polynomial parameters (quadratic) and 8 harmonic parameters, sinus and cosinus, as in Masarie and Tans (1995). We also remove outliers outside three times the residual standard deviations as such extreme values cannot be reasonably reproduced at the horizontal grid resolution of LMDz. Typical values for observation errors are $20 \ nmol \ mol^{-1}$ for $CH_4$, 0.3 ‰ for $\delta(^{13}C, CH_4)$ and 7 ‰ for $\delta(D, CH_4)$.

### 2.5.2    Satellite data used for comparison

The Greenhouse Gases Observing Satellite (GOSAT) carrying a Fourier Transport Spectrometer within the Thermal And Near-infrared Sensor for carbon Observation (TANSO-FTS) (Kuze et al., 2016) provides radiance measurements in a spectral band centered on a value close to 1.6 $\mu$m, in which $CH_4$ has a high absorption capacity. The University of Leicester's retrieval algorithm is able to produce column-average dry air amount fractions of $CH_4$ from these radiances. Although this quantity is commonly referred to with the symbol $XCH_4$ in the existing literature, it is denoted here by $\overline{X}(CH_4)$ because it is considered

a more valid notation. We use the version 9.0 of the GOSAT Proxy $\overline{X}(CH_4)$ dataset provided by the University of Leicester (Parker et al., 2020), for evaluation of the $\overline{X}(CH_4)$ after the inversion process. To this end, vertical profiles simulated by LMDz-SACS are sampled at the observation location and time and convolved with the retrieval of the prior vertical profiles and column averaging kernels provided by the University of Leicester. Finally, within each grid cell, all the individual $\overline{X}(CH_4)$ differences between satellite observations and model outputs are averaged.

Satellite data is not assimilated here because, at present, inversions assimilating both satellite and surface data have not been performed with LMDz-SACS. Before using satellite data and isotope data together in an inversion, we need to rigorously assess the added value of satellite data without isotope constraints.

### 2.6    Initial conditions

To infer initial conditions on $CH_4$ and $\delta(^{13}C, CH_4)$ for 1998, we run an inversion between 1988 and 1998 using the same

prior emissions and isotopic signatures as that of INV_REF. We assimilate $CH_4$ measurements from the NOAA GML network (56 stations) and $\delta(^{13}C, CH_4)$ measurements retrieved at 5 stations across the globe by the University of Washington (UW) between 1988 and 1996 (Quay et al., 1999; Bousquet et al., 2006) that we offset by 0.1 ‰ to account for measurement differences between INSTAAR and UW (Umezawa et al., 2018).

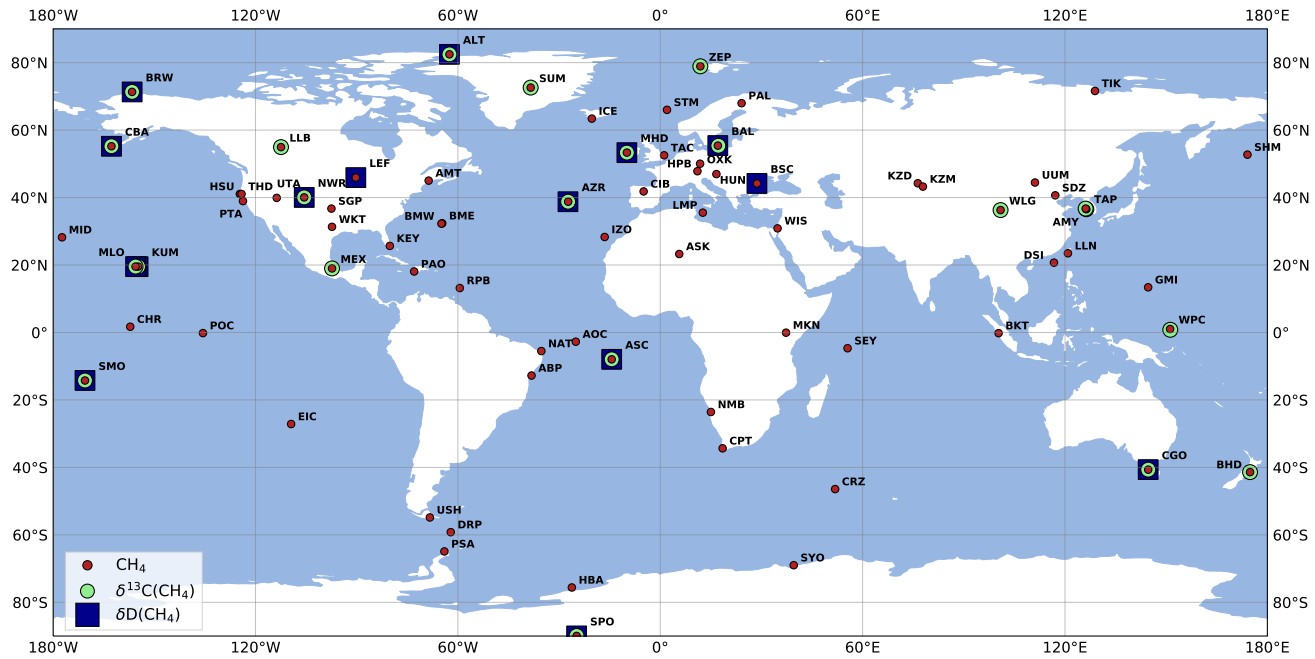

**Figure 2.** Locations of $CH_4$, $\delta(^{13}C, CH_4)$ and $\delta(D, CH_4)$ surface stations from the NOAA GML network. Samples from several stations are retrieved and analyzed by INSTAAR to provide $\delta(^{13}C, CH_4)$ and $\delta(D, CH_4)$ observations. More information about the stations can be found in Appendix A. Note that mobile stations (AOC, PAO, POC and WPC) are indicated by a single point for clarity.

We also run a forward simulation from 1998 to 2010 to obtain a good spatial distribution of the $\delta(D, CH_4)$ field and then apply a global offset to match the observed mean $\delta(D, CH_4)$ value between 2005 and 2010. As we acknowledge that both methods are not perfect considering the equilibration times of these isotopic compositions (Tans, 1997), we also prescribe large uncertainties in these initial conditions: 10 % for $CH_4$, 3 % for $\delta(^{13}C, CH_4)$ and 20 % for $\delta(D, CH_4)$. To optimize the initial conditions, the globe is regularly discretized using latitudinal and longitudinal bands. A step of 30 °degrees is applied to generate the bands, resulting in $6 \times 12 = 72$ regions. One scaling factor is optimized for each of these regions.

### 2.7 Description of the sensitivity tests

The reference inversion was first introduced in Sect. 2.2 and its setup is detailed in the previous sections. Three sensitivity tests were conducted to investigate the influence of the setup of our system on posterior estimates when assimilating isotopic observations:

– INV_CH4 is an inversion that assimilates only $CH_4$ observations, does not assimilate $\delta(^{13}C, CH_4)$ nor $\delta(D, CH_4)$ observations and thus only optimizes $CH_4$ emissions for the five categories.

- INV_DD assimilates $CH_4$, $\delta(^{13}C, CH_4)$ and $\delta(D, CH_4)$ observations and optimizes both $\delta_{source}(^{13}C, CH_4)$ and $\delta_{source}(D, CH_4)$ source signatures. Note that $\delta(D, CH_4)$ observations spans only the period from 2005 to 2010 and therefore, the full run cannot be fully constrained by this data.

- INV_LOCKED is an inversion that assimilates $CH_4$ and $\delta(^{13}C, CH_4)$ observations but does not optimize $\delta(^{13}C, CH_4)$ source signatures (fixed to prior values). This run considers source signatures fixed to prior values and thus investigates the influence of over-constrained isotopic signatures on posterior estimates.

We also investigate the influence of the OH inter-annual variability (IAV) on our results. The OH IAV in the troposphere is usually derived from inversions using $CH_3CCl_3$ constraints (Montzka et al., 2011; Rigby et al., 2017; Turner et al., 2017; Naus et al., 2019) or using global atmospheric chemistry–climate models (He et al., 2020; Dalsøren et al., 2016). $CH_3CCl_3$ inversion-based studies suggest a decrease in post-2005 OH after a peak in 2000-2002. By contrast, the chemistry modelling studies derive a post-2005 stabilization after a quasi-continuous increase between 1990 and 2005, consistent with the OH IAV estimated by LMDz-INCA (see Fig. 3). We therefore perform two more sensitivity tests:

- INV_TURNER is designed to investigate the influence of the IAV on our results. We apply the IAV suggested by Turner et al. (2017) to the OH-INCA field. The associated OH field is named OH-TURNER and its global concentrations decrease by 7 % between 2002 and 2014.

- INV_FLATOH removes the IAV from our OH-INCA field by prescribing the concentrations of the year 2000 over the full period. The associated field is named OH-FLAT.

All the sensitivity tests are summarized in Table 4.

**Table 4.** Description of the sensitivity tests.

| Name | Simulated tracers | Source signatures optimization | OH field |
|---|---|---|---|
| INV_REF | $^{12}CH_4$ $^{13}CH_4$ | $\delta_{source}(^{13}C, CH_4)$ | OH-INCA |
| INV_CH4 | $CH_4$ | None | OH-INCA |
| INV_LOCKED | $^{12}CH_4$ $^{13}CH_4$ | $\delta_{source}(^{13}C, CH_4)$ | OH-INCA |
| INV_DD | $^{12}CH_4$ $^{12}CH_3D$ $^{13}CH_4$ $^{13}CH_3D$ | $\delta_{source}(^{13}C, CH_4)$ $\delta_{source}(D, CH_4)$ | OH-INCA |
| INV_FLATOH | $^{12}CH_4$ $^{13}CH_4$ | $\delta_{source}(^{13}C, CH_4)$ | OH-FLAT |
| INV_TURNER | $^{12}CH_4$ $^{13}CH_4$ | $\delta_{source}(^{13}C, CH_4)$ | OH-TURNER |

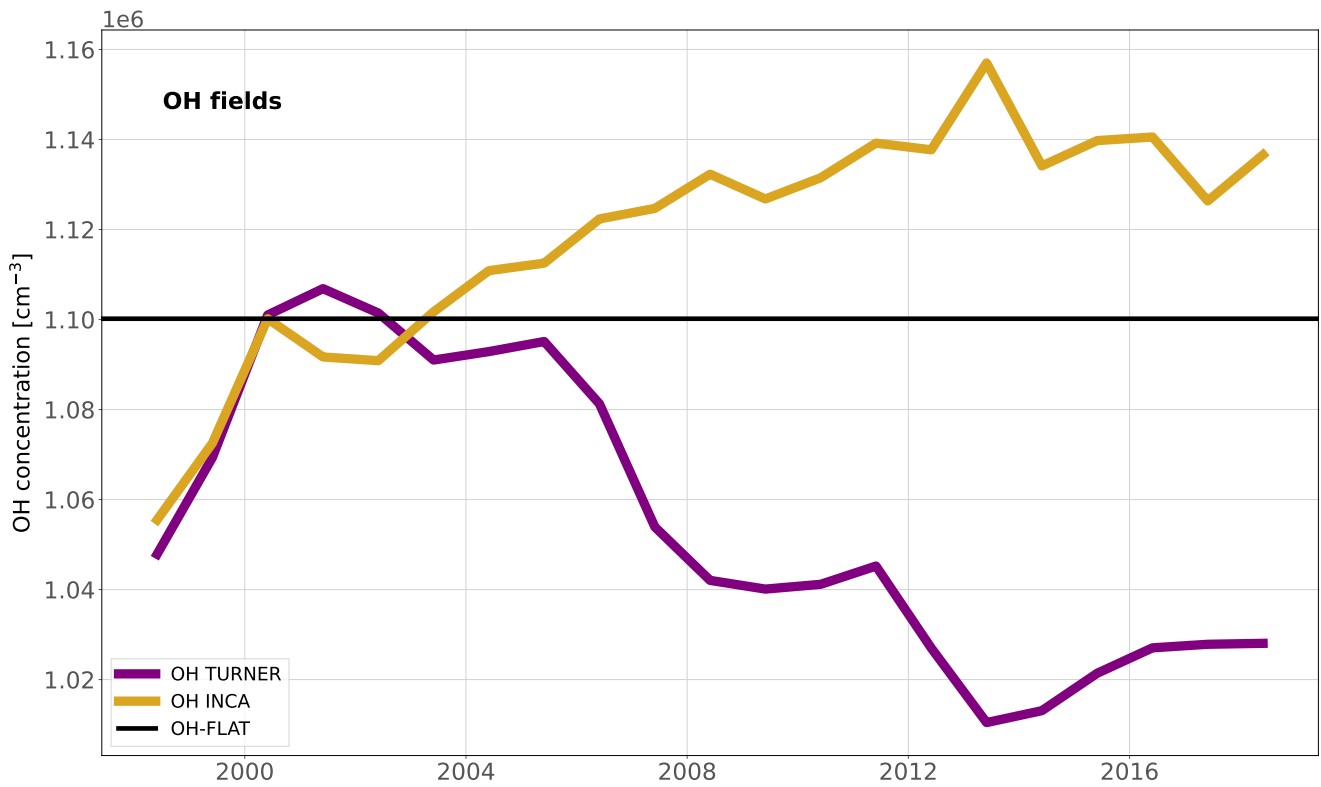

**Figure 3.** Time-series of the global volume-weighted tropospheric OH annual concentrations for the 1998-2018 period. OH-INCA has been simulated by the LMDz-INCA chemistry model. OH-TURNER has been obtained by applying the IAV from Turner et al. (2017) to the OH-INCA field. OH-FLAT has no inter-annual variability and concentrations are set equal to those of OH-INCA in 2000. In 1980, the OH-INCA mean concentration is very close to $10.0 \times 10^5$ cm$^{-3}$, also taken as a reference value by Turner et al. (2017). This year is therefore taken as a reference to derive the anomalies.

## 2.8 Analysis period

Thanwerdas et al. (2022a) suggested that the results of the inversion should be discarded up to 2-3 years after the beginning and 2-3 years before the end of the assimilation window. For the beginning of the window, although the term "spin-up" is not quite appropriate here because the cause of the discard is slightly different, the outcome is still similar. A spin-up time, for an inversion, typically refers to a period at the beginning of the inversion where the errors in the assumed initial concentrations field might influence the posterior fluxes. If the spin-up is taken too short, the inversion may fit the data by compensating

errors in the initial condition with artificial emission adjustments. If the initial concentrations are also optimized, which is the case here, this effect can be reduced (Houweling et al., 2017). However, source signatures are optimized and for a certain period after the start of the inversion, it is easier for the system to optimize the initial $\delta(^{13}\text{C}, \text{CH}_4)$ fields rather than the source signatures to fit the $\delta(^{13}\text{C}, \text{CH}_4)$ data. Thanwerdas et al. (2022a) found that the optimized source signatures slowly move away

from the prior value over time. After 2-3 years, the posterior value finally reaches a new and rather stable state. In other words, as the influence of initial conditions on the isotopic composition decrease, the system prefers to optimize the source signatures, hence slowly reaching the posterior value. As the equilibration time for $\delta(^{13}\text{C},\text{CH}_4)$ and $\delta(\text{D},\text{CH}_4)$ is larger than the $\text{CH}_4$ equilibration time (Tans, 1997), this affects more the source signatures than the fluxes. For the end of the inversion, it is mainly caused by a lack of constraints, therefore using the term "spin-down" is correct.

In addition, the strong 1997-1998 El Niño event leads to fire emission anomalies of about $20\,\text{Tg}\,\text{a}^{-1}$ according to GFEDv4s data and studies (Bousquet et al., 2006; Langenfelds et al., 2002). Similarly, the following 1999-2000 La Niña event produced a wetlands emission anomaly that persisted until 2002 (Zhang et al., 2018). Finally, OH concentrations were also likely affected by this El Niño - Southern Oscillation (ENSO) phase (Zhao et al., 2020a). We therefore only analyze the results of the 2002-2014 period to limit the consequences of these effects. This period of time is large enough to explain the variations causing the post-2007 $\text{CH}_4$ renewed growth and the associated $\delta(^{13}\text{C},\text{CH}_4)$ shift to more negative values.

## 2.9 Computational aspects

The same convergence criterion was used for all inversions in order to ensure consistency between the results. The minimization process was stopped when at least 35 iterations (forward + adjoint runs) have been performed and the gradient norm ratio has fallen below 1 % of its initial value for four successive iterations.

A similar number of iterations (approx. 40) were necessary for all sensitivity tests. About 260 CPU hours were necessary to run a single iteration on LSCE computational clusters consisting of Intel® Xeon® Gold 5317 central processing units (CPUs) with a frequency of 3.00 GHz. For this work, 8 CPUs were run in parallel, resulting in a runtime of 32.5 hours for a single iteration. More CPUs cannot increase the overall performance because of some I/O (input/output) limitations of our offline model. With only one tracer to simulate, INV_CH4 therefore necessitated about 2 months to reach the convergence criterion. Because the runtime is proportional to the number of simulated tracers, it necessitated twice this runtime for the other inversions (two tracers), except for INV_DD which necessitated four times this runtime (four tracers).

While the number of CPU hours needed for these complex inversions remains reasonable, the overall runtime is excessive. It is therefore an important limitation of our system. Further developments on parallelization methods are being implemented to enable a significant reduction of the computational cost (e.g., Chevallier, 2013). This method consists of breaking down the full assimilation window into multiple sub-windows, and running smaller inversions in parallel for each sub-window. If source signatures remain constant, we expect the results to closely resemble those of a longer-term window inversion. Conversely, if source signatures are optimized, the influence of initial conditions on the atmospheric isotopic composition might persist over a time that is larger than the length of the sub-window. In this case, source signatures might remain unchanged and the results could be impacted. Therefore, it is crucial to rigorously validate this parallelization method before interpreting its outcomes.

## 3 Results

In this section, we first verify the quality of the model's fit to the constraining observations and evaluate it against independent data (Sect. 3.1 and 3.2). After this, we examine the posterior estimates of our reference inversion for emissions and source signatures and compare it to prior estimates (Sect. 3.3 and 3.4). Subsequently, we attribute the post-2007 $CH_4$ increase and $\delta(^{13}C, CH_4)$ downward shift to a changes in $CH_4$ emissions and source signatures (Sect. 3.5 and 3.6). Finally, we analyze the sensitivity of our results to setup modifications (Sect. 3.7 and 3.8).

### 3.1 Model-observation agreement

Before analyzing the optimized emissions and source signatures, we verify the quality of the model's fit to the constraining observations and evaluate it against independent data. A good fitting show that the system is operational over long time periods and that posterior emissions and source signatures are consistent with the observed state of the atmosphere.

The observed globally-averaged $CH_4$ amount fraction as well as the observed globally-averaged $\delta(^{13}C, CH_4)$ isotopic composition at the surface are well captured by all the posterior simulations (see Fig. 4, right panels). INV_REF shows a Pearson's moment correlation coefficient $r$ of 0.994 for $CH_4$ (RMSE is 2.8 $nmol\,mol^{-1}$) and 0.936 for $\delta(^{13}C, CH_4)$ (RMSE is 0.04 ‰). The posterior simulation therefore captures much better the observations than the prior simulation (RMSEs of 71.2 $nmol\,mol^{-1}$ and 1.44 ‰). The inversion that best captures the $\delta(^{13}C, CH_4)$ isotopic composition is INV_DD with a RMSE of 0.02 ‰. It shows that assimilating $\delta(D, CH_4)$ observations slightly increases the agreement with isotopic observations without additional iterations.

The 2002-2007 $\delta(^{13}C, CH_4)$ stabilization is well reproduced by the model in INV_REF, showing a mean RMSE of 0.02 ‰ over the period. However, the post-2007 trend is not as consistent (0.05 ‰), mainly due to an overestimation of the decreasing rate (0.03 ‰ $a^{-1}$ against 0.02 ‰ $a^{-1}$). The simulated $\delta(^{13}C, CH_4)$ seasonal cycle amplitude is also slightly smaller than the observed one (0.12 ‰ against 0.14 ‰), although the two signals are well phased. Note that our results are however still within the prescribed observation uncertainty range.

For the sake of completeness, we also provide the comparison between $\delta(D, CH_4)$ observations and prior and posterior simulations from INV_DD in Appendix B (Fig. B1). After the inversion, simulations capture much better the observed $\delta(D, CH_4)$ data, reducing the RMSE from 9.3 ‰ to 1.2 ‰. Although $\delta(D, CH_4)$ data is much more limited than $\delta(^{13}C, CH_4)$ data, linear regressions indicate a small negative trend ($-0.23 \pm 0.12$ ‰ $a^{-1}$) between 2005 and 2009. Additionally, the trend is positive between 2005 and 2007 ($+0.86 \pm 0.42$ ‰ $a^{-1}$) and negative between 2007 and 2009 ($-0.47 \pm 0.25$ ‰ $a^{-1}$). It shows $\delta(D, CH_4)$ observations might also carry some information about the post-2007 $CH_4$ renewed growth.

The model-observation agreement varies across the stations both for $CH_4$ and $\delta(^{13}C, CH_4)$ (see Fig. 4a and 4c). $CH_4$ at Marine Boundary Layer (MBL) stations (i.e., where site samples consist mainly of well-mixed MBL air) are very well reproduced by the model (mean RMSE of 17.7 $nmol\,mol^{-1}$ and mean bias of $-1.87$ $nmol\,mol^{-1}$). The model has more difficulties in simulating amount fractions at several polluted stations such as Lac La Biche, Canada (54.95 °N, 112.45 °W), Shangdianzi, People's Republic of China (40.65 °N, 117.12 °E), Anmyeon-do, Republic of Korea (36.54 °N, 126.33 °E) or Southern Great

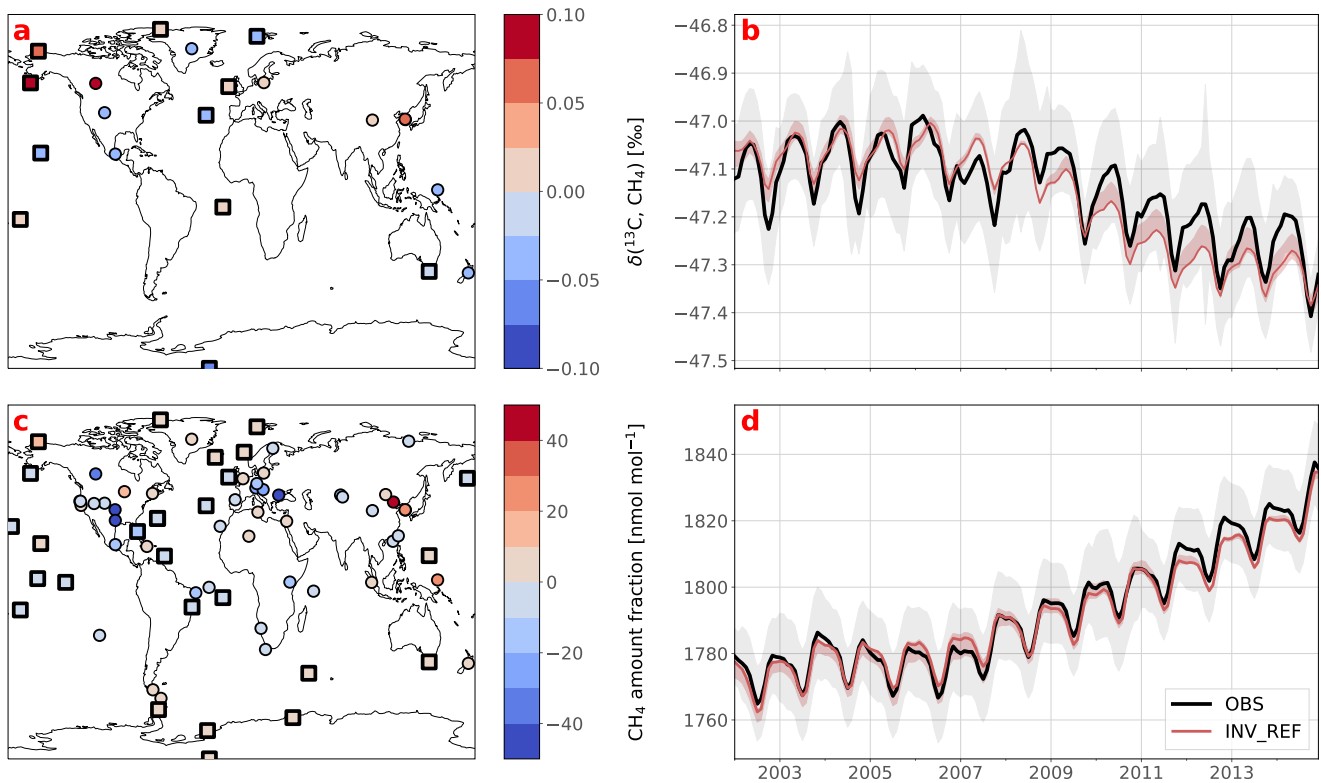

**Figure 4.** Posterior agreement between INV_REF and assimilated observations. a) and c) Mean posterior $CH_4$ and $\delta(^{13}C, CH_4)$ biases at the surface stations over the 2002-2014 period. Marine Boundary Layer stations are indicated by squares rather than circles. b) and d) Observed (black solid line) and simulated (red solid line) globally-averaged trends of $CH_4$ and $\delta(^{13}C, CH_4)$. The red shaded area shows the minimum and maximum values over the sensitivity tests. The grey shaded area shows the standard error of the globally-averaged observed trend. This error is based on the error prescribed in the matrix **R**, i.e., the sum of the measurement and model errors. Same figure with prior data is provided in Appendix B (Fig. B2).

Plains, United States (36.62 °N, 97.48 °W), presumably owing to transport errors, representation errors and/or inaccurate estimates of $CH_4$ prior fluxes around these stations. $\delta(^{13}C, CH_4)$ at MBL stations is generally correctly simulated with RMSEs of 0.2-0.3 ‰, comparable to the prescribed uncertainties. However, posterior simulations slightly overestimate $\delta(^{13}C, CH_4)$ in Northern America and underestimate it in Central America and Temperate North America. It suggests corresponding an over- and under-estimation of flux-weighted source signatures in these regions. This is further investigated in Sect. 3.4.

### 3.2 Comparison of model-optimized with satellite-derived column average $CH_4$ amount fractions

For comparison, we performed one forward simulation with posterior fluxes obtained with INV_REF to compare our simulated $\overline{X}(CH_4)$ to independent (i.e. not assimilated) satellite observations in 2010 and evaluate the optimized atmosphere (see Fig. 5). The posterior mean bias is $-13.0$ nmol mol$^{-1}$, indicating that the GOSAT observations are overall higher than our

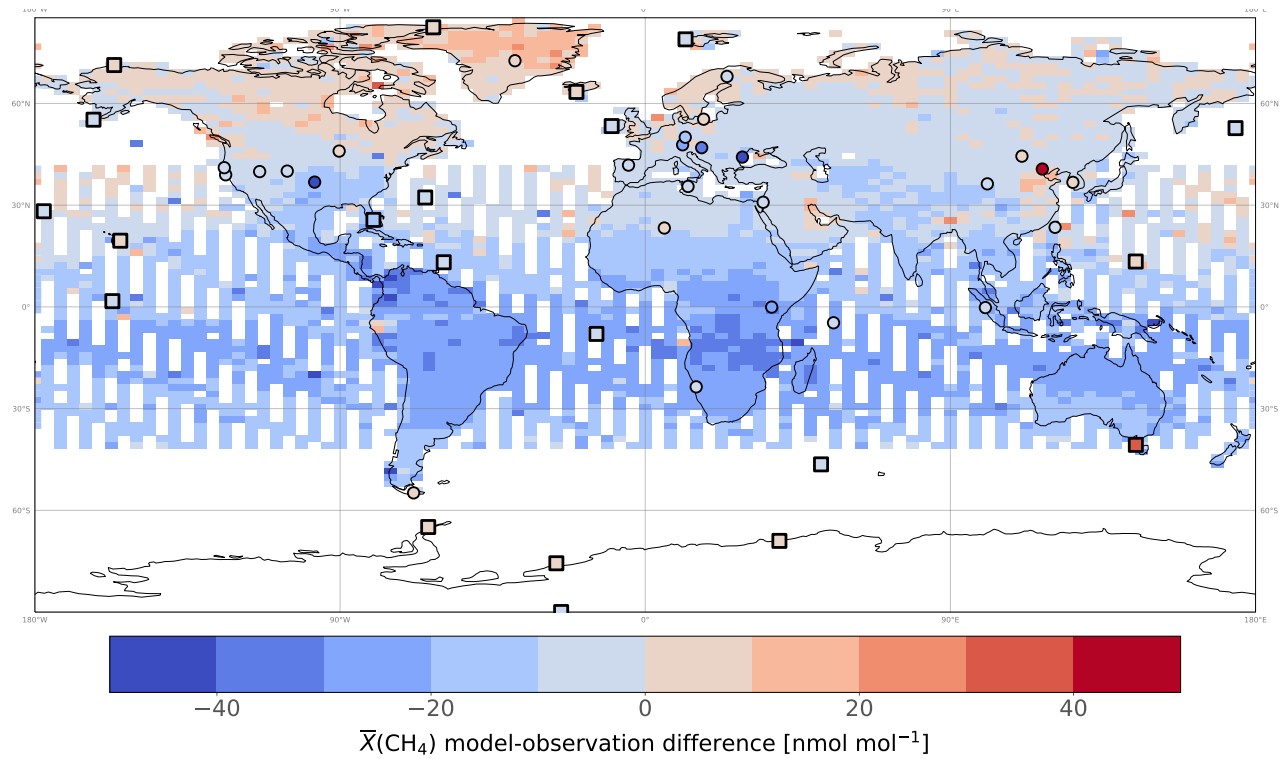

**Figure 5.** $\overline{X}(CH_4)$ mean posterior model-observation differences averaged over 2010 and gridded at the model resolution. Model outputs are obtained using the posterior estimates of INV_REF and applying the averaging kernels provided by the University of Leicester. Model-observation differences at assimilated surface stations are also displayed for comparison. We acknowledge that temporal sampling from surface stations and from satellite data is not identical and might affect the comparison. To limit this effect, only stations providing at least one observation for each month of 2010 are displayed to reduce the seasonal influence. MBL stations are indicated by squares rather than circles.

optimized $\overline{X}(CH_4)$, even after the inversion. Further analysis reveals that the mean bias ($-17.5$ nmol mol$^{-1}$) in the tropics (30 °S-30 °N) is larger than the bias in the northern mid-latitudes (30 °N-60 °N ; $-5.7$ nmol mol$^{-1}$) or in the northern high-latitudes (60 °N-90 °N ; $+3.0$ nmol mol$^{-1}$), in absolute values. Parker et al. (2020) also reported a negative $\overline{X}(CH_4)$ mean bias ($-6.55$ nmol mol$^{-1}$) using the TM5 model and posterior estimates deduced from a surface-based inversion. Similar to
ours, TM5 biases are mostly located in the tropics and northern mid-latitudes. Ostler et al. (2016) reported that model errors in simulating stratospheric $CH_4$ amount fractions could contribute to the $\overline{X}(CH_4)$ bias. However, they did not find a strong improvement for LMDz and TM5 when replacing model simulations with MIPAS (Michelson Interferometer for Passive Atmospheric Sounding) stratospheric $CH_4$. While this suggests that the biases in the GOSAT-simulation presented here are probably not caused by stratospheric discrepancies, it is important to note that the same authors conclude that current satellite
measurements of stratospheric $CH_4$ may lack the precision necessary to eliminate these biases.

If we assume that the biases are solely the result of emission discrepancies, our findings indicate that the estimated tropical posterior emissions from our inversions might still be underestimated. Although posterior biases are lower at the surface stations in the tropics (mean value is $-5.3$ nmol mol$^{-1}$), the number of stations is limited in this area, especially in South America and Central Africa. Saunois et al. (2020), using configurations similar to ours but no isotopic constraints, found that differences between emissions from GOSAT-based and surface-based inversions mainly occurred in the tropical regions.

The in-situ-only and the GOSAT-only inversions performed by Lu et al. (2021) achieved 113 and 212 respective independent pieces of information, highlighting that additional constraints can be gained using satellite data. As tropical fluxes likely had a significant influence on the renewed increase of CH$_4$ around 2007, it would be interesting to assimilate satellite observations to increase the constraints in the tropics. Furthermore, jointly assimilating satellite observations and isotopic observations might be valuable because tropical emissions largely dominate the total release of CH$_4$ in the world. Consequently, a change in tropical emissions might influence the global flux-weighted source signature and impact the results of an inversion performed with isotopic constraints. At present, our system is capable of performing such a joint assimilation. However, since we analyze here the influence of adding isotope constraints, we prefer to assimilate only surface data as a first step.

### 3.3 Posterior-prior emission differences

Global emissions are estimated by INV_REF at 590 Tg a$^{-1}$ when averaged over the 2002-2014 period, larger than prior estimates by 28 Tg a$^{-1}$ (see Fig. 7). This change mainly arises from an increase in Asia ($+15$ Tg a$^{-1}$), Central and South America ($+6$ Tg a$^{-1}$) and Africa ($+3$ Tg a$^{-1}$). About 50 % and 25 % of the increase in Asia is due to the AGW and FFG categories, respectively, suggesting that the prior estimates in EDGARv4.3.2 are underestimated in these regions. However, global emissions estimated by inverse modelling are strongly dependent on the chemical loss prescribed in the CTM. OH is responsible for most of this loss and therefore the prescribed OH field greatly influences the results of the inversion. In particular, Zhao et al. (2020b) showed that a $1 \times 10^5$ cm$^{-3}$ increase in prescribed OH concentrations leads to an increase of CH$_4$ global posterior emissions by 40 Tg a$^{-1}$. Here, we are more interested in the emission trends of various CH$_4$ emission categories and their contributions to total emissions. Therefore, we have only used a unique OH field with several trends. The influence of these trends is further discussed in Sect. 3.8 with the dedicated inversions.

The posterior global distribution of emissions across the individual categories is only slightly different from the prior one (see Fig. 6). Relative posterior-prior emission differences averaged over 2002-2014 are larger for WET ($+7$ %) than for AGW ($+4$ %), FFG ($+6$ %) or BB ($+5$ %). The increase in WET emissions is mainly located in the Amazon basin (43 %) and is responsible for a small shift in the WET contribution to the total emissions (from 32.1 % to 32.6 %), offset by a similar reduction in the AGW contribution (from 37.9 % to 37.4 %). Tropics (90 °S-30 °N), northern mid-latitudes (30 °N-60 °N) and high-latitudes contribute about 60 %, 35 % and 5 % to the global emissions, respectively. Apart from a small reduction in the contribution from high-latitudes, the latitudinal distributions of emissions are not modified.

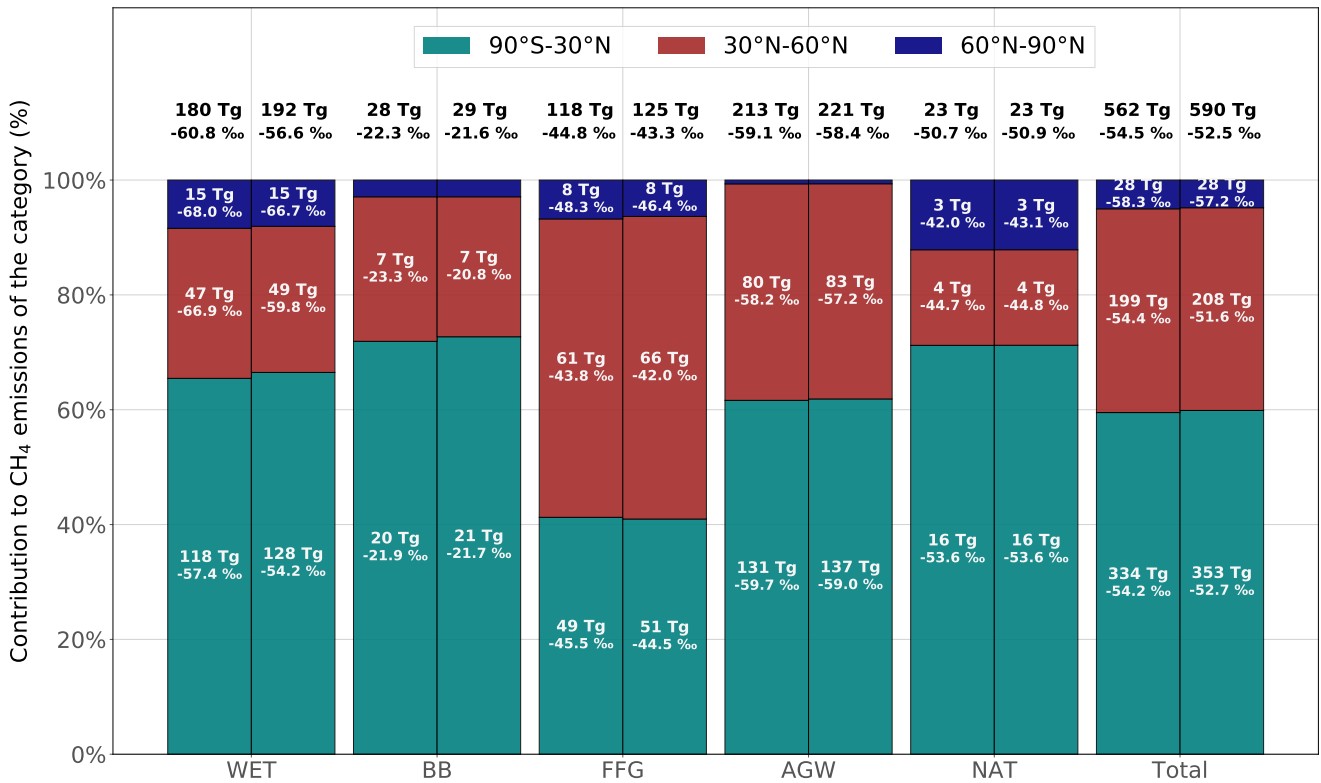

**Figure 6.** Prior (left) and posterior (right) contributions from three latitudinal regions to global CH$_4$ emissions, for each category. Posterior emissions are taken from INV_REF. The latitudinal bands are 1) tropics (90 °S-30 °N), 2) northern mid-latitudes (30 °N-60 °N) and 3) high-latitudes (60 °N-90 °N). Emissions and source signatures for each region and category are given in the associated bars. The total emissions and global source signatures for each category is given on top of each bar.

## 3.4 Posterior-prior source signature differences

Global and regional source isotopic signatures are calculated using a flux-weighted average to account for the global and regional source mixture, respectively. Therefore, they may be modified by the system due to a source mixture change and/or a
source signature change in a specific region.

The inversion system shifts the global source signature considerably upward from −54.5 ‰ to −52.5 ‰ (see Fig. 6). The global signature is highly constrained by the fractionation coefficients and the concentrations of radicals (OH, Cl and O$^1$D) prescribed in the CTM. Our posterior global source signature is indeed higher (less negative) compared to other estimates (Sherwood et al., 2017; Rigby et al., 2017; Schaefer et al., 2016), mainly because we chose to prescribe Cl concentrations
and an OH fractionation that are at the low end of the existing ranges. Each additional percent of oxidation caused by the prescribed Cl sink would approximately lead to a global source signature lower by about 0.5 ‰ (Thanwerdas et al., 2022b; Strode et al., 2020). Using other recent estimates of Cl tropospheric concentrations (see Sect. 2.1), our posterior global source

signature would range between $-54.5$ ‰ and $-52.5$ ‰. In addition, using the fractionation value derived by Cantrell et al. (1990) instead of that derived by Saueressig et al. (2001) would likely shift the global signature downward by another 1.5 ‰.

AGW, BB and FFG source signatures are shifted upward by 0.7, 0.7 and 1.3 ‰ respectively. Most notably, the posterior global flux-weighted WET signature is considerably higher ($-56.6$ ‰) than the prior estimate ($-60.8$ ‰) (see Fig. 6) with regard to recent estimates (Sherwood et al., 2017; Feinberg et al., 2018; Ganesan et al., 2018; Oh et al., 2022). This global shift mainly arises from upward regional source signature shifts in the tropics ($+3.2$ ‰) and in the northern mid-latitudes ($+7.1$ ‰), together contributing 97 % of the posterior-prior global WET isotopic signature difference. The remaining contribution is due

to an increase in the contribution from tropical WET emissions. Our posterior global estimate of WET source signature strongly disagree with the recent estimates. In Appendix B, Fig B3 compares our prior and posterior signatures to observations from the Supplementary Data 1 provided by Oh et al. (2022). Overall, prior estimates show a better agreement with observations than posterior estimates. This poor agreement suggests that prescribed uncertainties might be too large (at least for WET) and supports the idea that the reference inversion yields such an adjustment on WET signature only to stay close to the prior

fossil/microbial flux partitioning. Notably, the isotopic signature in Canada is shifted upward from $-70.0$ ‰ (prior) to $-59.4$ ‰ (posterior) whereas that of Russia is shifted downward from $-68.7$ ‰ to $-73.8$ ‰. Although it demonstrates that the system is capable of applying offsets with different signs across different regions, it also appears unphysical as the processes driving the source signatures in these high-latitude regions are similar (Ganesan et al., 2018; Oh et al., 2022). Nevertheless, the number of observations of WET source signatures is small and local uncertainties are considerable, especially in the tropics where

emissions from WET are the largest and where our system applies the most impactful adjustment. It is therefore difficult to invalidate the posterior adjustment. Further investigation including a better assessment and prescription of random and systematic uncertainties is needed.

Figure B4 and Figure B5, in Appendix B, show the full temporal variations for prior and posterior source signatures. For FF, high variations indicate a change in activities associated to fossil fuel extraction, e.g. switching from one location with a

specific signature to another, transitioning from one fuel type (oil, gas, coal) to another or a combination of both. For example, the substantial shift around 2009 in the United States was caused by a large increase in emissions from the extraction of natural gas. As we chose not to prescribe temporal error correlations between different years, the system is free to optimize each year independently to better fit $\delta(^{13}C, CH_4)$ observations. For certain continental regions, such as Africa, Temperate Asia or South Asia, the inter-annual variability of the source signature adjustments is large and rather unrealistic, especially when compared

to the emission adjustments in the same regions (see Fig. 7). It is unlikely that these changes occurred without detectable changes in emissions in the same areas, especially considering Temperate Asia, which exhibits larger emissions from FF than in the United States. These results suggest the need for prescribing yearly temporal error correlations to dampen this artificial inter-annual variability. However, the example from the United States also indicate that large changes can occur and it is reasonable to assume, considering the lack of isotopic data, that such changes might go unnoticed by the prior data for other

regions. Therefore, while implementing stronger temporal correlations could be a solution to mitigate unrealistic inter-annual variability for this category, it diminishes the likelihood of detecting potential substantial changes that remain undetected by the prior data. Nevertheless, it might be sufficient to reduce the prescribed uncertainties in the source signatures in order to

balance out the pressure applied by the system on the emissions and the source signatures. Overall, the same reasoning applies to AGW, although there is no evidence in the prior data that AGW source signatures can change as rapidly as FF. Due to the scarcity of existing data on the temporal variability of source signatures, designing a data-driven methodology to estimate potential temporal correlations, especially at the regional scale, remains highly challenging. Investigating the correlations that the system creates between the uncertainties associated to source signatures and fluxes could offer a promising avenue for extending the analysis. Due to the high computational cost of an inversion performed with our inversion system, it is impossible to derive robust posterior uncertainties. This impossibility is a major drawback and additional studies with this system cannot be performed in the future without tackling this issue.

### 3.5  Attribution of the post-2007 $CH_4$ increase

We now present trend results, comparing the time period before (2002-2007) and after (2007-2014) the renewed increase. Global posterior emissions show a net increase of 24.0 $Tg\,a^{-1}$ (see Fig. 7). It occurred in most of the regions, besides Europe ($-1.8\ Tg\,a^{-1}$), Canada ($-0.7\ Tg\,a^{-1}$) and Oceania ($-0.2\ Tg\,a^{-1}$). China, South Asia (mainly India), Temperate Asia, South-East Asia and Africa accounted for 40, 18, 18, 10 and 9 % of the associated positive increase ($+26.7\ Tg\,a^{-1}$), respectively. These results are consistent with prior information that estimated a rise of 27.5 $Tg\,a^{-1}$. The large contribution from China to the global increase since 2002 agrees well with the regional estimate (40 %) from Thompson et al. (2015). We also estimate that Central and South America did not contribute to the renewed growth, in contrast with Chandra et al. (2021) who suggest a large contribution from Brazil (11.5 %) in the global emission increase. In this region, we find that small increases in AGW emissions ($+1.4\ Tg\,a^{-1}$) and FFG emissions ($+0.7\ Tg\,a^{-1}$) are offset by decreases in WET emissions ($-0.8\ Tg\,a^{-1}$) and BB emissions ($-0.8\ Tg\,a^{-1}$).

Global AGW emissions increased by 14.2 $Tg\,a^{-1}$ between 2002-2007 and 2007-2014. Europe is the only region where these emissions substantially decreased ($-1.7\ Tg\,a^{-1}$). 80 % of the net AGW increase occurred in Asia, and the rest in Africa and South America. FFG emissions increased by 14.9 $Tg\,a^{-1}$, with 50 % of this increase occurring in China. These emissions notably decreased in Africa ($-0.7\ Tg\,a^{-1}$), Europe ($-0.2\ Tg\,a^{-1}$) and Canada ($-0.1\ Tg\,a^{-1}$). By contrast, WET emissions decreased by 2.4 $Tg\,a^{-1}$, with 71 % of the net decrease located in Central and South America (33 %), Canada (25 %) and Africa (13 %). BB emissions also decreased by 2.7 $Tg\,a^{-1}$, mainly in South-East Asia (55 % of the net decrease) and South America (24 %). Note the analysis period does not include the 2015 El Niño event.

Our results therefore suggest that the post-2007 $CH_4$ renewed growth (until 2014) was equally and mainly driven by increases in AGW (49 %) and FFG (51 %) global emissions. The decreases in WET and BB global emissions as well as the increase in OH global concentrations partially balanced this renewed growth. These findings are in partial agreement with recent studies (Chandra et al., 2021; Jackson et al., 2020; Thompson et al., 2018; Saunois et al., 2017), although only Jackson et al. (2020) explained the renewed growth with equal contributions from the AGW and FFG categories. The small decrease in BB emissions is consistent with other estimates (Thompson et al., 2018; Worden et al., 2017) but the decrease in WET emissions does not agree with recent findings that suggest either a constant trend or a positive trend (Chandra et al., 2021; Zhang et al., 2018; McNorton et al., 2018; Poulter et al., 2017; Bader et al., 2017).

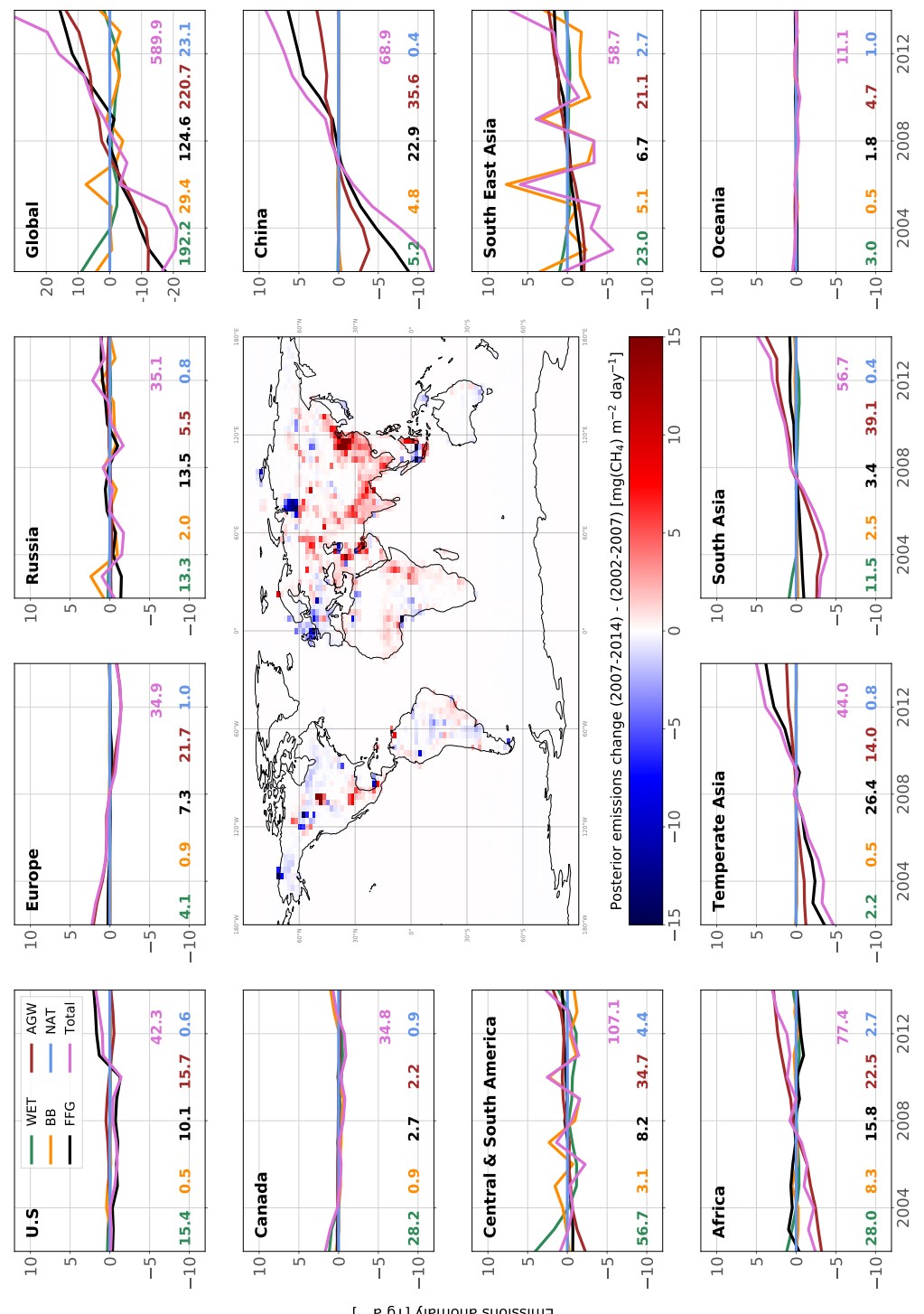

**Figure 7.** Map of posterior emissions change from INV_REF between 2002-2007 and 2007-2014 (center panel) and time-series of posterior emission estimates from INV_REF for multiple regions and all categories (panels around the map). For each panel, time-series are anomalies around a 2002-2014 mean value. For each category of each panel, the associated mean value is displayed in the same color as the solid line. Units of variations and means are Tg a$^{-1}$. Design inspired by Figure 1 in Chandra et al. (2021).

However, posterior global WET emissions show negative anomalies between 1998 and 1999 and positive anomalies between 1999 and 2004. These anomalies are mainly located in South America, where about 30 % of WET emissions originate. Zhang et al. (2018) suggested that the 1998-2000 ENSO (El Niño-Southern Oscillation) caused negative anomalies of WET emissions between 1998 and 2000 because of El Niño and subsequent positive anomalies between 2000 and 2002 because of La Niña. The fact that positive anomalies persist until 2004 rather than 2002 in our posterior emissions cannot be easily explained. Also, the positive anomalies last 4-5 years in total, which is not consistent with the 2-3 years inferred by Zhang et al. (2018). As AGW emissions are also large in South America, the inversion system might be wrongly attributing large but decreasing emissions between 2002 and 2004 to WET emissions rather than AGW emissions. If the period 2002-2004 that exhibit large positive anomalies is discarded, we find a small increase of 0.3 Tg a$^{-1}$ in global WET emissions between 2004-2007 and 2007-2014, therefore more consistent with the studies mentioned before.

## 3.6 Attribution of the post-2007 $\delta(^{13}C, CH_4)$ downward shift

The posterior global flux-weighted $\delta_{\text{source}}(^{13}C, CH_4)$ decreased from $-52.1$ ‰ in 2002 to $-53.1$ ‰ in 2010 and and then experienced an upturn to $-52.5$ ‰ in 2012-2014. Notably, between 2007 and 2010, the decline in source signature was rapid ($-0.2$ ‰ a$^{-1}$), propagating into the atmosphere and contributing to the similar trend that appears in the observed globally-averaged $\delta(^{13}C, CH_4)$. The subsequent increase after 2012 lead to a stabilization of the associated atmospheric signal.

Between 2002-2007 and 2007-2014, all regional isotopic signatures were shifted downward (about $-0.5$ ‰ over the globe), except in China ($+0.6$ ‰). This is mainly explained by an increase in emissions from $^{13}C$-depleted sources in most of the regions but also a decrease in emissions from $^{13}C$-enriched sources and a decrease in AGW and FFG source signatures.

Additionally, we use a simple mathematical framework to attribute the shift in global flux-weighted signature to the different emission categories. Here, $\bar{\delta}$ denotes the global flux-weighted $\delta_{\text{source}}(^{13}C, CH_4)$. A first-order estimate of $\bar{\delta}$ is given by:

$$\bar{\delta} = \sum_{i=1}^{N} (\delta_i \times \frac{f_i}{F}) \tag{5}$$

$f_i$ denotes the global CH$_4$ emissions from a specific category, $N$ the number of emission categories (five here), $F = \sum_{i=1}^{N} f_i$ the total CH$_4$ emissions and $r_i = \frac{f_i}{F}$ the contributions from each category to the total emissions. A small variation of the global flux-weighted source signature, $d\bar{\delta}$, can therefore be calculated using the derivatives $d\delta_i$ and $df_i$:

$$d\bar{\delta} = \sum_{i=1}^{N} (\frac{\partial \bar{\delta}}{\partial \delta_i} \times d\delta_i) + \sum_{i=1}^{N} (\frac{\partial \bar{\delta}}{\partial f_i} \times df_i) \tag{6}$$

with

$$\begin{cases} \dfrac{\partial \bar{\delta}}{\partial \delta_i} = \dfrac{f_i}{F} \\ \dfrac{\partial \bar{\delta}}{\partial f_i} = \delta_i \times \dfrac{(1 - r_i)}{F} - \sum_{\substack{j=1 \\ j \neq i}}^{N} (\delta_j \times \dfrac{f_j}{F^2}) \end{cases} \tag{7}$$

Using this simplified linear relationship, we find that the 0.34 ‰ decrease in $\bar{\delta}$ (see Fig. 8) between 2002-2007 and 2007-575 2014 was due to:

1. a decrease in AGW global source signature (resulting in a shift in $\bar{\delta}$ by $-0.22$ ‰)

2. a small decrease in BB emissions (resulting in a shift in $\bar{\delta}$ by $-0.15$ ‰)

3. a large increase in AGW emissions (resulting in a shift in $\bar{\delta}$ by $-0.14$ ‰)

4. a decrease in FFG isotopic signature (resulting in a shift in $\bar{\delta}$ by $-0.09$ ‰).

This decrease is partially offset by:

5. a large increase in FFG emissions (resulting in a shift in $\bar{\delta}$ by $+0.24$ ‰)

6. a small decrease in wetlands emissions (resulting in a shift in $\bar{\delta}$ by $+0.02$ ‰)

**Table 5.** Upper part of the table: Changes in $CH_4$ emissions (emi.) and isotopic signatures (sign.) between 2002-2007 and 2007-2014 from each emission category, for each sensitivity tests. Lower part of the table: contributions from emissions and isotopic signatures changes to the global flux-weighted source signature ($\bar{\delta}$) shift between 2002-2007 and 2007-2014 from each emission category, for each sensitivity test.

| | Total | | AGW | | FFG | | WET | | BB | |
|---|---|---|---|---|---|---|---|---|---|---|
| | Emi. | Sign. | Emi. | Sign. | Emi. | Sign. | Emi. | Sign. | Emi. | Sign. |
| Sensitivity test | Change in global emissions (in $\mathrm{Tg\,a^{-1}}$) and global flux-weighted source signature (in ‰) between 2002-2007 and 2007-2014 | | | | | | | | | |
| PRIOR INV_REF | +27.5 | +0.09 | +14.6 | −0.07 | +14.8 | +0.48 | +0.0 | +0.00 | −2.0 | +0.04 |
| INV_REF | +24.0 | −0.34 | +14.2 | −0.60 | +14.9 | −0.42 | −2.4 | +0.00 | −2.8 | +0.18 |
| INV_DD | +22.7 | −0.32 | +13.9 | −0.60 | +14.7 | −0.34 | −3.0 | −0.00 | −2.9 | +0.16 |
| INV_CH4 | +28.6 | N/A | +15.3 | N/A | +14.5 | N/A | +1.1 | N/A | −2.3 | N/A |
| INV_LOCKED | +27.1 | −0.36 | +13.6 | −0.08 | +17.1 | +0.53 | +6.2 | −0.01 | −9.8 | +0.07 |
| INV_FLATOH | +17.5 | −0.35 | +12.2 | −0.61 | +12.8 | −0.44 | −4.8 | −0.02 | −2.7 | +0.17 |
| INV_TURNER | −8.9 | −0.50 | +4.60 | −0.67 | +5.5 | −0.74 | −15.4 | −0.15 | −3.6 | +0.17 |
| Sensitivity test | Contribution of changes in emissions and source signatures to the $\bar{\delta}$ shift for the different emission categories | | | | | | | | | |
| PRIOR INV_REF | N/A | +0.09 | −0.12 | −0.02 | +0.26 | +0.10 | +0.00 | +0.00 | −0.12 | +0.00 |
| INV_REF | N/A | −0.34 | −0.14 | −0.22 | +0.24 | −0.09 | +0.02 | +0.00 | −0.15 | +0.01 |
| INV_DD | N/A | −0.32 | −0.14 | −0.22 | +0.24 | −0.07 | +0.02 | +0.00 | −0.16 | +0.01 |
| INV_LOCKED | N/A | −0.36 | −0.16 | −0.03 | +0.24 | +0.13 | −0.08 | +0.00 | −0.50 | +0.01 |
| INV_FLATOH | N/A | −0.35 | −0.12 | −0.23 | +0.21 | −0.09 | +0.04 | −0.01 | −0.15 | +0.01 |
| INV_TURNER | N/A | −0.50 | −0.05 | −0.25 | +0.09 | −0.15 | +0.10 | −0.05 | −0.19 | +0.01 |

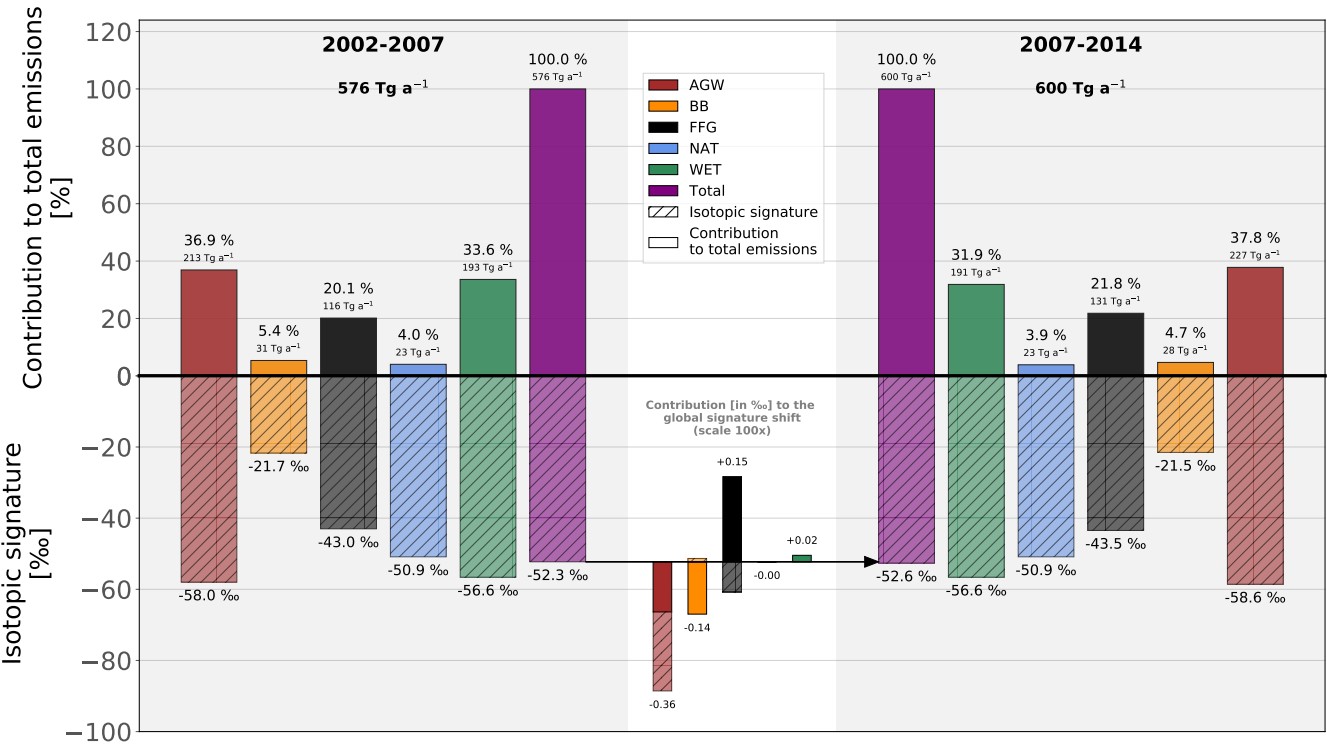

**Figure 8.** Contribution of the changes in $CH_4$ emissions and $\delta_{\mathrm{source}}(^{13}C, CH_4)$ source isotopic signatures to the global source signature shift between 2002-2007 (left side) and 2007-2014 (right side) in INV_REF. The upper part of the figure shows the contributions from the individual emission categories to total emissions. Associated bars are non-transparent and non-hashed. Percentages and emissions are displayed on top of the bars. The lower part of the figure shows the isotopic signatures of each category. Associated bars are slightly transparent and hashed. The lower center with a white background shows the contributions from emissions (non-transparent and non-hashed) and source isotopic signatures (slightly transparent and hashed) changes to the total source signature shift ($-0.34$ ‰) between the two periods. This part is magnified (x100) for clarity. Results from the other sensitivity tests are given in Table 5.

### 3.7 Sensitivity of the results to isotopic constraints

The reference inversion assimilates both $CH_4$ and $\delta(^{13}C, CH_4)$. To quantify the impact of assimilating $\delta(^{13}C, CH_4)$ data,

INV_CH4 assimilates $CH_4$ observations only and does not simulate the isotopic composition. Differences between INV_REF and INV_CH4 therefore provide insight into the influence of the isotopic constraint. Notably, these two inversions show similar results for $CH_4$ emissions. As both inversions are constrained by the same global sink, global emissions estimated by INV_CH4 are only 0.3 Tg a$^{-1}$ lower on average over the 2002-2014 period. Tropical emissions are increased compared to INV_REF and the contribution from tropical emissions to total emissions is shifted from 59.8 % to 60.3 % (+2.5 Tg a$^{-1}$), mainly due to an

increase in WET tropical emissions (+1.6 Tg a$^{-1}$) offset by decreases in the northern mid-latitudes (−1.2 Tg a$^{-1}$) and high-latitudes (−0.2 Tg a$^{-1}$). WET emissions increase by 1.1 Tg a$^{-1}$ in INV_CH4 between 2002-2007 and 2007-2014 (Table 5),

instead of decreasing in INV_REF. The increases in AGW, FFG and WET emissions contribute 50 %, 47 % and 3 % to the post-2007 renewed growth, respectively, and are therefore slightly different from the results of INV_REF. To summarize, adding the isotopic constraint (INV_REF as compared to INV_CH4) slightly decreases the contribution from tropical emissions to global

emissions, removes a very small contribution from WET emissions to the post-2007 renewed growth and slightly changes the contributions from emission increases to the post-2007 renewed growth.

Differences between INV_REF and INV_CH4 are small presumably as a result of the large prior uncertainties in source signatures that allow the inverse system to adjust the atmospheric isotopic compositions at a low cost by changing signatures rather than emissions. To test this hypothesis, we run INV_LOCKED assuming a perfect knowledge of isotopic signatures

(no uncertainties in the prior source isotopic signatures). Although the global total emissions obtained with INV_REF and INV_LOCKED are very similar, the individual contributions from each emission category are modified (Fig. 9, panel c). On average over the 2002-2014 period, FFG emissions are increased by 24 % compared to INV_REF, mainly due to large relative increases in China and Middle-East (+30 to 50 %). Global FFG emissions amounts to 153 $Tg\,a^{-1}$, therefore more consistent with the large revisions (150-200 $Tg\,a^{-1}$) derived by (Schwietzke et al., 2016) with recent isotopic data. WET emissions

located in boreal regions and in South America are decreased by around 30 % whereas WET emissions from Central Africa are slightly increased, leading to a global WET emissions decrease by 14 %. Finally, BB emissions are increased by 41 % and AGW emissions are slightly decreased by 7 %, with globally-uniform changes (Fig. 9, panels a and b). Furthermore, INV_LOCKED explains the renewed growth with contributions from enhanced FFG (46 %), AGW (37 %) and WET (17 %) between 2002-2007 and 2007-2014 (Table 5). WET emissions therefore actively participate in the post-2007 $CH_4$ growth in

INV_LOCKED, as opposed to INV_REF. The FFG, AGW and WET emissions increases are however offset by a large decrease in BB emissions, nearly three times larger than in the other inversions. Also, emission IAVs are increased for all categories. In particular, BB emissions peaks in 2006 and 2009 are much higher in INV_LOCKED than in INV_REF, relatively to the mean over the period. Such variations are probably too large to be realistic when compared to prior data and other inversion studies (Chandra et al., 2021; Basu et al., 2022, e.g.,). However, they provide an upper bound for emission trends as constrained by

isotopic values. In many inversion studies (e.g., Rice et al., 2016; Schwietzke et al., 2016; Turner et al., 2017; Rigby et al., 2017; Thompson et al., 2018; McNorton et al., 2018), isotopic signatures are fixed and our results suggest that this may lead to significant errors in $CH_4$ emission trends. It stresses the importance to find the right balance between over-constrained signatures, as in INV_LOCKED, and likely under-constrained signatures as in INV_REF. At present, isotopic constraints are either too loose to bring critical information about sectorial and regional $CH_4$ emissions or our estimates of the associated

uncertainties are over-estimated in our methodology, which is also a possibility that we will address in future studies.

Finally, assimilating $\delta(D,CH_4)$ observations and optimizing $\delta_{source}(D,CH_4)$ source signatures in INV_DD have a very small influence on our posterior emission estimates, as indicated in Table 5. The most significant difference observed is a small positive shift of +0.5 ‰ in the BB posterior source signature compared to INV_REF. Consequently, with our setups, assimilating $\delta(D,CH_4)$ does not appear to provide any substantial additional constraint on the $CH_4$ budget estimate. Several factors

may contribute to this result: 1) the existing network provides comparatively fewer $\delta(D,CH_4)$ observations in comparison to $\delta(^{13}C,CH_4)$ observations, 2) $\delta(D,CH_4)$ observations spans only the period from 2005 to 2010 and therefore, the full run

cannot be fully constrained by this data and 3) the constraints may be too weak due to an overestimation of the prescribed uncertainties in $\delta_{\text{source}}(D, CH_4)$ sources signatures. As including $\delta(D, CH_4)$ in the inversion doubles the computational cost compared to a setup like INV_REF, we recommend not assimilating $\delta(D, CH_4)$ in our system until either the computational cost can be reduced, more observations become available or lower uncertainties are established. However, a hypothetical network of $\delta(D, CH_4)$ measurements, obtained at a reasonable frequency and spanning a longer period of time could efficiently complement $\delta(^{13}C, CH_4)$ observations and provide a wealth of information (Rigby et al., 2012). More specifically, reactions with OH, $O^1D$ and Cl have fractionation coefficients that depend on the isotope. Therefore, incorporating $\delta(D, CH_4)$ constraints might help to disentangle the effects of the associated sinks and provide additional insights into the global sink and its mixture. However, optimizing the sinks introduces additional degrees of freedom and complexifies the inverse problem. With the current system and at such a high resolution for the optimized variables, we recommend against the simultaneous optimization of both the source signatures and the sinks. However, a coarser resolution for the optimized variables, or at least for the sink, might be able to accommodate a simultaneous optimization.

## 3.8 Sensitivity of the results to OH IAV

Last but not least, we have tested the impact of OH trends on our results. INV_FLATOH and INV_REF show very similar results (Table 5). The main difference is that INV_FLATOH infers a smaller increase in total emissions ($+18 \, \text{Tg} \, a^{-1}$) between 2002-2007 and 2007-2014 than in INV_REF ($+24 \, \text{Tg} \, a^{-1}$). As a smaller sink is prescribed in INV_FLATOH compared to INV_REF, the increase in total emissions required to fit the observations of $CH_4$ amount fractions is also smaller. The contributions from each emission categories to total emissions are little affected ($\pm 0.2 \, \%$). The contributions from AGW and FFG emissions to the increase in the total emissions between the two periods are exactly the same as in INV_REF. In addition, as the inter-hemispheric OH ratio is not modified, the contributions from tropics, mid- and high-latitudes are identical for each emission category. Overall, the differences between INV_REF and INV_FLAT are negligible and do not affect the conclusions deduced from the INV_REF results.

On the contrary, INV_TURNER infers a decline by 1.6 % in global emissions between 2002-2007 and 2007-2014 (Table 5), mainly driven by a large decrease in WET emissions ($-15 \, \text{Tg} \, a^{-1}$) and a slightly larger decrease in BB emissions ($-4 \, \text{Tg} \, a^{-1}$) than in INV_REF. With this prescribed OH IAV, the post-2007 renewed $CH_4$ growth is therefore entirely caused by a large decline in the global OH sink between the two periods. Changes in AGW and FFG emissions in INV_TURNER are still positive but 2-3 times smaller than in INV_REF. Using all the information provided by the sensitivity tests inferring a net increase in emissions (i.e. without INV_TURNER), this increase is principally attributed to fossil sources ($50 \pm 3 \, \%$) and agriculture and waste sources ($47 \pm 5 \, \%$). Nevertheless, there is a substantial variation in the results between configurations that optimize source signatures and those that do not.

The decline in the OH sink between the two periods affects the $\delta(^{13}C, CH_4)$ atmospheric signal in two opposite ways:

1. If the OH sink is the only sink, a decline in OH concentrations has no effect on $\delta(^{13}C, CH_4)$ in the long term (several decades) because the mean fractionation is not affected. However, in the short term (a decade), as OH concen-

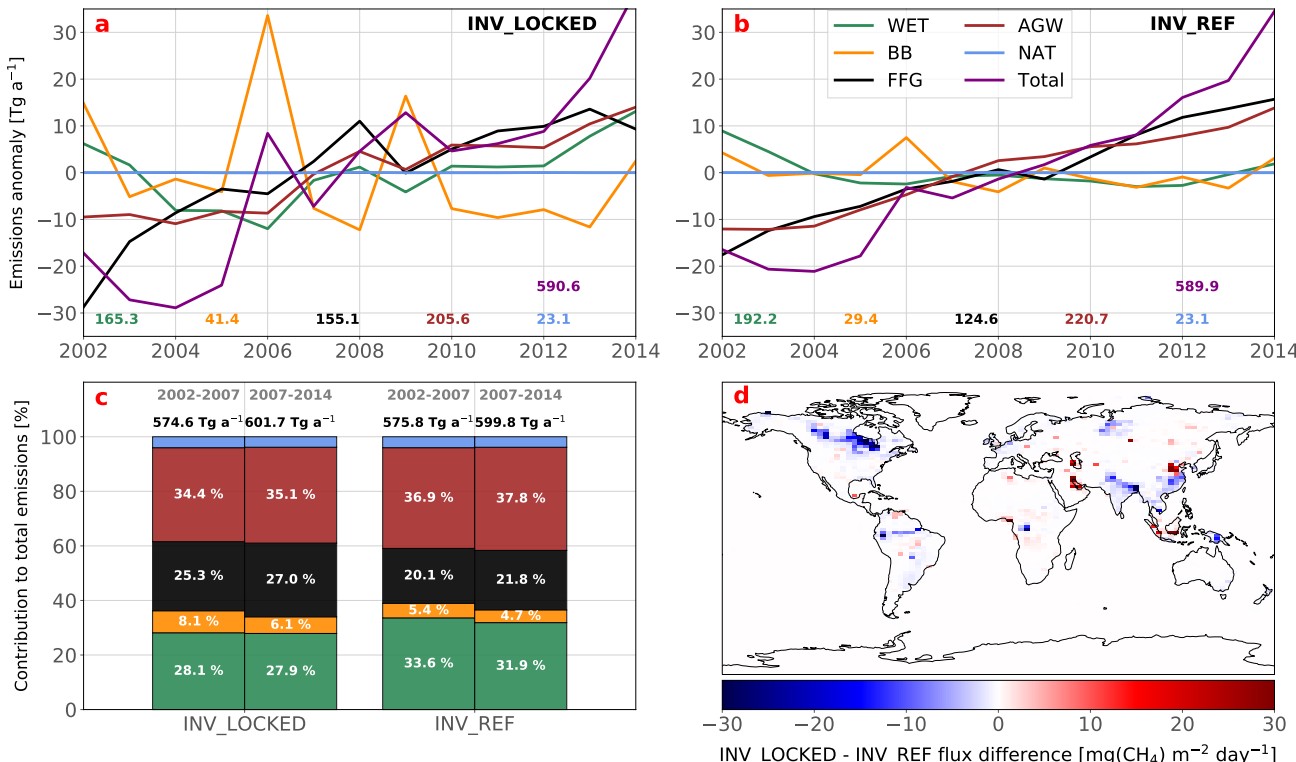

**Figure 9.** Comparison between INV_REF and INV_LOCKED results. Upper panels show the time-series of emissions estimated by INV_REF (panel a) and INV_LOCKED (panel b). For these panels, time-series are anomalies around a 2002-2014 mean value. For each category of these panels, the associated mean value is displayed in the same color as the solid line. Lower panels show the contributions from each emissions category to total emissions for 2002-2007 and 2007-2014 (panel c) and a map of the posterior total emissions differences between INV_LOCKED and INV_REF averaged over the 2002-2014 period (panel d).

trations decrease, $^{12}CH_4$ and $^{13}CH_4$ atmospheric lifetimes increase. Due to the fractionation effect, there is a time lag between increases in $^{12}CH_4$ and $^{13}CH_4$ amount fractions. $^{12}CH_4$ accumulates faster than $^{13}CH_4$, leading to a decrease in $\delta(^{13}C, CH_4)$.

2. The total fractionation effect in the atmosphere is the result of an average of all fractionation effects associated to the different sinks (OH, O$^1$D, Cl, soils) weighted by their contributions to the total sink. Therefore, if the OH sink is reduced, the contributions from the other sinks (with larger fractionation effects) increase. Consequently, the total fractionation effect is also increased and $\delta(^{13}C, CH_4)$ values are shifted upward.

As the INV_TURNER infers a more depleted global source signature change ($-0.50$ ‰) than INV_REF ($-0.34$ ‰), we can conclude that the $\delta(^{13}C, CH_4)$ downward shift induced by the first mechanism is smaller than the upward shift induced by the second mechanism, resulting in a net upward shift. The enhanced depletion of the global source signature counterbalances this

net upward shift. In the inversion, such a depletion is mainly obtained by lowering the source signatures of AGW, FFG and WET sources between 2002-2007 and 2007-2014. Compared to INV_REF, the shifts in source signatures are almost identical for AGW but much larger for FFG and WET. The negative OH trend has been obtained by Rigby et al. (2017) and Turner et al. (2017) with box modelling and methyl-chloroform constraints (Patra et al., 2021). However, Naus et al. (2019) suggested that inter-hemispheric transport, stratospheric loss and source/sink spatial distributions are not properly represented using box

modelling, resulting in significant errors. They found a positive OH trend over the 1994–2015 period with a 3-D model, more consistent with the IAV of our OH-INCA field. Other studies agree with Naus et al. (2019) in finding a small or positive IAV for the recent years (Montzka et al., 2011; Lelieveld et al., 2016; Nicely et al., 2018). Therefore, the results from INV_TURNER inversion seem to be rather unlikely.

### 3.9 Comparison with Basu et al. (2022)

Basu et al. (2022), hereinafter in this subsection referred to as BA22, quantify the global $CH_4$ budget and investigate the post-2007 renewed growth using the TM5-4DVAR inversion framework to assimilate both $CH_4$ and $\delta(^{13}C, CH_4)$ measurements. In our opinion, their work is strongly relevant and tackles this complex topic with an appropriate and robust methodology. As our goals are similar, we compare here our systems and methodologies.

First, it is worth mentioning that our system is capable of assimilating $\delta(D, CH_4))$ observational data and optimizing the

685 associated source signatures. Although this feature has a small influence on our results in this work, we believe that its relevance will grow as more $\delta(D, CH_4))$ data become available and the associated uncertainties decrease.

As already stated in their paper, the main difference between our systems is the optimization of source signatures. BA22 prefer to investigate the influence of the source signature uncertainties with different sensitivity tests adopting various source signature maps. This choice relies on the fact that they can run a large number of inversions at low cost using a parallel

configuration. It is a good strategy to assess the influence of systematic errors in source signatures. In our work, we did not investigate this influence and we decided to optimize source signatures in order to consistently account for random errors in source signatures and emissions at the same time. As both interact and impact the atmospheric composition in very complex ways, it seemed important to us to perform at least one inversion combining all the uncertainties.

We agree with BA22 that the second major difference between the two studies lies in the construction of the prior $CH_4$

fluxes. However, they suggest that we constructed a prior that approximately matches the atmospheric $CH_4$ growth rate in Thanwerdas et al. (2022a). In Thanwerdas et al. (2022a) and in the present study, we derived our prior fluxes and source signatures solely on the basis of bottom-up estimates and literature data. For $CH_4$ emissions, the fact that prior simulations match the atmospheric $CH_4$ growth rate show that bottom-up estimates are roughly consistent with atmospheric data, even before the inversion process.

We prefer not to adjust prior fluxes to match observational $\delta(^{13}C, CH_4)$ data because in this case, we assume that bottom-up estimates suffer strong systematic uncertainties, which is difficult to demonstrate. Adjusting prior $CH_4$ fluxes also assumes that $CH_4$ emissions derived by bottom-up estimates are more likely to be wrong than source signatures estimates. We believe that the opposite is more plausible because observational data for source signatures is very scarce at present. Therefore, we prefer to

start from robust and validated data and let the inversion system combine them with the assimilated atmospheric observations and the random uncertainties. BA22 start with a flat prior and the posterior results deviate significantly from the prior. As the prior data do not seem to have a strong influence on the posterior results, adjusting prior fluxes prior to the inversion should have no effect on the results. However, this is yet to be confirmed in our case, i.e. with a nonlinear observation operator.

BA22 are able to calculate posterior uncertainties using a large ensemble of inversions. It is a precious feature that we do not possess at present. This is made possible by the relatively low computational cost of their configuration (adjusted prior and linear formulation), but also by the fact that they divide the full assimilation window into shorter sub-windows (5 years) that are run in parallel. A one-year overlap with previous and next sub-windows is applied. It would be interesting to compare the posterior results obtained with this parallelised configuration to an inversion with a complete assimilation window. As the relaxation time for isotopic composition in the atmosphere in response to a perturbation is much larger (decades Tans, 1997) than for $CH_4$ itself, we are concerned that using such short time periods might affect posterior results, especially if the observation operator is nonlinear. Modifying the prior data to fit the observed isotopic composition as in BA22 might be a prerequisite for the success of this method.

It is clear that these setup differences propagate to posterior results. Using the additional $\delta(^{13}C, CH_4)$ data, BA22 find that fossil $CH_4$ emissions and microbial emissions contributed about 15 % and 85 %, respectively, to the post-2007 $CH_4$ growth. As presented in the previous sections, our results are completely different. Most notably, they find a contribution of 30 % from fossil emissions to total emissions on average over 1999-2016. While our reference inversion finds a much smaller number (21 %), our inversion with fixed source signatures (INV_LOCKED) gives a closer value (26 %). The small source partitioning discrepancy between INV_LOCKED and BA22's inversion results might also be caused by a difference in prescribed isotopic fractionation, as suggested by the sensitivity analysis of Lan et al. (2021).

We cannot fully explain why BA22's conclusions about the causes of the post-2007 $CH_4$ renewed growth differ so substantially from our own. However, it appears that BA22 also use a robust methodology to study the global $CH_4$ budget and the renewed growth. Despite significant differences, we find good complementarity between our approaches and hope to learn from each other in order to improve our systems and reconcile our results.

## 4   Conclusion and discussion

We used variational inversion modelling with the 3-D CTM LMDz-SACS to investigate the drivers of the post-2007 renewed growth of atmospheric $CH_4$. We assimilated $CH_4$, $\delta(^{13}C, CH_4)$ and $\delta(D, CH_4)$ atmospheric observations, and optimized both fluxes and source isotopic signatures of five independent emission categories for the period 1998-2017. Implementing multiple setups allowed us to investigate the influence of isotopic constraints and OH IAV on our results.

Most of our inversions find the post-2007 renewed growth was caused, with equal contributions (51-49 %), by large increases in fossil fuels and geological emissions (FFG) as well as in agriculture and waste (AGW) emissions between 2002-2007 and 2007-2014. These were partially balanced by small decreases in wetlands (WET) emissions and biofuels and biomass burning (BB) emissions and a small OH increase during this period.

Isotopic constraints, i.e., assimilating $\delta(^{13}\text{C}, \text{CH}_4)$ and $\delta(\text{D}, \text{CH}_4)$ observations, have only little influence on the posterior emission estimates. Compared to a $\text{CH}_4$-only inversion, an inversion assimilating $\delta(^{13}\text{C}, \text{CH}_4)$ observations and optimizing source signatures only slightly reduces tropical emissions ($-2.5 \text{ Tg a}^{-1}$), mainly from wetlands. Notably, the global flux-weighted WET source signature is shifted upward (less negative) due to a shift in the tropics ($+3.2 \text{ \textperthousand}$) and in the northern mid-latitudes ($+7.1 \text{ \textperthousand}$). To fit the $\delta(^{13}\text{C}, \text{CH}_4)$ observations, the system prefers to adjust the source signatures than the $\text{CH}_4$ emissions. Surely, the large uncertainties associated with source signatures make them less costly to modify. Our findings also reveal that the global downward shift $\delta(^{13}\text{C}, \text{CH}_4)$ between 2002-2004 and 2007-2014 was caused by an increase in $^{13}\text{C}$-depleted AGW emissions and a decrease in $^{13}\text{C}$-enriched BB emissions but also by decreases in AGW and FFG source signatures. For example, a small change compared to uncertainties in AGW source signatures ($-0.6 \text{ \textperthousand}$) between the two periods results in a $-0.24 \text{ \textperthousand}$ downward shift of the global source signature in the reference inversion. These results might be very dependent on the prescribed Cl concentrations, especially in the troposphere, and we decided to use the most recent and consistent Cl concentration estimates to minimize the associated error.

If the $\delta_{\text{source}}(^{13}\text{C}, \text{CH}_4)$ source signatures are considered to be perfectly known, i.e., without uncertainties, the relative contributions of the different emissions categories are significantly changed by the inversion. Contributions from FFG and BB emissions are increased and those from AGW and WET emissions are decreased. In addition, WET emissions are found to contribute (13 %) to the post-2007 renewed growth with AGW (37 %) and FFG (46 %) emissions. Such a partition between fossil and microbial sources is more consistent with recent inversion estimates based on isotopic data. However, none of these recent results account for random uncertainties in source signatures. It shows that reducing the prescribed uncertainties in source signatures is a necessary condition for providing more accurate emission estimates when assimilating isotopic data.

OH IAV has also an influence on the results when a negative trend consistent with the IAV inferred by Turner et al. (2017) is applied. In this case, the post-2007 renewed growth is entirely caused by the decline in OH concentrations and AGW and FFG emissions only slightly increase over the 2002-2014 period. As recent findings suggest that such a decrease in OH concentrations is unlikely, the results from the other sensitivity inversions should be preferentially considered. Overall, using all the information provided by the sensitivity tests presented in this work, the net increase in global emissions is principally attributed to fossil sources ($50 \pm 3$ %) and agriculture and waste sources ($47 \pm 5$ %).

As this new inversion setup (with isotopic constraints) is used over a long time period for the first time, methods are deliberately simplified in order to provide a background for future inversions and improvements. For instance, our methods to prescribe error statistics in the matrix **B** have obviously room for improvement, even with the limited amount of data available at the present time. The uncertainties we prescribed in source signatures in the reference inversion might be slightly overestimated. A more robust estimate of current regional random uncertainties in source signatures is necessary before running other inversions with isotopic data.

Also, the main limitation of our inversion system is the associated computational cost and the absence of posterior uncertainties. Formally, posterior uncertainties are given by the Hessian of the cost function (Meirink et al., 2008). This matrix can hardly be computed at an achievable cost, considering the size of the inverse problem. Other means must be implemented to obtain the posterior uncertainty, such as estimating a lower-rank approximation of the Hessian using Monte Carlo ensembles

of the variational inversion to represent the prior uncertainties (Chevallier, 2007). However, the amount of time required to run a single inversion is too large at present, preventing the derivation of robust posterior statistics but also the accounting of systematic uncertainties. Recent developments in the CIF (Chevallier et al., 2023; Chevallier, 2013) may help us to significantly reduce our computational costs and run Monte Carlo ensembles. While these new features have not been tested with realistic configurations yet, preliminary results are promising.

The inversion system proposed in this work benefits from the advantages of both 3-D modelling and variational inversion methods, and also includes the optimization of the source isotopic signatures. Additionally, it accounts for the observation operator nonlinearity, which is an important component of isotopic data assimilation, particularly when source signatures are also optimized. To our knowledge, such a system is unique and allowed to reconcile emissions and source signatures with the limitation of still-large random uncertainties in the isotopic signatures. More developments are necessary to improve the robustness of the estimates and the relevance of such a system but we believe that this study represents a significant step towards a better quantification of the $CH_4$ sectorial and regional emissions and of the global $CH_4$ budget.

# Appendix A: $CH_4$, $\delta(^{13}C, CH_4)$ and $\delta(D, CH_4)$ surface in-situ observation sites

**Table A1.** List of $CH_4$ surface in-situ observation sites that provided measurements assimilated in the inversions between 1998 and 2018. AOC, PAO, POC and WPC are mobile stations. Their characteristics are compiled into a single line, providing latitude and longitude ranges of the measurements. Stations that retrieved samples consisting mainly of well-mixed Marine Boundary Layer (MBL) air are indicated in bold.

| Site code | Station name | Country/Territory | Network | Latitude | Longitude | Elevation (m a.s.l.) | Date range (MM/YYYY) |
|---|---|---|---|---|---|---|---|
| **ABP** | Arembepe | Brazil | NOAA | 12.76 °S | 38.16 °W | 6 | 10/2006 - 01/2010 |
| **ALT** | Alert | Canada | NOAA | 82.45 °N | 62.51 °W | 195 | 01/1998 - 12/2018 |
| AMT | Argyle | United States | NOAA | 45.03 °N | 68.68 °W | 157 | 09/2003 - 12/2008 |
| AMY | Anmyeon-do | Republic of Korea | NOAA | 36.54 °N | 126.33 °E | 125 | 12/2013 - 12/2018 |
| AOC | Atlantic Ocean Cruise | N/A | NOAA | 30.30 °S / 35.00 °N | -75.11 °W / 13.57 °E | 22 | 05/2004 - 02/2005 |
| **ASC** | Ascension Island | United Kingdom | NOAA | 7.97 °S | 14.40 °W | 90 | 01/1998 - 12/2018 |
| ASK | Assekrem | Algeria | NOAA | 23.26 °N | 5.63 °E | 2715 | 01/1998 - 12/2018 |
| **AZR** | Terceira Island | Portugal | NOAA | 38.77 °N | 27.38 °W | 24 | 01/1998 - 12/2018 |
| BAL | Baltic Sea | Poland | NOAA | 55.43 °N | 16.95 °E | 28 | 01/1998 - 06/2011 |
| BHD | Baring Head Station | New Zealand | NOAA | 41.41 °S | 174.87 °E | 90 | 10/1999 - 12/2018 |
| BKT | Bukit Kototabang | Indonesia | NOAA | 0.20 °S | 100.32 °E | 875 | 01/2004 - 12/2018 |
| **BME** | St. Davids Head | United Kingdom | NOAA | 32.37 °N | 64.65 °W | 17 | 01/1998 - 01/2010 |
| **BMW** | Tudor Hill | United Kingdom | NOAA | 32.26 °N | 64.88 °W | 60 | 01/1998 - 12/2018 |
| **BRW** | Barrow Atmospheric Baseline Observatory | United States | NOAA | 71.32 °N | 156.60 °W | 13 | 01/1998 - 12/2018 |
| BSC | Black Sea | Romania | NOAA | 44.18 °N | 28.66 °E | 5 | 01/1998 - 12/2011 |
| **CBA** | Cold Bay | United States | NOAA | 55.20 °N | 162.72 °W | 25 | 01/1998 - 12/2018 |
| **CGO** | Cape Grim | Australia | NOAA | 40.68 °S | 144.68 °E | 164 | 01/1998 - 12/2018 |
| **CHR** | Christmas Island | Republic of Kiribati | NOAA | 1.70 °N | 157.15 °W | 5 | 11/1998 - 12/2018 |
| CIB | Centro de Investigacion de la Baja Atmosfera (CIBA) | Spain | NOAA | 41.81 °N | 4.93 °W | 850 | 05/2009 - 12/2018 |
| CMO | Cape Meares | United States | NOAA | 45.48 °N | 123.97 °W | 35 | 03/1998 - 03/1998 |
| CPT | Cape Point | South Africa | NOAA | 34.35 °S | 18.49 °E | 260 | 02/2010 - 12/2018 |
| **CRZ** | Crozet Island | France | NOAA | 46.43 °S | 51.85 °E | 202 | 01/1998 - 11/2018 |
| DRP | Drake Passage | nan | NOAA | 57.65 °S | 64.18 °W | 10 | 04/2003 - 12/2018 |

**Table A1.** Following Table A1

| Site code | Station name | Country/Territory | Network | Latitude | Longitude | Elevation (m a.s.l.) | Date range (MM/YYYY) |
|---|---|---|---|---|---|---|---|
| DSI | Dongsha Island | Taiwan | NOAA | 20.70 °N | 116.73 °E | 8 | 03/2010 - 12/2018 |
| EIC | Easter Island | Chile | NOAA | 27.15 °S | 109.45 °W | 55 | 01/1998 - 12/2018 |
| **GMI** | Mariana Islands | Guam | NOAA | 13.39 °N | 144.66 °E | 6 | 01/1998 - 12/2018 |
| GOZ | Dwejra Point | Malta | NOAA | 36.05 °N | 14.89 °E | 6 | 01/1998 - 02/1999 |
| **HBA** | Halley Station | United Kingdom | NOAA | 75.61 °S | 26.21 °W | 35 | 01/1998 - 02/2018 |
| HPB | Hohenpeissenberg | Germany | NOAA | 47.80 °N | 11.02 °E | 990 | 04/2006 - 12/2018 |
| HSU | Humboldt State University | United States | NOAA | 41.05 °N | 124.73 °W | 7 | 05/2008 - 05/2017 |
| HUN | Hegyhatsal | Hungary | NOAA | 46.95 °N | 16.65 °E | 344 | 01/1998 - 12/2018 |
| **ICE** | Storhofdi | Iceland | NOAA | 63.40 °N | 20.29 °W | 127 | 01/1998 - 12/2018 |
| ITN | Grifton | United States | NOAA | 35.37 °N | 77.39 °W | 505 | 01/1998 - 06/1999 |
| IZO | Izana | Spain | NOAA | 28.30 °N | 16.48 °W | 2377 | 01/1998 - 12/2018 |
| KCO | Kaashidhoo | Republic of Maldives | NOAA | 4.97 °N | 73.47 °E | 6 | 03/1998 - 07/1999 |
| **KEY** | Key Biscayne | United States | NOAA | 25.67 °N | 80.20 °W | 6 | 01/1998 - 12/2018 |
| **KUM** | Cape Kumukahi | United States | NOAA | 19.52 °N | 154.82 °W | 8 | 01/1998 - 12/2018 |
| KZD | Sary Taukum | Kazakhstan | NOAA | 44.45 °N | 75.57 °E | 412 | 01/1998 - 08/2009 |
| KZM | Plateau Assy | Kazakhstan | NOAA | 43.25 °N | 77.88 °E | 2524 | 01/1998 - 08/2009 |
| LEF | Park Falls | United States | NOAA | 45.93 °N | 90.27 °W | 868 | 01/1998 - 12/2018 |
| LLB | Lac La Biche | Canada | NOAA | 54.95 °N | 112.45 °W | 546 | 01/2008 - 02/2013 |
| LLN | Lulin | Taiwan | NOAA | 23.46 °N | 120.86 °E | 2867 | 08/2006 - 12/2018 |
| LMP | Lampedusa | Italy | NOAA | 35.51 °N | 12.61 °E | 50 | 10/2006 - 12/2018 |
| MEX | High Altitude Global Climate Observation Center | Mexico | NOAA | 18.98 °N | 97.31 °W | 4469 | 01/2009 - 12/2018 |
| **MHD** | Mace Head | Ireland | NOAA | 53.33 °N | 9.90 °W | 26 | 01/1998 - 12/2018 |
| **MID** | Sand Island | United States | NOAA | 28.22 °N | 177.37 °W | 8 | 01/1998 - 12/2018 |
| MKN | Mt. Kenya | Kenya | NOAA | 0.06 °S | 37.30 °E | 3649 | 12/2003 - 06/2011 |
| MLO | Mauna Loa | United States | NOAA | 19.53 °N | 155.58 °W | 3437 | 01/1998 - 12/2018 |
| NAT | Farol De Mae Luiza Lighthouse | Brazil | NOAA | 5.51 °S | 35.26 °W | 20 | 09/2010 - 12/2018 |
| NMB | Gobabeb | Namibia | NOAA | 23.58 °S | 15.03 °E | 461 | 07/1998 - 12/2018 |
| NWR | Niwot Ridge | United States | NOAA | 40.05 °N | 105.58 °W | 3526 | 01/1998 - 12/2018 |
| OXK | Ochsenkopf | Germany | NOAA | 50.03 °N | 11.81 °E | 1185 | 03/2003 - 12/2018 |
| PAL | Pallas-Sammaltunturi | Finland | NOAA | 67.97 °N | 24.12 °E | 570 | 12/2001 - 12/2018 |

**Table A1.** Following Table A1

| Site code | Station name | Country/Territory | Network | Latitude | Longitude | Elevation (m a.s.l.) | Date range (MM/YYYY) |
|---|---|---|---|---|---|---|---|
| PAO | Pacific-Atlantic Ocean | N/A | NOAA | 30.20 °S<br>67.86 °N | 164.58 °W<br>9.93 °W | 10 | 03/2006 - 10/2006 |
| **POC** | Pacific Ocean | N/A | NOAA | 36.67 °S<br>35.07 °N | 180.00 °W<br>179.83 °E | 20 | 04/1998 - 07/2017 |
| **PSA** | Palmer Station | United States | NOAA | 64.92 °S | 64.00 °W | 15 | 01/1998 - 12/2018 |
| PTA | Point Arena | United States | NOAA | 38.95 °N | 123.73 °W | 22 | 01/1999 - 05/2011 |
| **RPB** | Ragged Point | Barbados | NOAA | 13.16 °N | 59.43 °W | 20 | 01/1998 - 12/2018 |
| SDZ | Shangdianzi | Peoples Republic of China | NOAA | 40.65 °N | 117.12 °E | 298 | 09/2009 - 09/2015 |
| SEY | Mahe Island | Seychelles | NOAA | 4.68 °S | 55.53 °E | 7 | 01/1998 - 12/2018 |
| SGP | Southern Great Plains | United States | NOAA | 36.62 °N | 97.48 °W | 374 | 04/2002 - 12/2018 |
| **SHM** | Shemya Island | United States | NOAA | 52.72 °N | 174.10 °E | 28 | 01/1998 - 10/2018 |
| **SMO** | Tutuila | American Samoa | NOAA | 14.25 °S | 170.57 °W | 47 | 01/1998 - 12/2018 |
| **SPO** | South Pole | United States | NOAA | 89.98 °S | 24.80 °W | 2821 | 01/1998 - 12/2018 |
| **STM** | Ocean Station M | Norway | NOAA | 66.00 °N | 2.00 °E | 7 | 01/1998 - 11/2009 |
| SUM | Summit | Greenland | NOAA | 72.60 °N | 38.42 °W | 3214 | 01/1998 - 12/2018 |
| **SYO** | Syowa Station | Japan | NOAA | 69.00 °S | 39.58 °E | 16 | 01/1998 - 12/2018 |
| TAC | Tacolneston | United Kingdom | NOAA | 52.52 °N | 1.14 °E | 236 | 06/2014 - 01/2016 |
| TAP | Tae-ahn Peninsula | Republic of Korea | NOAA | 36.73 °N | 126.13 °E | 21 | 01/1998 - 12/2018 |
| THD | Trinidad Head | United States | NOAA | 41.05 °N | 124.15 °W | 112 | 04/2002 - 06/2017 |
| TIK | Hydrometeorological Observatory of Tiksi | Russia | NOAA | 71.60 °N | 128.89 °E | 29 | 08/2011 - 09/2018 |
| USH | Ushuaia | Argentina | NOAA | 54.85 °S | 68.31 °W | 32 | 01/1998 - 12/2018 |
| UTA | Wendover | United States | NOAA | 39.90 °N | 113.72 °W | 1332 | 01/1998 - 12/2018 |
| UUM | Ulaan Uul | Mongolia | NOAA | 44.45 °N | 111.10 °E | 1012 | 01/1998 - 12/2018 |
| WIS | Weizmann Institute of Science at the Arava Institute | Israel | NOAA | 30.86 °N | 34.78 °E | 482 | 01/1998 - 12/2018 |
| WKT | Moody | United States | NOAA | 31.32 °N | 97.33 °W | 708 | 02/2001 - 10/2010 |
| WLG | Mt. Waliguan | Peoples Republic of China | NOAA | 36.27 °N | 100.92 °E | 3815 | 01/1998 - 12/2018 |
| WPC | Western Pacific Cruise | N/A | NOAA | 30.67 °S<br>32.46 °N | 135.55 °E<br>170.47 °E | 8 | 05/2004 - 06/2013 |
| **ZEP** | Ny-Alesund | Norway and Sweden | NOAA | 78.91 °N | 11.89 °E | 479 | 01/1998 - 12/2018 |

**Table A2.** List of $\delta(^{13}C, CH_4)$ surface in-situ observation sites that provided measurements assimilated in the inversions between 1998 and 2018. WPC is a mobile station. Its characteristics are compiled into a single line, providing latitude and longitude ranges of the measurements. Stations that retrieved samples consisting mainly of well-mixed MBL air are indicated in bold.

| Site code | Station name | Country/Territory | Network | Latitude | Longitude | Elevation (m a.s.l.) | Date range (MM/YYYY) |
|---|---|---|---|---|---|---|---|
| **ALT** | Alert | Canada | NOAA | 82.45 °N | 62.51 °W | 195 | 08/2000 - 12/2017 |
| AMY | Anmyeon-do | Republic of Korea | NOAA | 36.54 °N | 126.33 °E | 125 | 12/2013 - 12/2017 |
| **ASC** | Ascension Island | United Kingdom | NOAA | 7.97 °S | 14.40 °W | 90 | 10/2000 - 12/2017 |
| **AZR** | Terceira Island | Portugal | NOAA | 38.75 °N | 27.08 °W | 24 | 08/2000 - 12/2017 |
| BAL | Baltic Sea | Poland | NOAA | 55.35 °N | 17.22 °E | 28 | 04/2008 - 06/2011 |
| BHD | Baring Head Station | New Zealand | NOAA | 41.41 °S | 174.87 °E | 90 | 03/2009 - 11/2017 |
| **BRW** | Barrow Atmospheric Baseline Observatory | United States | NOAA | 71.32 °N | 156.60 °W | 16 | 01/1998 - 12/2017 |
| **CBA** | Cold Bay | United States | NOAA | 55.20 °N | 162.72 °W | 25 | 08/2000 - 12/2017 |
| **CGO** | Cape Grim | Australia | NOAA | 40.68 °S | 144.68 °E | 164 | 01/1998 - 12/2017 |
| **KUM** | Cape Kumukahi | United States | NOAA | 19.52 °N | 154.82 °W | 3 | 01/1999 - 12/2017 |
| LLB | Lac La Biche | Canada | NOAA | 54.95 °N | 112.45 °W | 546 | 01/2008 - 02/2013 |
| MEX | High Altitude Global Climate Observation Center | Mexico | NOAA | 18.98 °N | 97.31 °W | 4469 | 01/2009 - 12/2017 |
| **MHD** | Mace Head | Ireland | NOAA | 53.33 °N | 9.90 °W | 26 | 01/1999 - 12/2017 |
| MLO | Mauna Loa | United States | NOAA | 19.53 °N | 155.58 °W | 3402 | 01/1998 - 12/2017 |
| NWR | Niwot Ridge | United States | NOAA | 40.05 °N | 105.58 °W | 3526 | 01/1998 - 12/2017 |
| **SMO** | Tutuila | American Samoa | NOAA | 14.25 °S | 170.57 °W | 47 | 01/1998 - 12/2017 |
| **SPO** | South Pole | United States | NOAA | 89.98 °S | 24.80 °W | 2815 | 01/1998 - 12/2017 |
| SUM | Summit | Greenland | NOAA | 72.60 °N | 38.42 °W | 3214 | 04/2010 - 12/2017 |
| TAP | Tae-ahn Peninsula | Republic of Korea | NOAA | 36.73 °N | 126.13 °E | 21 | 09/2000 - 12/2017 |
| WLG | Mt. Waliguan | Peoples Republic of China | NOAA | 36.27 °N | 100.92 °E | 3815 | 07/2001 - 12/2017 |
| WPC | Western Pacific Cruise | N/A | NOAA | 30.67 °S  32.46 °N | 135.55 °E  170.47 °E | 10 | 11/2005 - 06/2013 |
| **ZEP** | Ny-Alesund | Norway and Sweden | NOAA | 78.91 °N | 11.89 °E | 479 | 10/2001 - 12/2017 |

**Table A3.** List of $\delta(D, CH_4)$ surface in-situ observation sites that provided measurements assimilated in the inversion INV_DD between 2005 and 2010. Stations that retrieved samples consisting mainly of well-mixed MBL air are indicated in bold.

| Site code | Station name | Country/Territory | Network | Latitude | Longitude | Elevation (m a.s.l.) | Date range (MM/YYYY) |
|---|---|---|---|---|---|---|---|
| **ALT** | Alert | Canada | NOAA | 82.45 °N | 62.51 °W | 205 | 04/2005 - 12/2009 |
| **ASC** | Ascension Island | United Kingdom | NOAA | 7.97 °S | 14.40 °W | 90 | 04/2005 - 03/2010 |
| **AZR** | Terceira Island | Portugal | NOAA | 38.76 °N | 27.36 °W | 24 | 02/2005 - 10/2009 |
| BAL | Baltic Sea | Poland | NOAA | 55.41 °N | 17.06 °E | 28 | 10/2004 - 02/2010 |
| **BRW** | Barrow Atmospheric Baseline Observatory | United States | NOAA | 71.31 °N | 156.58 °W | 27 | 04/2005 - 03/2010 |
| BSC | Black Sea | Romania | NOAA | 44.18 °N | 28.66 °E | 5 | 03/2005 - 03/2008 |
| **CBA** | Cold Bay | United States | NOAA | 55.20 °N | 162.71 °W | 25 | 05/2005 - 03/2010 |
| **CGO** | Cape Grim | Australia | NOAA | 40.66 °S | 144.66 °E | 164 | 01/2005 - 07/2009 |
| **KUM** | Cape Kumukahi | United States | NOAA | 19.51 °N | 154.81 °W | 8 | 05/2005 - 03/2010 |
| LEF | Park Falls | United States | NOAA | 45.91 °N | 90.26 °W | 868 | 04/2005 - 05/2008 |
| **MHD** | Mace Head | Ireland | NOAA | 53.31 °N | 9.90 °W | 26 | 03/2005 - 08/2009 |
| MLO | Mauna Loa | United States | NOAA | 19.53 °N | 155.56 °W | 3437 | 04/2005 - 11/2009 |
| NWR | Niwot Ridge | United States | NOAA | 40.03 °N | 105.56 °W | 3526 | 05/2005 - 01/2010 |
| **SMO** | Tutuila | American Samoa | NOAA | 14.23 °S | 170.56 °W | 47 | 03/2005 - 09/2009 |
| **SPO** | South Pole | United States | NOAA | 89.96 °S | 24.80 °W | 2815 | 02/2005 - 01/2010 |

 **Appendix B: Additional results**

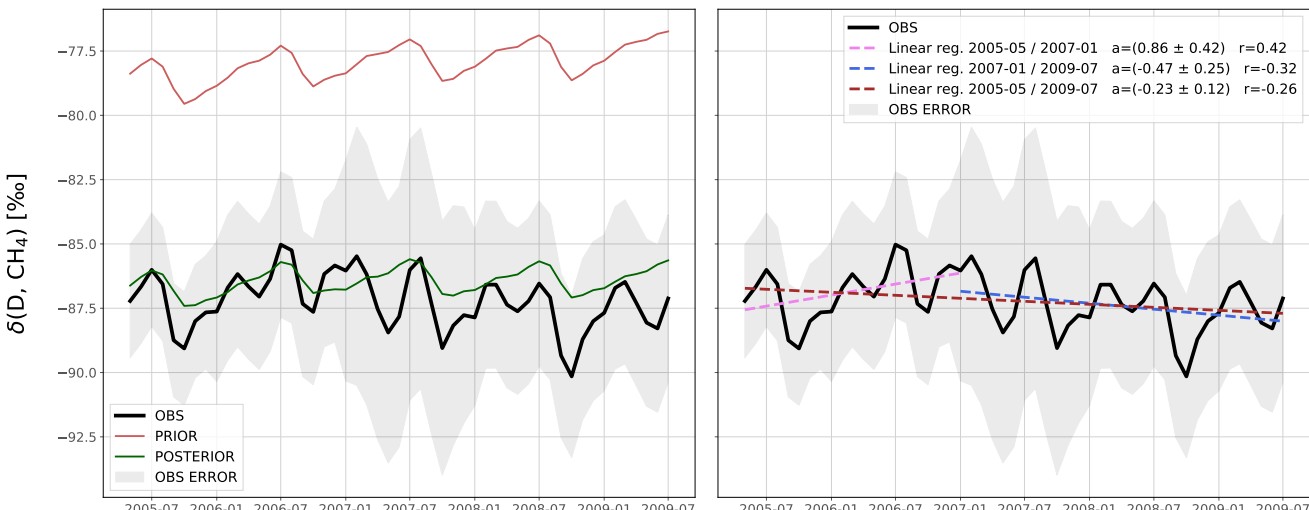

**Figure B1.** The left panel shows a comparison between $\delta(D, CH_4)$ observations, prior and posterior simulations. The right panel shows linear regressions applied on the monthly- and globally-averaged $\delta(D, CH_4)$ observations. We performed three linear regressions: 1) one over the full data period 2005-05 to 2009-07 (brown line), 2) one over the period 2005-05 to 2007-01 (violet line) and one over the period 2007-01 to 2009-07 (blue line). For each linear regression, the coefficient ($a$), its standard error, and the Pearson's correlation coefficient ($r$) are displayed in the legend. Note that the x-axis stops before 2010, as we have only selected months with sufficient data for the average to be representative of the whole globe.

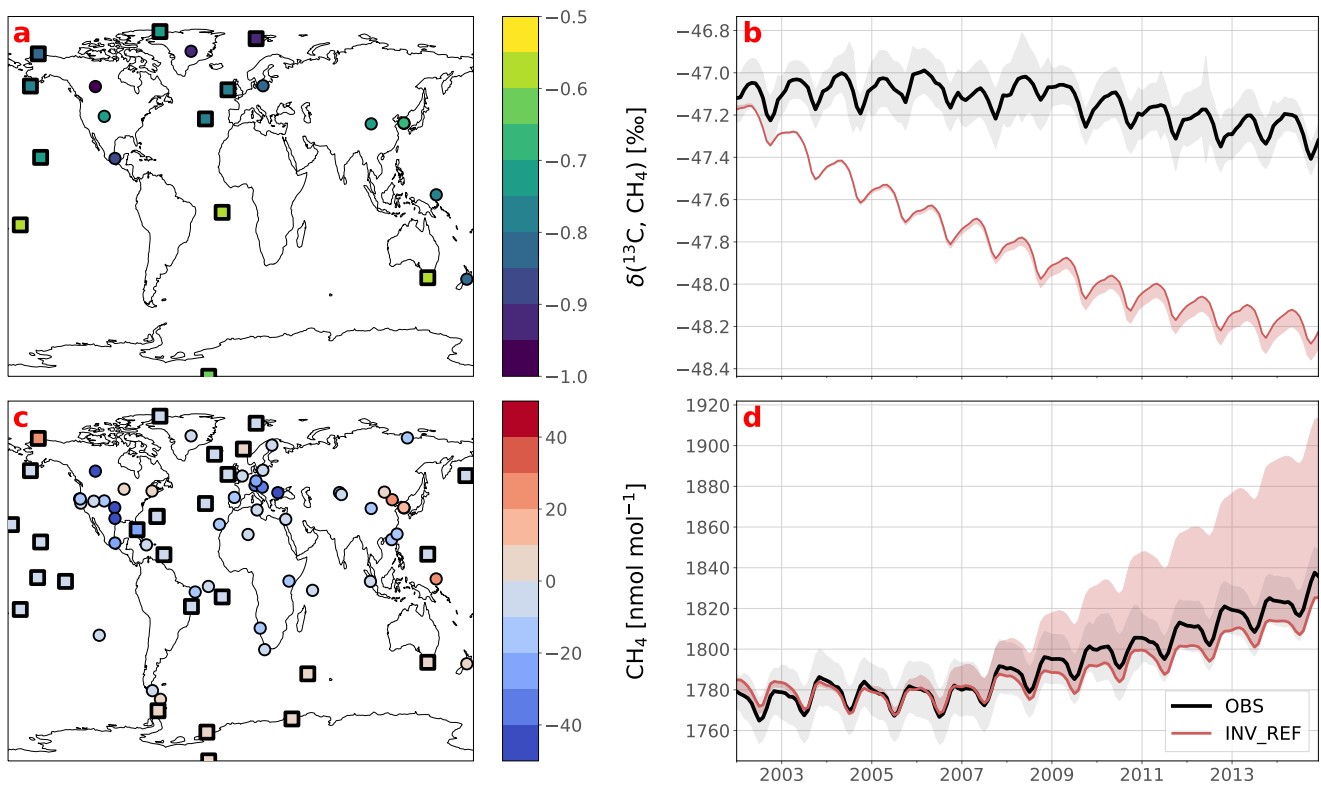

**Figure B2.** Same as Figure 4 but with prior data. Note that the scale for panel c) has been modified and is not centered on zero any-more because prior agreement with $\delta(^{13}C, CH_4)$ data is too low. The large red shaded area in panel d) is caused by a change in OH sink (INV_TURNER).

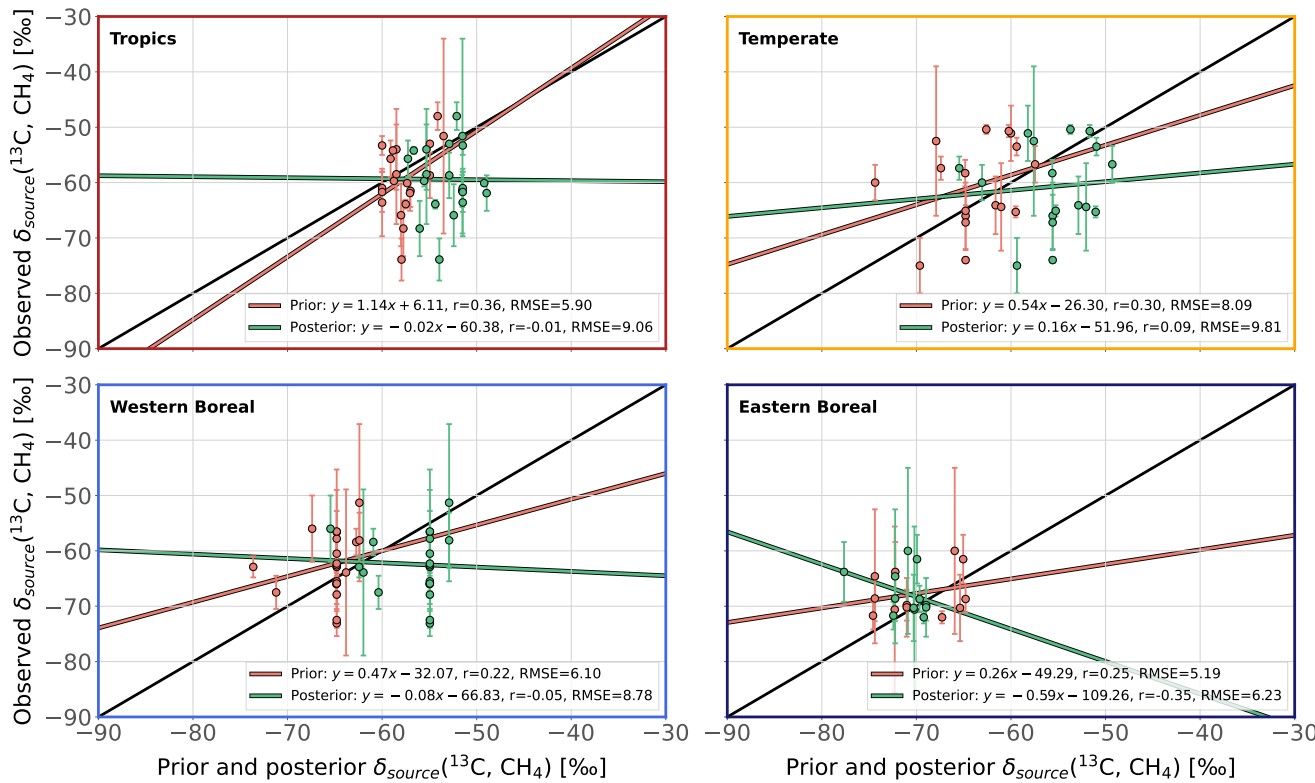

**Figure B3.** Comparison between prior-posterior and observed wetlands $\delta_{\text{source}}(^{13}\text{C},\text{CH}_4)$. Observations are taken from the Supplementary Data 1 provided by Oh et al. (2022). For each observation, prior and posterior values are sampled using the gridcell corresponding to the latitude and longitude provided in the dataset. Error bars for each observation point represent the observation uncertainty. Each panel shows a comparison with observations located in a selected region: Tropics (<30 °N/S), Temperate (30–50 °N/S), Western Boreal (50–90 °N and <15 °W) and Eastern Boreal (50–90 °N and >15 °W). For each panel, the identity line and two linear fitting lines (prior in green and posterior in red) are displayed. The parameters of the fitting lines, the Pearson's correlation coefficients ($r$) and the root-mean-square error (RMSE) are given in the legend box.

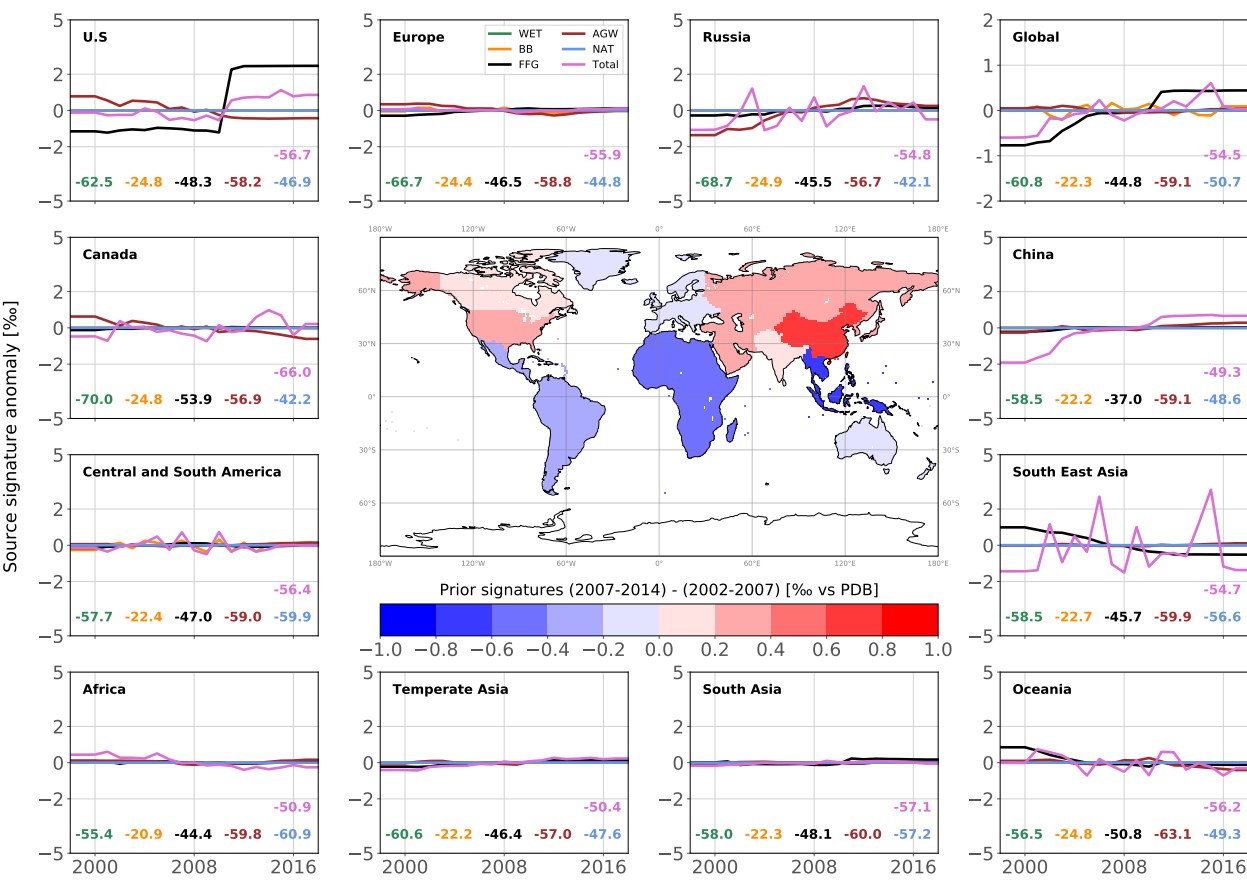

**Figure B4.** Same as Figure 7 but for prior source signatures $\delta_{\text{source}}(^{13}\text{C}, \text{CH}_4)$. For each panel, time-series are anomalies around a 2002-2014 mean value. Units of variations and means are ‰. Note that x-axis ranges from 1998 to 2018 to illustrate the effects of the spin-up and spin-down mentioned in Sect. 2.8. Also, note that the regions used here are slightly different from the regions selected for the optimization.

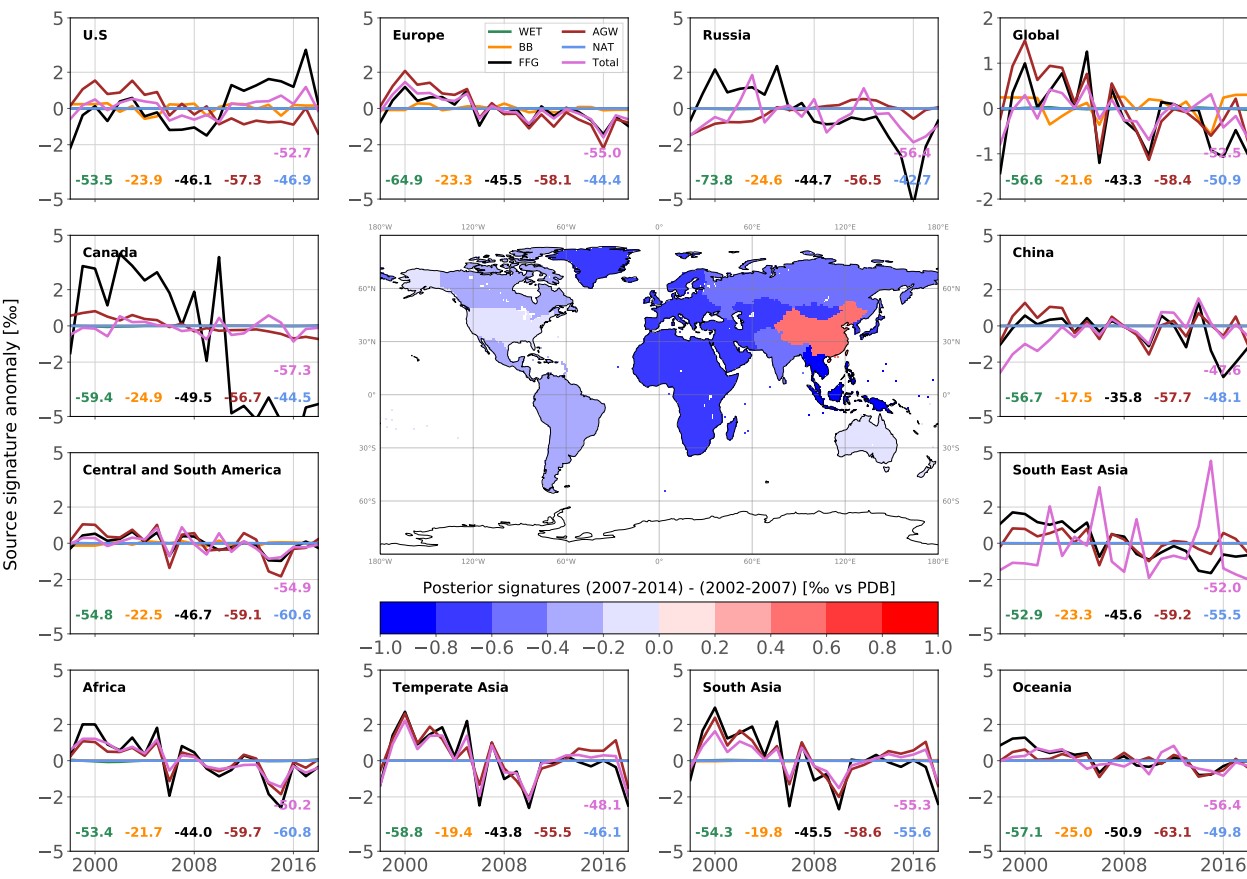

**Figure B5.** Same as Figure B4 but for posterior source signatures $\delta_{\mathrm{source}}(^{13}\mathrm{C},\mathrm{CH}_4)$ from INV_REF. Note that the green (WET) and blue (NAT) lines are flat because 1) prior signatures are constant over time, 2) these categories do not result from the aggregation of multiple subcategories and 3) we optimize only one scaling factor per region for the entire period. Therefore, these values do not vary with time. Also, note that BB source signatures vary only because the regions used here are slightly different from the regions selected for the optimization. Therefore, the flux-weighted average produces some temporal variability.

**Table B1.** Posterior $CH_4$ emissions for the globe and three different latitudinal bands averaged over 2002-2014 for all inversions and all categories. Unit is $\text{Tg a}^{-1}$.

| Global | | | | | | |
|---|---|---|---|---|---|---|
| | Total | AGW | FFG | WET | BB | NAT |
| INV_REF | 589.9 | 220.7 | 124.6 | 192.2 | 29.4 | 23.1 |
| INV_CH4 | 589.6 | 221.4 | 123.6 | 192.4 | 29.1 | 23.1 |
| INV_DD | 590.4 | 220.8 | 124.8 | 192.3 | 29.4 | 23.1 |
| INV_LOCKED | 590.6 | 205.6 | 155.1 | 165.3 | 41.4 | 23.1 |
| INV_FLATOH | 575.2 | 216.6 | 120.7 | 186.1 | 28.7 | 23.1 |
| INV_TURNER | 561.1 | 212.5 | 116.8 | 180.6 | 28.2 | 23.1 |

| Northern high-latitudes (60 °N - 90 °N) | | | | | | |
|---|---|---|---|---|---|---|
| | Total | AGW | FFG | WET | BB | NAT |
| INV_REF | 28.5 | 1.4 | 7.9 | 15.5 | 0.9 | 2.8 |
| INV_CH4 | 27.8 | 1.4 | 7.4 | 15.3 | 0.8 | 2.8 |
| INV_DD | 28.6 | 1.4 | 8.0 | 15.5 | 0.9 | 2.8 |
| INV_LOCKED | 26.8 | 1.4 | 8.0 | 13.3 | 1.2 | 2.8 |
| INV_FLATOH | 28.2 | 1.4 | 7.8 | 15.3 | 0.8 | 2.8 |
| INV_TURNER | 28.0 | 1.4 | 7.8 | 15.2 | 0.8 | 2.8 |

| Northern mid-latitudes (30 °N - 60 °N) | | | | | | |
|---|---|---|---|---|---|---|
| | Total | AGW | FFG | WET | BB | NAT |
| INV_REF | 208.3 | 82.7 | 65.6 | 48.9 | 7.2 | 3.8 |
| INV_CH4 | 206.2 | 82.7 | 64.8 | 47.7 | 7.1 | 3.8 |
| INV_DD | 208.6 | 82.7 | 65.8 | 49.0 | 7.2 | 3.8 |
| INV_LOCKED | 214.4 | 78.4 | 87.0 | 36.7 | 8.5 | 3.9 |
| INV_FLATOH | 203.4 | 81.4 | 63.1 | 47.9 | 7.1 | 3.8 |
| INV_TURNER | 198.6 | 80.2 | 60.3 | 47.1 | 7.1 | 3.8 |

| Tropics (90 °S - 30 °N) | | | | | | |
|---|---|---|---|---|---|---|
| | Total | AGW | FFG | WET | BB | NAT |
| INV_REF | 353.2 | 136.6 | 51.0 | 127.8 | 21.4 | 16.5 |
| INV_CH4 | 355.7 | 137.3 | 51.4 | 129.4 | 21.1 | 16.5 |
| INV_DD | 353.2 | 136.6 | 51.0 | 127.8 | 21.4 | 16.5 |
| INV_LOCKED | 349.5 | 125.8 | 60.2 | 115.3 | 31.7 | 16.5 |
| INV_FLATOH | 343.5 | 133.7 | 49.8 | 122.9 | 20.7 | 16.5 |
| INV_TURNER | 334.5 | 130.8 | 48.6 | 118.3 | 20.3 | 16.4 |

*Data availability.* The code files of the CIF version used in the present paper are registered under the following DOI: doi.org/10.5281/zenodo.6304912 (Berchet et al., 2022). The $CH_4$ (Lan et al., 2022), $\delta(^{13}C,CH_4)$ (White et al., 2021) and $\delta(D,CH_4)$ (White et al., 2016) observational data can be downloaded directly from the NOAA-GML website (esrl.noaa.gov/gmd/aftp/data/trace_gases, last access: 12 July 2021). All the other relevant data used to perform the inversions is registered under the following DOI: doi.org/10.5281/zenodo.10390430 (Thanwerdas, 2023).

*Author contributions.* JT designed and ran the inversions and performed the data analysis presented in this paper. MS, AB, IP, and PB provided scientific, technical expertise and contributed to the scientific analysis of this work. JT prepared the paper, with contributions from all co-authors.

*Competing interests.* The authors declare that they have no conflict of interest.

*Acknowledgements.* This work has been supported by the CEA (Commissariat à l'Energie Atomique et aux Energies Alternatives). The study extensively relies on the meteorological data provided by the ECMWF. Calculations were performed using the computing resources of LSCE, which are maintained by Julien Bruna, François Marabelle and the rest of the LSCE IT team. The authors wish to thank the measurement teams from the NOAA GML and from INSTAAR for their continuous and high-quality work. In particular, the authors would like to express special thanks to Xin Lan from NOAA and to Sylvia E. Michel, Bruce H. Vaughn and Reid Clark from INSTAAR for their invaluable help and helpful comments, which greatly improved the quality of the submitted draft. Finally, the authors are grateful to the two anonymous referees and the editor for their invaluable insights, which greatly enhanced the quality of the paper.

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
