# Peer review of "Investigation of the post-2007 methane renewed growth with high-resolution 3-D variational inverse modelling and isotopic constraints"

_EGUsphere, 2023_

## Author Comment (AC1)

**Response to Referees' Comments**

**Joël Thanwerdas[1,a*], Marielle Saunois[1], Antoine Berchet[1], Isabelle Pison[1], and Philippe Bousquet[1]**

[1]Laboratoire des Sciences du Climat et de l'Environnement, CEA-CNRS-UVSQ, IPSL, Gif-sur-Yvette, France.

[a]now at: Empa, Swiss Federal Laboratories for Materials Science and Technology, Dübendorf, Switzerland.

We thank the two referees for their invaluable insights, which have greatly enhanced the quality of the paper. We provide here a comprehensive response to the comments received. Referee#1's comments are in red and Referee#2's comments are in blue. For each comment, an answer is provided in normal text, *citations from the text are in italic* and **the modifications from the new version of the manuscript are provided in bold and small text**. Note that modifications have been included only when deemed substantial enough.

Attached to this response, we also provide the new version of the manuscript and a track-changes document.

**General comments**

The time dimension of the isotopic signatures in the control vector is not clear. The results suggest the INV_REF optimizes these signatures per year. The justification for doing this is missing. The discussion section mentions that an intermediate solution (between fixed and full optimization of signatures) may be preferable. To me it would make sense to optimize time domain averaged signatures only, potentially with a trend for processes for which this might be relevant. I wonder how variable to year-to-year signature adjustments are, which is not shown. The risk of the current approach – if I understand it correctly - is that uncertainties of annual signatures strongly covary with the uncertainties of emissions, and therefore emission IAV ends up as IAV in isotopic signatures.

Section 2.4 is expected to offer the information that Referee #2 is seeking.

In the first paragraph, we explain the rationale behind optimizing only one scaling factor per continental region.

The second paragraph details our approach. For each continental region, we optimize a scaling factor per year for FF and AGW, and a single scaling factor for the entire period (1998-2018) for WET, BB, and NAT.

The FF and AGW categories are the result of the aggregation of multiple sub-categories whose emissions can fluctuate over time within optimized regions. Therefore, the isotopic signatures of these two categories might change over time within each continental region due to variations in the source mixture. For instance, US coal production decreased between 2000 and 2018, while natural gas and crude oil production increased. The methane emissions stemming from these different types of production have distinct signatures according to the literature. Consequently, the FF isotopic signature in this region is likely to change, even if the individual signatures (for oil, natural gas, and coal) remain unchanged. A similar concept applies to AGW due to changes in C3/C4 diets.

For the other categories, we lack sufficient data to support changes in source signatures. Therefore, we prefer to optimize a single scaling factor for the entire period.

We could have opted to optimize scaling factors for FF and AGW at longer intervals, such as every 3, 5, or 10 years. However, for our initial testing, examining year-to-year variations seemed more insightful.

We believe that it is primarily the Interannual Variability (IAV) of the source mixture that could manifest as IAV in the isotopic signatures. Rather than altering the IAV of the source mixture in a specific region, the system may choose to modify the isotopic signatures, as is likely the case in our inversions. To mitigate this "target shift," it is crucial to obtain a more accurate estimate of the uncertainties in source signatures for different categories. Although the methodology for determining these uncertainties can certainly be improved, we believe that optimizing only a single scaling factor for the entire period for categories like FF and AGW is not a suitable approach. This approach sidesteps a significant aspect of the problem, which is that source signatures for these categories (and their uncertainties) can indeed change over time.

To make the information about the temporal optimization of source signatures clearer, we included additional columns in Table 2. We also added two figures in Appendix B (Figure B3 and Figure B4) showing the temporal variations of prior and posterior source signatures and introduced the figures in Sect 3.6.

[Figure]

**Figure B3.** Same as Figure 7 but for prior signatures. For each panel, time-series are anomalies around a 2002-2014 mean value. Units of variations and means are ‰. Note that x-axis ranges from 1998 to 2018 to illustrate the effects of the spin-up and spin-down mentioned in Sect. 2.8. Also, note that the regions used here are slightly different from the regions selected for the optimization.

[Figure]

**Figure B4.** Same as Figure B3 but for posterior source signatures from INV_REF. Note that the green (WET) and blue (NAT) lines are flat because 1) prior signatures are constant over time, 2) these categories do not result from the aggregation of multiple sub-categories and 3) we optimize only one scaling factor per region for the entire period. Therefore, these values do not vary with time. Also, note that BB source signatures slightly vary only because the regions used here are slightly different from the regions selected for the optimization. Therefore, the flux-weighted average produces some temporal variability.

The most promising added value of dD measurements is to facilitate the independent estimation of methane sources and sinks. However, the current setup only supports the optimization of emissions. Sensitivity tests are done to evaluate the use of different OH scenarios, but it is not clear which of them is most consistent with the dD data. Without in depth analysis of the benefit of dD for estimating methane sinks I do not think it is justified to conclude that the usefulness of dD measurements is limited.

In the OH scenarios, unfortunately, only $^{12}CH_4$ and $^{13}CH_4$ are transported and simulations cannot be compared to $\delta(D, CH_4)$ measurements. We did not try to optimize sinks because optimizing source signatures already adds a large number of degrees of freedom to the system.

In addition, we do not understand the statement: "The most promising added value of dD measurements is to facilitate the independent estimation of methane sources and sinks". We would appreciate some reference supporting it. Reactions with OH, O($^1$D) and Cl have fractionation coefficients that depend on the isotope so it might help us to gain some information about the sink mixture. However, we do not believe the methane sources and sinks can be **independently** optimized with the help of $\delta(D, CH_4)$ measurements because both emissions and sinks affect the $\delta(D, CH_4)$ isotopic composition in the atmosphere. Both estimates would be strongly correlated if only isotopic data were used. Without a good estimate of $\delta(D, CH_4)$ source signatures, we cannot precisely estimate the effect of the sources on the isotopic composition and hence, the effect of the sinks.

Furthermore, if the Referee is referring to line 584 in Section 3.7, it is important to note that our objective is not to diminish the value of δ(D, CH$_4$) measurements in a general context. Instead, we wish to emphasize that assimilating these measurements yields only marginal changes within our specific setups, while substantially increasing computational costs. We have slightly changed this paragraph.

**Finally, assimilating δ(D,CH4) observations and optimizing δ$_{source}$(D,CH$_4$) source signatures in INV_DD have a very small influence on our posterior emission estimates, as indicated in Table 5. The most significant difference observed is a small positive shift of+0.5 ‰ in the BB posterior source signature compared to INV_REF. Consequently, with our setups, assimilating δ(D,CH4) does not appear to provide any substantial additional constraint on the CH$_4$ budget estimate. Several factors may contribute to this result: 1) the existing network provides comparatively fewer δ(D,CH$_4$) observations in comparison to δ(13C,CH4) observations, 2) δ(D,CH4) observations spans only the period from 2005 to 2010 and therefore, the full run cannot be fully constrained by this data and 3) the constraints may be too weak due to an overestimation of the prescribed uncertainties in δ$_{source}$(D,CH$_4$) sources signatures. As including δ(D,CH$_4$) in the inversion doubles the computational cost compared to a setup like INV_REF, we recommend not assimilating δ(D,CH$_4$) in our system until either the computational cost can be reduced, more observations become available or lower uncertainties are established.**

Some further discussion is needed of the absence of posterior uncertainties in the current setup. Strictly speaking, without posterior uncertainties the statistical significance of the emission adjustments that are found cannot be judged. I can accept an argument that the current system is not prepared to generate them yet, although I do consider it a major shortcoming. For this reason, some discussion is needed of options and future plans to address this important methodological limitation of the system that is presented.

This question is specifically addressed in the GMD paper presenting the inversion system (Thanwerdas et al., 2022), section 3.5. It is also addressed, to a lesser extent, in the conclusion:

*Running a large ensemble of sensitivity experiments is therefore too costly, preventing the accounting of systematic uncertainties and the derivation of robust posterior statistics. Future developments will focus on implementing parallelization methods in order to drastically reduce the computational burden without affecting the results.*

Formally, posterior uncertainties are given by the Hessian of the cost function (Meirink et al. 2008). This matrix can hardly be computed at an achievable cost, considering the size of the inverse problem. Other means must be implemented to obtain the posterior uncertainty, such as estimating a lower-rank approximation of the Hessian, using Monte Carlo ensembles of the variational inversion to represent the prior uncertainties (Chevallier et al., 2007), or computing multiple configurations covering a given range of possibilities. As explained in the conclusion, the reduction of the computational cost and the use of parallelization methods are the two most promising approaches. At present, the computational cost have been largely reduced following a GPU porting of the LMDz code (Chevallier et al., 2023). In addition, it is now possible to split the full assimilation time window into shorter windows so the total runtime can be reduced. These new features are yet to be tested with realistic setups but preliminary results are promising.

In addition, M1QN3 is not the only optimization algorithm that can be utilized to perform variational inversions in the CIF. The CONGRAD algorithm (Fisher, 1998), which follows a conjugate gradient method combined with a Lanczos algorithm, is also implemented. In particular, it facilitates the computation of posterior uncertainties considerably. However, CONGRAD has not yet been tested with $\delta(^{13}C,CH4)$ data and is only designed for linear problems. Therefore, optimization of isotopic signatures would not be possible. Furthermore, we recently noticed that CONGRAD was less efficient than M1QN3 in minimizing the cost function. While exploring alternative algorithms is a possible solution, we do not believe it to be the most optimal course of action to pursue.

Some additional information has been added to the conclusion.

**Also, the main limitation of our inversion system is the associated computational cost and the absence of posterior uncertainties. Formally, posterior uncertainties are given by the Hessian of the cost function (Meirink et al., 2008). This matrix can hardly be computed at an achievable cost, considering the size of the inverse problem. Other means must be implemented to obtain the posterior uncertainty, such as estimating a lower-rank approximation of the Hessian using Monte Carlo ensembles of the variational inversion to represent the prior uncertainties (Chevallier, 2007). However, the amount of time required to run a single inversion is too large at present, preventing the derivation of robust posterior statistics but also the accounting of systematic uncertainties. Recent developments in the CIF (Chevallier et al., 2023; Chevallier, 2013) may help us to significantly reduce our computational costs and run Monte Carlo ensembles. While these new features have not been tested with realistic configurations yet, preliminary results are promising.**

**Specific Comments**

The abstract lists percentage changes, but could an estimate in the uncertainties of these changes be included?

We would be willing to make the change but it is not clear what part of the abstract the Referee is referring to. We provide an answer depending on where it could be.

Line 12, we provide the percentage of total emission increase attributed to FF and AGW in INV_REF. If we provided an uncertainty based on all the sensitivity inversions for these numbers, it would mean that we give the same weight to, e.g., the reference inversion and the inversion testing the OH from Turner et al. (2017). We prefer not to do so because it would change the take-home message. Also, if we remove the sensitivity experiments where the OH from Turner et al. (2017) is used and where the source signatures are not optimized, the results are very much alike (±1%). We could add it but we do not think it is relevant here as we believe the reader will not easily understand where this uncertainty comes from (at least without reducing the clarity of the abstract).

Line 18, we provide an update of these values for INV_LOCKED compared to INV_REF. We cannot provide an uncertainty because we are just comparing two distinct inversions.

Line 42 – it would be useful to indicate the best estimates of the percentage of methane that is removed by each of these sinks.

We agree with this comment and added information to the text.

**This species is mainly removed from the atmosphere through oxidation by the radical hydroxyl (OH), which represents about 92 % of the total sink (Saunois et al., 2020; Thanwerdas et al., 2022b). Other sinks such as oxidation by atomic oxygen ($O^1D$), chlorine (Cl) and methanotrophs in the soil contribute about 1.5 %, 1.5 %, and 5 %, respectively, to the total removal of $CH_4$ (Saunois et al., 2020; Thanwerdas et al., 2022b). Note that these numbers come with non-negligible uncertainties and vary from one study to another.**

Line 89 – A range of the source values for δ13C is given, but not for δD. It would be useful to give an indication of this. The introduction should also explain how the sink processes affect the isotopic composition of atmospheric methane, noting the strong isotopic fractionation by the sinks in particular for δD.

We agree with this comment and added a full paragraph on the fractionation effect in the introduction.

**Variations in atmospheric isotopic composition are not caused by sources only. Reactions between sink species (OH, $O^1D$ and Cl) and CH4 have rates that depend on the isotopologue. This effect is called fractionation and is represented, for a specific reaction, using the ratio of the reactions rates with the lightest and the heaviest member of a couple of isotopologues (e.g., $^{12}CH_4$ and $^{13}CH_4$). The fractionation effect explains why the atmospheric isotopic composition is not equal to the flux-weighted mean source signature of all the $CH_4$ sources. It acts at shifting this mean source composition towards less negative values when $CH_4$ enters the atmosphere and gets removed by the sinks. This effect is particularly important for $\delta_{source}(D,CH_4)$ because the flux-weighted mean source signature and the observed isotopic composition are approximately −330 ‰ and −95**

‰, respectively (Sherwood et al., 2017). For δ($^{13}$C,CH$_4$), this effect is smaller, shifting the source signature from approximately −53.6 ‰ to −47.3 ‰ in the atmosphere (Sherwood et al., 2017).

It should be noted that although the model is run from 1998 to 2019 the δD isotope data are only available for a part of this so the full run cannot be constrained with this isotope.

We agree with this comment. Although we included the period of time covered by the δ(D,CH$_4$) isotope data when introducing the observations, we did not emphasize enough that δ(D,CH$_4$) isotope data is scarce over the full assimilation window. We added a comment when introducing the corresponding sensitivity experiment and also when presenting the results.

*Note that δ(D,CH$_4$) observations spans only the period from 2005 to 2010 and therefore, the full run cannot be fully constrained by this data.*

Table 1 – how well are these fractionation factors known? I think there needs to be some discussion of this as it could have a significant effect on the results if slightly different values are used. Was this considered in the sensitivity studies?

At present, the literature provides very few estimates of the fractionation factors for the different methane sinks.

For Cl+CH$_4$ and O($^1$D)+CH$_4$, to the best of our knowledge, the fractionation factors were only estimated by Saueressig et al. (1995, 1996, 2001).

For OH, Burkholder et al. (2019) recommends using the Saueressig et al. (2001) rates but suggests increasing the uncertainty in the OH fractionation to include Cantrell et al. (1990) as a possibility. As shown by Basu et al. (2022), switching to the value from Cantrell et al. (1990) has a large influence on the results, although they do not optimize source signatures in their setup. In our paper, we preferred to allocate computation time to a sensitivity inversion testing a different OH field rather than testing a different OH fractionation factor. Furthermore, Saueressig et al. (2001) indicate their data is of considerably higher experimental precision and reproducibility than previous studies, in particular Cantrell et al. (1990). Therefore, we decided not to test another value.

All these estimates come with uncertainty ranges that we could also consider in our inversions (e.g. with a Monte-Carlo approach). In our case, the main limitation remains the large computational cost of one inversion. With the new system being developed at the moment, we hope to be able to run more sensitivity tests as the computation time is largely reduced. Then, it will be easier to test different fractionation values.
We modified the text to include a discussion about the fractionation factors.

**The fractionation effect must also be represented in the modelling framework. Table 1 provides the fractionation coefficients applied for each loss reaction. For the OH sink, we adopted the estimate derived by Saueressig et al. (2001). Burkholder (2020) recommends using the Saueressig et al. (2001) rates but suggests increasing the uncertainty in the OH fractionation to account for Cantrell et al. (1990) estimate (1.0054). As shown by Basu et al. (2022), switching from Saueressig et al. (2001) to Cantrell et al. (1990) estimates has a large influence on the results, despite the authors do not optimize source signatures in their setup. As Saueressig et al. (2001) indicate their data is of considerably higher experimental precision and reproduci-**

**bility than previous studies, in particular Cantrell et al. (1990), we prefer to allocate computation time to a sensitivity inversion testing a different OH field rather than testing a different OH fractionation coefficient. In addition, these estimates of fractionation coefficients come with uncertainty ranges that we could also consider in our inversions (e.g. with a Monte-Carlo approach). In our case, the main limitation remains the large computational cost of one inversion (see Sect. 2.8). In the future, we hope to be able to increase the number of sensitivity tests and account for this uncertainty. For this work, the values we adopt are the best estimates for each fractionation coefficient.**

There is a good explanation of how the isotopic signatures were selected in section 2.4. I agree with the authors that the regional uncertainties in δ13C that have been used are probably overestimates, but it is difficult to improve this estimate. How useful would more data on isotopic signatures of δD be, and are there particular sectors for which this is most lacking?

Data for $\delta(D,CH_4)$ isotopic signatures is mostly missing, similarly to $\delta(^{13}C,CH_4)$, for non-fossil source, as for. Additionally, the range of observed values are larger than those of $\delta(^{13}C,CH_4)$ and therefore leads to larger uncertainties.

In our work, we decided to use the same estimates as those adopted in Warwick et al. (2016) because we wanted to see how our simulated concentrations would compare to theirs. However, more recent (yet very similar) estimates are provided by Sherwood et al. (2021). Using this database, we might have been able to derive a value for each continental region for fossil fuel data. However, it would have been more problematic for non-fossil data as the $\delta(D,CH_4)$ data is very limited. It will be even more complicated if we want to derive a proper estimate of uncertainties in the future...

We added a sentence to address this lack of data for non-fossil sources in Section 2.4.

**For future studies, additional $\delta(D,CH_4)$ data would be invaluable to derive realistic regional estimates, especially for non-fossil sources.**

In section 2.9 it is mentioned that just the period 2002 to 2014 is included in the run. I think it should be mentioned in the abstract what time period is being considered, because if there have been significant changes from a particular source since (e.g. increase in emissions from wetland, such as in Zhang et al., 2023) then that wouldn't have been seen.

We agree with this comment added a sentence to the abstract to increase clarity.

**Here, we investigate the post-2007 increase in atmospheric CH₄ using the differences between 2002-2007 and 2007-2014.**

Just a few years of δD data observational data have been used. Is there any evidence of an isotope trend in this, or is the time period too short to see this? Perhaps this should be commented on in section 3.1.

We agree with this comment and included a new figure (see below) to show a comparison between δ(D,CH4) observations and prior and posterior simulations from INV_DD. Additionally, we investigated the linear trends that can be extracted from the δ(D,CH4) data. A new paragraph has been added to the section 3.1.

**For the sake of completeness, we also provide the comparison between $\delta(D,CH_4)$ observations and prior and posterior simulations from INV_DD in Appendix B (Fig. B1). After the inversion, simulations capture much better the observed $\delta(D,CH_4)$**

data, reducing the RMSE from 9.3 ‰ to 1.2 ‰. Although δ(D,CH₄) data is much more limited than δ($^{13}$C,CH₄) data, linear regressions show a small negative trend (−0.23 ‰ a⁻¹) between 2005 and 2009. Additionally, the trend is positive be-tween2005 and 2007 (+0.86 ‰ a⁻¹) and negative between 2007 and 2009 (−0.47 ‰ a⁻¹). It shows δ(D,CH₄) observations might also carry some information about the post-2007 CH4 renewed growth. However, the Pearson's correlation coefficients (r) suggest a low confidence in these trend estimates.

[Figure]

**Figure B1.** The left panel shows a comparison between $\delta(D, CH_4)$ observations, prior and posterior simulations. The right panel shows linear regressions applied on the monthly- and globally-averaged $\delta(D, CH_4)$ observations. We performed three linear regressions: 1) one over the full data period 2005-05 to 2009-07 (brown line), 2) one over the period 2005-05 to 2007-01 (violet line) and one over the period 2007-01 to 2009-07 (blue line). For each linear regression, the coefficient and the $R^2$ score are displayed in the legend. Note that the x-axis stops before 2010, as we have only selected months with sufficient data for the average to be representative of the whole globe.

Line 5: 'amount fractions'. What is the reason to divert from the common use of mole fractions? 'amount' is much less clearly defined as 'mole', so I do not see the advantage of using 'amount'.

The editor requested multiple changes before releasing the preprint to ensure the paper follows the SI System of Units and the Recommendations of the IUPAC Green Book. In our opinion, the most significant changes are:

- "mole fractions" have been modified to "amount fractions"
- "ppbv" have been modified to "nmol mol⁻¹"
- "molec. cm⁻³" have been modified to "cm⁻³"

The editor provided the following justification:
*A better name for "mole fractions" is "amount fraction" (just as "mass fraction" is used, rather than "kilogram fraction", i.e., the name of the quantity is "amount", rather than "mole").*

Note that we do not fully agree with these changes considering that most papers in ACP do not follow these recommendations. However, we understand the request because if every paper follows the same guidelines, then we suppose clarity and consistency across the papers would benefit from it.

Line 84: 'PDB'. Confusion should be avoided about isotopic standards. Vienna PDB is used as the standard according to the text, but the value of the old PDB standard is mentioned. Furthermore, the notation in the equations suggests that the old PDB is used instead of VPDB.

We apologize for this mistake, it is indeed the old PDB standard that we used. At the time we started to create the inversion system, we decided to use the old standard because it was commonly adopted in the literature (e.g., Houweling et al., 2000; Lassey, 2007; Warwick et al., 2016; Strode et al. 2020). We tested the difference between the two standards and as long as the same standard is used for all the operations (conversion of fluxes and signatures to isotopic fluxes, conversion of simulated isotopic mixing ratios back to simulated methane mixing ratio and isotopic composition, …), there is no influence on the results. Strode et al. (2020) report similar findings.

However, as the Global Monitoring Division (GMD) observations now use the V-PDB standard with a value of 0.011183, we will move to this new standard for the future studies.

We modified the text accordingly.

Line 230: How are source signatures discretized in time in the control vector?

If the Referee is referring to $\delta_{source}(^{13}C,CH_4)$ source signatures, then the information is provided at the beginning of Section 2.4 and also in the answer to the first general comment above.

If the Referee is referring to $\delta_{source}(D,CH_4)$ source signatures, we indeed realized that we did not provide the information. As it is more difficult to find information about $\delta_{source}(D,CH_4)$ source signatures, we decided to optimize only one scaling factor for the entire period, for all the categories. We apologize for not providing this information before.

To make this information clearer, we included additional columns in Table 2 and we slightly modified the text.

Line 278: Clarify the meaning of 'aggregated' here. Do you mean 'averaged'?

Yes, this is a poor choice of wording. We indeed mean 'averaged'. We modified the text accordingly.

Line 291: How are initial conditions discretized spatially in the control vector?

We apologize for not including this information in the preprint.
We added this information to the text.

**To optimize the initial conditions, the globe is regularly discretized using latitudinal and longitudinal bands. A step of 30° degrees is applied to generate the bands, resulting in 6 x 12 = 72 regions. One scaling factor is optimized for each of these regions.**

Line 330: 'using a sequence of short periods' Some discussion of equilibration times in the context of isotopic inversions is presented in Houweling et al, 2017. To get this issue resolved once and for all, best would be to do an inversion using short and long-time windows to determine of the longer equilibration time of isotopes really makes a difference in inversions that do optimize the initial condition. Note that the equilibration time of CH4 itself is also significantly longer than then spin-up/down-time that is used. If that is not a problem in inversions using only CH4 data, then why would it be a problem in inversion that use isotopic data?

We agree this issue should be resolved once and for all. This is what we intend to do in the future (Line 330: *Such a parallelization method should therefore be rigorously validated before interpreting its output*.).

However, if we perform such a test, certain questions still need to be answered:

- Should we optimize the initial conditions of each sub-window?
- What is the optimal overlap?
- What is the optimal duration of a sub-window?

Houweling et al. (2017) mentioned that the effect of long relaxation times on the results can be reduced by optimizing the initial conditions. So we believe the initial conditions of each sub-window should be optimized. Also, Chevallier et al. (2013) developed a more complex methodology than in Basu et al. (2022) to parallelize an inversion within a variational framework. For a specific window, this methodology accounts for the global concentration increment generated by the previous sub-windows. This is not always accounted for when splitting the full assimilation window into multiple sub-windows.

This method consists of breaking down the full assimilation window into multiple sub-windows, and running smaller inversions in parallel for each sub-window. If source signatures remain constant, we expect the results to closely resemble those of a longer-term window inversion. Conversely, if source signatures are optimized, the influence of initial conditions on the atmospheric isotopic composition might persist over a time that is larger than the length of the sub-window. In this case, source signatures might remain unchanged and the results could be impacted. (for more on this topic, refer to the discussion on 'spin-up' in the following response).

To summarize, we believe this kind of experiment definitely requires a dedicated study, as the methodology is not unique and the parameters influencing the outcomes are numerous. As highlighted in our paper, we cannot consider any further inversion study with isotopic constraints without also focusing on assessing the impact of such a parallelization method on posterior results (with or without optimizing source isotopic signatures).

We added some information to the text in Sect. 2.9 (Computational aspects).

**While the number of CPU hours needed for these complex inversions remains reasonable, the overall runtime is excessive. It is therefore an important limitation of our system. Further developments on parallelization methods are being implemented to enable a significant reduction of the computational cost (e.g., Chevallier, 2013). This method consists of breaking down the full assimilation window into multiple sub-windows, and running smaller inversions in parallel for each sub-window. If source signatures remain constant, we expect the results to closely resemble those of a longer-term window inversion. Conversely, if source signatures are optimized, the influence of initial conditions on the atmospheric isotopic composition might persist over a time that is larger than the length of the sub-window. In this case, source signatures might remain unchanged and the results could be impacted. Therefore, it is crucial to rigorously validate this parallelization method before interpreting its results.**

Line 335: 3-4 years is still within the range of transport spin up times (for the largest time scales of atmospheric mixing). Tans et al discuss much longer time scales (see my previous comment)

A spin-up time, for an inversion, typically refers to a period at the beginning of the inversion where the errors in the assumed initial concentration field might influence the posterior fluxes. If the spin-up is taken too short, the inversion may fit the data by compensating errors in the initial condition with artificial emission adjustments.

Here, the initial concentrations are optimized so we might think that this effect is reduced. However, for 2-3 years after the start of the inversion, it is easier for the system to optimize the initial fields rather than the isotopic signatures. In Thanwerdas et al. (2022a), we found that the optimized source signatures slowly move away from the prior value over time. After 2-3 years, the posterior value finally reaches a new and rather stable state. As the influence of initial conditions on the isotopic composition decrease, the system prefers to optimize the source signatures, hence slowly reaching the posterior value. Naturally, this effect does not occur if we have only one scaling factor for the whole period. Also, this effect is significant because, in our setups, the system infers large adjustments to the source signatures in response to the large prescribed uncertainties. We expect this effect to be smaller if the prior uncertainties are reduced.

Also, we are not suggesting that the relaxation time and the spin-up should be equal. Houweling et al. (2017) mentions that the perturbation recovery time for $CH_4$ is also much longer than the spin-up time that is used in inversions using only $CH_4$ data (i.e., without using isotopes). Because the relaxation time is larger for the isotopic composition than for $CH_4$ alone, hence the spin-up time is necessarily larger.

We modified the explanation about the spin-up effect in Sect. 2.8 (Analysis period).

**Thanwerdas et al. (2022a) suggested that the results of the inversion should be discarded up to 2-3 years after the beginning and 2-3 years before the end of the assimilation window. For the beginning of the window, although the term "spin-up" is not quite appropriate here because the cause of the discard is slightly different, the outcome is still similar. A spin-up time, for an inversion, typically refers to a period at the beginning of the inversion where the errors in the assumed initial concentrations field might influence the posterior fluxes. If the spin-up is taken too short, the inversion may fit the data by compensating errors in the initial condition with artificial emission adjustments. If the initial concentrations are also optimized, which is the case here, this effect can be reduced (Houweling et al., 2017). However, source signatures are optimized and for a certain period after the start of the inversion, it is easier for the system to optimize the initial $\delta(^{13}C,CH_4)$ fields rather than the isotopic signatures to fit the $\delta(^{13}C,CH_4)$ data. Thanwerdas et al. (2022a) found that the optimized source signatures slowly move away from the prior value over time. After 2-3 years, the posterior value finally reaches a new and rather stable state. In other words, as the influence of initial conditions on the isotopic composition decrease, the system prefers to optimize the source signatures, hence slowly reaching the posterior value. As the equilibration time for $\delta(^{13}C,CH_4)$ and $\delta(D,CH_4)$ is larger than the $CH_4$ equilibration time (Tans, 1997), this affects more the source signatures than the fluxes. For the end of the inversion, it is mainly caused by a lack of constraints at the end of the period so using the term "spin-down" is correct.**

Line 385: Ostler et al indeed do not find a strong influence when replacing MIPAS, but they show a significant sensitivity to scaling MIPAS to ACE-FTS and conclude that the accuracy of these satellite measurements is probably not good enough yet to answer this question.

We agree and changed the formulation to include your remark.

**Ostler et al. (2016) reported that model errors in simulating stratospheric CH$_4$ amount fractions could contribute to the XCH$_4$ bias. However, they did not find a strong improvement for LMDz and TM5 when replacing model simulations with MIPAS (Michelson Interferometer for Passive Atmospheric Sounding) stratospheric CH4. While this suggests that the biases in the GOSAT-simulation presented here are probably not caused by stratospheric discrepancies, it is important to note that the same authors conclude that current satellite measurements of stratospheric CH$_4$ may lack the precision necessary to eliminate these biases.**

**If we assume that the biases are solely the result of emission discrepancies, our findings indicate that the estimated tropical posterior emissions from our inversions might still be underestimated.**

Figure 6: Which simulation is this?

The corresponding inversion is INV_REF.
We modified the caption accordingly.

Line 475: What is the justification of wetland flux anomalies persisting for as long as 4 years?

We agree that this part of Section 3.5 is poorly written. We changed the formulation and explained the WET anomalies better.

**However, posterior global WET emissions show negative anomalies between 1998 and 1999 and positive anomalies between 1999 and 2004. These anomalies are mainly located in South America, where about 30 % of WET emissions originate. Zhang et al. (2018) suggested that the 1998-2000 ENSO (El Niño-Southern Oscillation) caused negative anomalies of WET emissions between 1998 and 2000 because of El Niño and subsequent positive anomalies between 2000 and 2002 because of La Niña. The fact that positive anomalies persist until 2004 rather than 2002 in our posterior emissions cannot be easily explained. Also, the positive anomalies last almost 5 years in total, which is not consistent with the 2-3 years inferred by Zhang et al. (2018). As AGW emissions are also large in South America, the inversion system might be wrongly attributing large but decreasing emissions between 2002 and 2004 to WET emissions rather than AGW emissions. If the period 2002-2004 that exhibit large positive anomalies is discarded, we find a small increase of 0.3 Tg a$^{-1}$ in global WET emissions between 2004-2007 and 2007-2014, therefore more consistent with the studies mentioned before.**

line 643: For comparison to future studies, it would be useful to provide absolute emission increases in addition to fractional contributions to the total increase.

We believe the requested information is already provided in the upper part of Table 5 for all inversions (rows) and all categories (columns).

line 654: 'enriched' and 'depleted' seem reversed in this sentence.

Yes indeed, we apologize for this mistake and modified the text.

**Technical corrections**

Line 36 – capital C for Climate

This has been modified.

Lines 81,82 – explain what R13 and RD represent (ratios of 13C/12C and 2H/1H in the sample). Also note that these δ values are multiplied by 1000 to give values in ‰.

First part has been added.

However, we do not agree with the second part. Coplen et al. (2011) explains that "The factor 1000 is an extraneous numerical factor and should be deleted". But we added that the δ values are expressed in ‰.

Line 84, RPDB should be 1.12372 x 10-2

Yes, indeed. Apologies for this mistake. We modified it.

Line256, Arctic (not Artic). You should mention that INSTAAR is part of University of Colorado.

This has been modified.

Data availability: Is the δD data from INSTAAR archived anywhere?

We apologize for not including the δD in the preprint.  The data for δD is archived where $CH_4$ and δ13C data also are.

line 188: remove 'Database'

We suppose the Referee refers to both EDGAR and GFED. This has been modified for both.

line 293: 'setup IS detailed'

This has been modified.

line 374: section 3.4.

This has been modified.

line 447: 'applies' io 'apply'

This has been modified.

line 457: 'who suggest' io 'that suggests"

This has been modified.

**REFERENCES**

Basu, S., Lan, X., Dlugokencky, E., Michel, S., Schwietzke, S., Miller, J. B., Bruhwiler, L., Oh, Y., Tans, P. P., Apadula, F., Gatti, L. V.,Jordan, A., Necki, J., Sasakawa, M., Morimoto, S., Di Iorio, T., Lee, H., Arduini, J., and Manca, G.: Estimating emissions of methane consistent with atmospheric measurements of methane and δ13C of methane, Atmospheric Chemistry and Physics, 22, 15 351–15 377,https://doi.org/10.5194/acp-22-15351-2022, 2022.

Burkholder, J. B., Sander, S. P., Abbatt, J. P. D., Barker, J. R., Cappa, C., Crounse, J. D., Dibble, T. S., Huie, R. E., Kolb, C. E., Kurylo, M. J., Orkin, V. L., Percival, C. J., Wilmouth, D. M., and Wine, P. H.: Chemical Kinetics and Photochemical Data for Use in Atmospheric Studies, Evaluation No. 19, Tech. Rep. 19-5, Jet Propulsion Laboratory, Pasadena, CA, https://jpldataeval.jpl. nasa.gov/pdf/NASA-JPL%20Evaluation%2019-5.pdf (last access: 29 November 2022), 2019.

Cantrell, C. A., Shetter, R. E., McDaniel, A. H., Calvert, J. G., Davidson, J. A., Lowe, D. C., Tyler, S. C., Cicerone, R. J., and Greenberg, J. P.: Carbon kinetic isotope effect in the oxidation of methane by the hydroxyl radical, Journal of Geophysical Research: Atmospheres,95, 22 455–22 462, https://doi.org/10.1029/JD095iD13p22455, 1990.

Chevallier, F.: Impact of correlated observation errors on inverted CO2 surface fluxes from OCO measurements, Geophys. Res. Lett., 34, L24804, doi:10.1029/2007GL030463, 2007.

Chevallier, F.: On the parallelization of atmospheric inversions of CO2 surface fluxes within a variational framework, Geosci. Model Dev., 6, 783–790, https://doi.org/10.5194/gmd-6-783-2013, 2013.

Coplen, T.: Guidelines and recommended terms for expression of stable-isotope-ratio and gas-ratio measurement results. Rapid communications in mass spectrometry, vol. 25, no 17, p. 2538-2560, https://doi.org/10.1002/rcm.5129, 2011

Houweling, S., Dentener, F., and Lelieveld, J.: Simulation of preindustrial atmospheric methane to constrain the global source strength of natural wetlands, J. Geophys. Res.-Atmos., 105, 17243–17255, https://doi.org/10.1029/2000JD900193, 2000.

Houweling, S., Bergamaschi, P., Chevallier, F., Heimann, M., Kaminski, T., Krol, M., Michalak, A. M., and Patra, P.: Global inverse modelingof CH4sources and sinks: an overview of methods, Atmospheric Chemistry and Physics, 17, 235–256, https://doi.org/10.5194/acp-17-235-2017, 2017.

Lassey, K. R., Etheridge, D. M., Lowe, D. C., Smith, A. M., and Ferretti, D. F.: Centennial evolution of the atmospheric methane budget: what do the carbon isotopes tell us?, Atmos. Chem. Phys., 7, 2119–2139, https://doi.org/10.5194/acp-7-2119-2007, 2007.

Meirink, J. F., Bergamaschi, P., and Krol, M. C.: Four-dimensional variational data assimilation for inverse modelling of atmospheric methane emissions: Method and comparison with synthesis inversion, Atmos. Chem. Phys., 8, 6341–6353, doi:10.5194/acp-8-6341-2008, 2008b

Saunois, M., Stavert, A. R., Poulter, B., Bousquet, P., Canadell, J. G., Jackson, R. B., Raymond, P. A., Dlugokencky, E. J., Houweling, S.,Patra, P. K., Ciais, P., Arora, V. K., Bastviken, D., Bergamaschi, P., Blake, D. R., Brailsford, G., Bruhwiler, L., Carlson, K. M., Carrol,M., Castaldi, S., Chandra, N., Crevoisier, C., Crill, P. M., Covey, K., Curry, C. L., Etiope, G., Frankenberg, C., Gedney, N., Hegglin, M. I., Höglund-Isaksson, L., Hugelius, G., Ishizawa, M., Ito, A., Janssens-Maenhout, G., Jensen, K. M., Joos, F., Kleinen, T., Krummel,P. B., Langenfelds, R. L., Laruelle, G. G., Liu, L., Machida, T., Maksyutov, S., McDonald, K. C., McNorton, J., Miller, P. A., Melton,J. R., Morino, I., Müller, J., Murguia-Flores, F., Naik, V., Niwa, Y., Noce, S., O'Doherty, S., Parker, R. J., Peng, C., Peng, S., Peters, G. P.,Prigent, C., Prinn, R., Ramonet, M., Regnier, P., Riley, W. J., Rosentreter, J. A., Segers, A., Simpson, I. J., Shi, H., Smith, S. J., Steele, L. P., Thornton, B. F., Tian, H., Tohjima, Y., Tubiello, F. N., Tsuruta, A., Viovy, N., Voulgarakis, A., Weber, T. S., van Weele, M., van der Werf, G. R., Weiss, R. F., Worthy, D., Wunch, D., Yin, Y., Yoshida, Y., Zhang, W., Zhang, Z., Zhao, Y., Zheng, B., Zhu, Q., Zhu, Q., and Zhuang,Q.: The Global Methane Budget 2000–2017, Earth System Science Data, 12, 1561–1623, https://doi.org/10.5194/essd-12-1561-2020, 2020.

Saueressig, G., Bergamaschi, P., Crowley, J. N., Fischer, H., and Harris, G. W.: Carbon kinetic isotope effect in the reaction of CH4with Clatoms, Geophysical Research Letters, 22, 1225–1228, https://doi.org/10.1029/95GL00881, 1995.

Saueressig, G., Bergamaschi, P., Crowley, J. N., Fischer, H., and Harris, G. W.: D/H kinetic isotope effect in the reaction CH4+Cl, GeophysicalResearch Letters, 23, 3619–3622, https://doi.org/10.1029/96GL03292, 1996.

Saueressig, G., Crowley, J. N., Bergamaschi, P., Brühl, C., Brenninkmeijer, C. A. M., and Fischer, H.: Carbon 13 and D kinetic isotope effects in the reactions of CH4 with O(1D) and OH: New laboratory measurements and their implications for the isotopic composition of stratospheric methane, Journal of Geophysical Research: Atmospheres, 106, 23 127–23 138, https://doi.org/10.1029/2000JD000120,2001.

Sherwood, O. A., Schwietzke, S., Arling, V. A., and Etiope, G.: Global Inventory of Gas Geochemistry Data from Fossil Fuel, Microbial andBurning Sources, version 2017, Earth System Science Data, 9, 639–656, https://doi.org/10.5194/essd-9-639-2017, 2017.

Sherwood, O. A., Schwietzke, S., and Lan, X.: Globalδ13C-CH4source signature inventory 2020., https://doi.org/10.1029/2021GB007000,2021.

Strode, S. A., Wang, J. S., Manyin, M., Duncan, B., Hossaini, R., Keller, C. A., Michel, S. E., and White, J. W. C.: Strong sensitivity of the isotopic composition of methane to the plausible range of tropospheric chlorine, Atmos. Chem. Phys., 20, 8405–8419, https://doi.org/10.5194/acp-20-8405-2020, 2020.

Thanwerdas, J., Saunois, M., Berchet, A., Pison, I., Vaughn, B. H., Michel, S. E., and Bousquet, P.: Variational inverse modeling within the Community Inversion Framework v1.1 to assimilate δ13C(CH4) and CH4: a case study with model LMDz-SACS, Geoscientific Model Development, 15, 4831–4851, https://doi.org/10.5194/gmd-15-4831-2022, 2022a.

Thanwerdas, J., Saunois, M., Pison, I., Hauglustaine, D., Berchet, A., Baier, B., Sweeney, C., and Bousquet, P.: How do Cl concentrations matter for the simulation of CH4 and δ13C(CH4) and estimation of the CH4 budget through atmospheric inversions?, Atmospheric Chemistry and Physics, 22, 15 489–15 508, https://doi.org/10.5194/acp-22-15489-2022, 2022b

Turner, A. J., Frankenberg, C., Wennberg, P. O., and Jacob, D. J.: Ambiguity in the causes for decadal trends in atmospheric methane and hydroxyl, Proceedings of the National Academy of Sciences, 114, 5367–5372, https://doi.org/10.1073/pnas.1616020114, 2017

Warwick, N. J., Cain, M. L., Fisher, R., France, J. L., Lowry, D., Michel, S. E., Nisbet, E. G., Vaughn, B. H., White, J. W. C., and Pyle, J. A.: Using δ13C-CH4 and δD-CH4 to constrain Arctic methane emissions, Atmos. Chem. Phys., 16, 14891–14908, https://doi.org/10.5194/acp-16-14891-2016, 2016.

---

## Author Response (AR2)

**Response to Editor's Comments**

**Joël Thanwerdas[1,a*], Marielle Saunois[1], Antoine Berchet[1], Isabelle Pison[1], and Philippe Bousquet[1]**

[1]Laboratoire des Sciences du Climat et de l'Environnement, CEA-CNRS-UVSQ, IPSL, Gif-sur-Yvette, France.

[a]now at: Empa, Swiss Federal Laboratories for Materials Science and Technology, Dübendorf, Switzerland.

Again, we thank the two referees and the editor for reading attentively our paper and making comments that allow us to improve it. We provide here a response to the comments received. Comments are in red. For each comment, an answer is provided in normal text, *citations from the text are in italic* and **the modifications from the new version of the manuscript are provided in bold and small text**. Note that modifications have been included only when deemed substantial enough. Also, the incorrect sentence spotted by Referee #1 has been removed, it was a proofreading mistake from last revision.

Attached to this response, we also provide the new version of the manuscript and a track-changes document.

1) I understand that a mathematically rigorous uncertainty analysis is computationally expensive, but even so, some uncertainty estimate is required. I don't expect every number in the paper to have an uncertainty, but at the least the values in the abstract and conclusions (and the values they originate from) should come with some "educated guess" of how reliable they might be. As an example, the 51 %:49 % split between fossil and agriculture/waste source suggests a level of uncertainty of the order of 1 %. Is that realistic? Or could it be 60 %:40 %? Or 40 %:60 %? Similarly, for the relative increases stated further below. To be clear, the "educated guess" should also appear in the abstract itself.

An estimate of the uncertainty calculated using the sensitivity tests performed for this study has been added to the abstract, the main text and the conclusion. Here is the sentence added to the abstract:

**Additionally, some other sensitivity tests have been performed. While prescribed OH inter-annual variability can have a large impact on the results, assimilating δ(D, CH$_4$) observations in addition to the other constraints have a minor influence. Using all the information derived from these tests, the net increase in emissions is still primarily attributed to fossil sources (50 ± 3 %) and agriculture and waste sources (47 ± 5 %).**

2) The potential value of deuterium isotope ratio measurements as constraint on sink magnitudes and sink partitioning (between OH and Cl) should be discussed (see comment from referee #2).

A discussion has been added at the end of Section 3.7.

**As including δ(D, CH$_4$) in the inversion doubles the computational cost compared to a setup like INV_REF, we recommend not assimilating δ(D, CH$_4$) in our system until either the computational cost can be reduced, more observations become available or lower uncertainties are established. However, a hypothetical network of δ(D, CH$_4$) measurements, obtained at a reasonable frequency and spanning a longer period of time could efficiently complement δ($^{13}$C, CH$_4$) observations and provide a wealth of information (Rigby et al., 2012). More specifically, reactions with OH, O1D and Cl have fractionation coefficients that depend on the isotope. Therefore, incorporating δ(D, CH$_4$) constraints might help to disentangle the effects of the associated sinks and provide additional insights into the global sink and its mixture. However, optimizing the sinks introduces additional degrees of freedom and complexifies the inverse problem. With the current system and at such a high resolution for the optimized variables, we recommend against the simultaneous optimization of both the source signatures and the sinks. However, a coarser resolution for the optimized variables, or at least for the sink, might be able to accommodate a simultaneous optimization.**

3) The compensating effects of variations in source magnitude and isotopic signature should be discussed (see comment from referee #2).

A discussion has been added at the end of Section 3.4.

**Figure B4 and Figure B5, in Appendix B, show the full temporal variations for prior and posterior source signatures. For FF, high variations indicate a change in activities associated to fossil fuel extraction, e.g. switching from one location with a specific signature to another, transitioning from one fuel type (oil, gas, coal) to another or a combination of both. For example, the substantial shift around 2009 in the United States was caused by a large increase in emissions from the extraction of natural gas. As we chose not to prescribe temporal error correlations between different years, the system is free to optimize each year independently to better fit $\delta(^{13}C,CH_4)$ observations. For certain continental regions, such as Africa, Temperate Asia or South Asia, the inter-annual variability of the source signature adjustments is large and rather unrealistic, especially when compared to the emission adjustments in the same regions (see Fig. 7). It is unlikely that these changes occurred without detectable changes in emissions in the same areas, especially considering Temperate Asia, which exhibits larger emissions from FF than in the United States. These results suggest the need for prescribing yearly temporal error correlations to dampen this artificial inter-annual variability. However, the example from the United States also indicate that large changes can occur and it is reasonable to assume, considering the lack of isotopic data, that such changes might go unnoticed by the prior data for other regions. Therefore, while implementing stronger temporal correlations could be a solution to mitigate unrealistic inter-annual variability for this category, it diminishes the likelihood of detecting potential substantial changes that remain undetected by the prior data. Nevertheless, it might be sufficient to reduce the prescribed uncertainties in the source signatures in order to balance out the pressure applied by the system on the emissions and the source signatures. Overall, the same reasoning applies to AGW, although there is no evidence in the prior data that AGW source signatures can change as rapidly as FF. Due to the scarcity of existing data on the temporal variability of source signatures, designing a data-driven methodology to estimate potential temporal correlations, especially at the regional scale, remains highly challenging. Investigating the correlations that the system creates between the uncertainties associated to source signatures and fluxes could offer a promising avenue for extending the analysis. Due to the high computational cost of an inversion performed with our inversion system, it is impossible to derive robust posterior uncertainties. This impossibility is a major drawback and additional studies with this system cannot be performed in the future without tackling this issue.**

4) Data sets cannot just be available on request. Please make all relevant data available in a suitable permanent repository (i.e., the input emissions and the other relevant data from this study.)

As requested, a permanent repository with a DOI (10.5281/zenodo.10390430) has been created to store all the relevant data. Two datasets obtained from the authors of Wang et al. (2021) and Saunois et al. (2020) were not stored on permanent repositories. After contacting them, these authors agree to store their data on our permanent repository, provided that the origin of this data is properly acknowledged. For some reason, the modification of the data availability paragraph does not appear in the track-changes document.

5) In section 3.2 and the caption of Fig. 5, please replace XCH4 with column-average CH4 amount fraction.

- In Eqs. 1 and 2 and the line below, replace square brackets with X(12CH4) etc. (with X in italics)

- Replace the y-axis label of Fig. 4 d with X(CH4)/(nmol mol–1)

- l. 164: Write "atmospheric X(CH4)" before the delta symbols

- In section 2.5.2, define $\bar{X}$(CH4) or <X(CH4)> as column-average amount fraction, but explain in the text that it is "often referred to with the symbol XCH4", and use either $\bar{X}$(CH4), <X(CH4)> or XCH4 in the remaining text.

- Change the title of section 3.2 to "Comparison of model-optimized with satellite-derived column average CH4 amount fractions"

All these comments have been taken into account and modifications have been made. Note that the change from XCH$_4$ to $\bar{X}$(CH$_4$) does not appear in the track-changes manuscript because of the Latex command.

6) The differences in the linear regressions applied to different periods of time in Fig. B1 do not appear to be statistically significant. What are the uncertainties associated with the slopes? These should be added to section 3.1 and the legend of Fig. B1, e.g., (-0.23±0.25) ‰

We agree that this had to be done. We apologize for not providing it in the last revision. We added the uncertainty (standard error) associated to each slope of the regression lines, both in Fig. B1 and in the text to show that these trends are significant.

7) Please cite the final version of Wang et al.'s paper (https://acp.copernicus.org/articles/21/13973/2021/), not the Discussion document.

We apologize for this mistake. We changed the citation.

**REFERENCES**

Saunois, M., Stavert, A. R., Poulter, B., Bousquet, P., Canadell, J. G., Jackson, R. B., Raymond, P. A., Dlugokencky, E. J., Houweling, S., Patra, P. K., Ciais, P., Arora, V. K., Bastviken, D., Bergamaschi, P., Blake, D. R., Brailsford, G., Bruhwiler, L., Carlson, K. M., Carrol, M., Castaldi, S., Chandra, N., Crevoisier, C., Crill, P. M., Covey, K., Curry, C. L., Etiope, G., Frankenberg, C., Gedney, N., Hegglin, M. I., Höglund-Isaksson, L., Hugelius, G., Ishizawa, M., Ito, A., Janssens-Maenhout, G., Jensen, K. M., Joos, F., Kleinen, T., Krummel, P. B., Langenfelds, R. L., Laruelle, G. G., Liu, L., Machida, T., Maksyutov, S., McDonald, K. C., McNorton, J., Miller, P. A., Melton, J. R., Morino, I., Müller, J., Murguia-Flores, F., Naik, V., Niwa, Y., Noce, S., O'Doherty, S., Parker, R. J., Peng, C., Peng, S., Peters, G. P., Prigent, C., Prinn, R., Ramonet, M., Regnier, P., Riley, W. J., Rosentreter, J. A., Segers, A., Simpson, I. J., Shi, H., Smith, S. J., Steele, L. P., Thornton, B. F., Tian, H., Tohjima, Y., Tubiello, F. N., Tsuruta, A., Viovy, N., Voulgarakis, A., Weber, T. S., van Weele, M., van der Werf, G. R., Weiss, R. F., Worthy, D., Wunch, D., Yin, Y., Yoshida, Y., Zhang, W., Zhang, Z., Zhao, Y., Zheng, B., Zhu, Q., Zhu, Q., and Zhuang, Q.: The Global Methane Budget 2000–2017, Earth Syst. Sci. Data, 12, 1561–1623, https://doi.org/10.5194/essd-12-1561-2020, 2020.

Wang, X., Jacob, D. J., Downs, W., Zhai, S., Zhu, L., Shah, V., Holmes, C. D., Sherwen, T., Alexander, B., Evans, M. J., Eastham, S. D., Neuman, J. A., Veres, P. R., Koenig, T. K., Volkamer, R., Huey, L. G., Bannan, T. J., Percival, C. J., Lee, B. H., and Thornton, J. A.: Global tropospheric halogen (Cl, Br, I) chemistry and its impact on oxidants, Atmos. Chem. Phys., 21, 13973–13996, https://doi.org/10.5194/acp-21-13973-2021, 2021.